# Exploratory Investigation of Handwriting Disorders in School-Aged Children from First to Fifth Grade

**DOI:** 10.3390/children10091512

**Published:** 2023-09-06

**Authors:** Clémence Lopez, Laurence Vaivre-Douret

**Affiliations:** 1Unit 1018-CESP, PsyDev/NDTA Team, National Institute of Health and Medical Research (INSERM), Faculty of Medicine, University of Paris-Saclay, UVSQ, 91190 Villejuif, France; clemence.lopez89@gmail.com; 2Department of Medicine Paris Descartes, Faculty of Health, Université Paris Cité, 75006 Paris, France; 3Clinical Neurodevelopmental Phenotyping, University Institute of France (Institut Universitaire de France, IUF), 75005 Paris, France; 4Department of Child Psychiatry, AP-HP Centre, Necker-Enfants Malades University Hospital, 75015 Paris, France; 5Department of Endocrinology, IMAGINE Institute, Necker-Enfants Malades University Hospital, 75015 Paris, France; 6Necker-Enfants Malades University Hospital, “Neuro-Développement et Troubles des Apprentissages (NDTA)”, INSERM UMR 1018-CESP, Carré Necker Porte N4, 149, Rue de Sèvres, 75015 Paris, France

**Keywords:** handwriting disorders, dysgraphia, children, semiology, neuropsychomotor assessment, neuropsychological assessment, oculomotricity

## Abstract

Handwriting disorders (HDs) are prevalent in school-aged children, with significant interference with academic performances. The current study offers a transdisciplinary approach with the use of normed and standardized clinical assessments of neuropsychomotor, neuropsychological and oculomotor functions. The aim is to provide objective data for a better understanding of the nature and the etiology of HDs. Data from these clinical assessments were analyzed for 27 school-aged children with HD (first to fifth grade). The results underline a high heterogeneity of the children presenting HDs, with many co-occurrences often unknown. However, it was possible to highlight three levels of HDs based on BHK scores: mild HD not detected by the BHK test (26% of children), moderate HD (33%) and dysgraphia (41% of children). The mild nature of the HDs not detected by the BHK test appears to occur at a relatively low frequency of the associated disorders identified during clinical evaluations. On the contrary, dysgraphia appears to be associated with a high frequency of co-occurring disorders identified in the clinical assessment, with a predominance of oculomotor disorders (55% of children), leading to visual-perceptual difficulties and a high level of handwriting deterioration. Finally, children with moderate HD have fewer co-occurrences than children with dysgraphia, but have more difficulties than children with mild HD. This highlights the importance of differentiating between different degrees of HDs that do not respond to the same semiologies. Our findings support the interest in performing a transdisciplinary and standardized clinical examination with developmental standards (neuropsychomotor, neuropsychological and oculomotor) in children with HD. Indeed, HDs can therefore be associated with a multitude of disorders of different natures ranging from poor coordination of the graphomotor gesture to a more general and more complex impairment affecting perceptual-motor, cognitive and/or psycho-affective functions.

## 1. Introduction

As children spend 31–60% of their school day writing and performing other fine motor tasks [1], the development of handwriting skills is necessary for academic success [2,3] and the proper development of self-esteem [4,5]. According to the previous version of the Diagnostic and Statistical Manual of Mental Disorders (DSM-IV-TR [6]), handwriting disorders (HDs) can be diagnosed in the case of “writing skills significantly lower than expected given chronological age, of measured intelligence and of an appropriate education”. The DSM-5 [7] describes HDs as “Impairment in written expression”, and dysgraphia is not described in the DSM-5. However, handwriting disorders (HDs) affect between 10 and 30% of school-aged children [8,9,10]. This is observed both by health professionals in clinical consultations and by teachers, who struggle to adapt to the differences in the individual rhythms of these children and their learning difficulties. In this context, current studies on handwriting and its disorders attempt to provide new fundamental knowledge on the different processes involved in the development of handwriting, while clinical and therapeutic aspects remain little explored. Thus, handwriting disorder appears as an umbrella term defining a heterogeneous class of children exhibiting graphic impairments. The study of these disorders is complex, as their understanding, both on the semiological level and on the etiological level, is still in the literature only in its early stages, and the definition of dysgraphia is unclear. Since the 1960s, it has been characterized by poor writing quality without any neurological or intellectual disorder being able to explain it [11]. This definition has been clarified by other authors, who define dysgraphia as a disorder in written language partly linked to a lack of fine motor control in the execution of motor programs [12,13]. Recently, a relevant study [14] has shown phenotyping features in the genesis of pre-scriptural gestures in children to assess handwriting developmental levels because no recent research has previously thought to study the developmental prerequisites of handwriting organization. The better the quality of the handwriting gesture, the less variation there is in the inter-segmental organization coordinated during the writing task. This makes it possible to assess handwriting development levels in the context of screening for handwriting disorders [14]. Hence, another study was able to demonstrate for the first time the immaturity of the graphomotor gesture in children with a handwriting disorder, characterized both by a lack of synergistic coordination of the different segments of the writing arm and by an impairment of the temporal and kinematic characteristics of pre-scriptural traces (decrease in fluidity characterized by an increase in the number of strokes and velocity peaks and an increase in drawing time and in-air pauses) [15,16]. The results about the impairment of the temporal and kinematic characteristics of handwriting are also corroborated by Asselborn et al. [17]. Moreover, generally, the authors highlight a lack of motor programming or of motor execution. Wann [18] suggests a motor programming defect characterized by altered temporal organization of writing (dysfunction, high pause times) due to the child’s over-reliance on visual feedback. Lopez & Vaivre-Douret [19] suggest both proprioceptive/kinesthetic feedback deficits and a disruptive effect of visual control on the quality of pre-script drawings in these children, many of whom have kinesthetic memory and visuo-spatial deficits. Thus, the ability to direct strokes would remain dependent on sensory feedback, itself insufficiently effective, leading to difficulties in achieving proactive control of handwriting. Other authors [20,21,22] suppose an impairment of the motor execution processes, which is characterized by a spatial, temporal and kinematic irregularity of the writing characteristic of dysgraphic children. This would be the consequence of excessive neuromotor noises [21,23]. However, only one study has proposed a transdisciplinary investigation of handwriting disorders (HDs) [16]. The results highlighted a typology of three groups of HDs (mild; mild-to-moderate; dysgraphia), each being associated with co-occurrences of specific neurodevelopmental dysfunctions: a co-occurrence of psycho-affective disorders that can be considered a predictor of mild and moderate HDs; a co-occurrence of tone disorders and gross coordination that can be considered a predictor of mild HDs; and a co-occurrence of visual-spatial/constructive and attentional disorders, which can be considered a predictor of the most severe (dysgraphia) and moderate HDs. More specifically, a recent study proposed a transdisciplinary investigation of HDs in a cohort of children with a developmental coordination disorder (DCD) [24]. This highlighted a significant association between neurological soft signs and the presence of dysgraphia in a sample of 65 children with DCD [24]. The dysgraphia appeared to be closely related to several specific dysfunctions of the laterality, to a minor neurological dysfunction of the pyramidal tract manifested by a distal phasic stretch reflex in the lower limb, and to slowness in digital praxis. In addition to these few studies, it is important to enrich the literature concerning the analysis of the underlying clinical functions involved in handwriting disorders.

In the present study, we offer an in-depth transdisciplinary approach with the use of normed and standardized clinical assessments of neuropsychomotor, neuropsychological and oculomotor functions. We aimed to provide objective data to better understand the nature and etiology of HDs.

## 2. Material and Methods

### 2.1. Participants

Data from a sample of 27 children with handwriting disorders (HDs) aged 6 years 2 months to 10 years 11 months were collected from primary schools (grades 1 to 5) and from the usual out-patient consultation of pediatrics at the Cochin Port-Royal Hospital and of the child psychiatry department at Necker University Hospital in Paris, France. We have chosen children in first to fifth grade, as this is the elementary school cycle in France. Our sample of children was drawn in such a way as to exclude as much as possible any comorbid disorders that might have an impact on our results, notably oral and written language disorders. In total, 14% of the children were recruited from the Necker-Enfants Malades University Hospital and 86% from an elementary school, which allowed us to obtain a sample of handwriting disorders representative of those encountered in a population of general school children. Children were excluded from the study in case of prematurity (birth < 37 weeks of amenorrhea); sensory, visual, neurological or genetic disorders; dyslexia or severe language disorder, ADHD (according to the DSM-5 criteria [7]); autism spectrum disorder; psychopathology; or motor disorder caused by an injury or accident. None of them had repeated or skipped a grade or undergone any handwriting retraining at the time of the study. The institutional research ethics committee of Paris Descartes University approved the study procedures (CER·2018-72) conducted in accordance with the Declaration of Helsinki. All parents/legal guardians of participants provided written informed consent.

### 2.2. Design and Measures

Handwriting disorders were detected by the teachers and considered objective based on an analysis of their class exercise books by an experienced psychomotor therapist. In order to assess their handwriting level, each child began to undergo the French adaptation of the standardized assessment of handwriting, the BHK scale [25] adapted from the Concise Evaluation Scale for children’s handwriting [26]. Figure 1 shows the study design previously published, with permission of the editor to reproduce it [15]. Then, a neuropsychomotor assessment (NP-MOT) was administrated and followed by other tests (psychomotor, neurovisual, neuropsychological) proposed in different orders according to each child’s motivation and time constraints. All the assessments were administered on a single day, with breaks, for a total of around six hours of testing. The examination of oculomotor functions was recorded for about twenty minutes during a second appointment on a different day. The psychomotor therapist investigator in the study administered all the tests.

### 2.3. Handwriting Assessment

In addition to the BHK test, the children performed a previously validated cycloid loop line-copying test [14,15]. Data on postural organization and inter-segmental coordination of the writing arm were systematically collected by video recording as described in previous studies [14,15]. Features about the proximal (head, trunk axis, shoulder, elbow and forearm) and distal (wrist and fingers) gestural organization of the drawing process were collected. Variables relating to the material (sheet, drawing line, pen) positioning and observational clinical variables related to the semiology of the motor characteristics of the gesture (control, pressure, synkinesis) were recorded. In addition, spatio-temporal and kinematic measures were recorded using an Anoto digital pen with Elian Research software (Version 4.2, http://www.seldage.com, accessed on 24 September 2022), for which we have developed specific algorithms to record the measures above.

### 2.4. Clinical Assessments

#### 2.4.1. Neuropsychomotor Assessment

All children performed neuropsychomotor physical tasks with the NP-MOT battery [27], including assessment of minor neurological dysfunctions (MND) exploring neurological soft signs like synkinesis (NSS). The age-standardized child assessment using the French NP-MOT test battery is applicable to children as young as 4 years old. It has been found to have adequate test–retest reliability and internal consistency. Correlation coefficients of the NP-MOT with the BOTMP [28] range from 0.72 to 0.84 for motor coordination and balance. The NP-MOT battery enables physical assessment of passive/active muscular tone of limbs and axial tone (dangling and extensibility of the wrist, shoulder, foot, heel-ear angle, popliteal angles, adductor angles, trunk), highlighting NSS by denoting the existence of MND, such as limb pyramidal dysfunction. This is complemented by the assessment of basic motor function, control and regulation in gross motor tasks, gait, balance, coordination, manual dexterity, praxis, gnosopraxis (non-meaningful hand and finger imitation of gestures), digital perception, laterality, bodily spatial integration, rhythmic and auditory attention tasks. The standardized NP-MOT battery is a developmental assessment because each subtest or milestone is scored from qualitative and quantitative viewpoints according to age, with each score converted to a standard deviation vs. mean based on normative data for age and applicable to children as young as 4 years old [29]. There is a saturation of the maturation scores between 8 and 10 years, allowing the NP-MOT to assess older children or adults.

#### 2.4.2. Psychomotor Assessment

The MABC-2 children’s movement assessment battery (second edition) [30], adapted from the American battery [31] was used to assess psychomotor skills. It aims to assess motor impairments and is divided into three categories: manual dexterity (unimanual, bimanual test and visual–motor graphic tasks), target and catch (to throw a weighted bag/ball on a target and to catch a weighted bag or a ball) and balance (static balance, walking and jumping tasks).

The gnosopraxic imitation of gestures assessment, the EMG [32,33], was used to assess distal and digital gnosopraxic efficiency and to measure the child’s ideomotor adaptation skills. It consists of performing imitations of arbitrary simple (with the hands) and complex (with the fingers) gestures in the absence of verbal command. This is an adaptation of the Bergès–Lézine assessment [34], paying particular attention to the gesture programming in the notation.

The Body Schema Test—Revised [35] was used to assess the child’s representation of his own body and the relationships between different parts of his body. The task consists of a puzzle (non-contiguous pieces) of the body and face from the front (for children aged 3 to 8) and/or from a side view (for children aged 8 and over).

Spatial and temporal identification questions were asked in order to assess the knowledge and mastery of the spatio-temporal vocabulary.

A kinesthetic memorization test consisting of a reproduction test of asymbolic postures which had previously been printed and felt with the eyes closed has been proposed in order to assess the body’s perceptual skills [24].

#### 2.4.3. Neurovisual Assessment

Neurovisual aspects, including visual gnosis; visual-perceptual, perceptual visual-motor, visuospatial and visuo-constructive skills; and oculomotricity, were assessed.

Visual perception was assessed using form-recognition tasks [36], tangled lines and visual gnosia with outlines of animals, outlines of muddled fruits.

The KABC-II Shape Recognition subtest [37] consists of recognizing and naming drawings of various objects whose images have been altered (some lines of the drawing appear while others have been erased). This item assesses the child’s ability to mentally represent the missing parts of the drawing to form a complete mental image, making it possible to name the represented object.

The Developmental Test of Visual–Motor Integration (VMI) (6th ed) [38] assessed pure perceptual abilities (the perceptual subtest of the test consisting of visual recognition of identical insignificant geometric shapes) and visuomotor integration abilities (subtest copy of the test figure consisting of the reproduction of simple and more complex insignificant geometric figures).

The NEPSY-II Arrows subtest [39] consisted of judging the direction, orientation and angles of different lines.

Rey’s complex geometric figure [40] allows the evaluation of aptitudes for perception, structuring and spatial organization (visual-spatial and visuo-constructive praxis). By copying and then reproducing from memory a complex geometric figure, the test studies the ability to structure different elements in a graphic space.

The Code and Symbols subtests of the Wechsler intelligence scale for children and adolescents WISC-IV [41] for measuring a mental processing speed index (IVT) in connection with graphomotor capacities consisted of analyzing and distinguishing non-significant signs.

The NEPSY-II Cubes subtest [39] and the Kohs block design [42] respectively assessed 3D visuo-constructive skills and visual-spatial/constructive skills.

The examination of oculomotor functions was performed using an eye-tracking device made up of two infrared cameras positioned at the level of the inner corner of each eye (Ober Consulting Eye-Tracker Eyefant^®^ [43]) recording the movements of fixation, smooth visual pursuit and horizontal and vertical eye saccades. The device records in the horizontal and vertical planes at a sampling rate of 1000 Hz, a spatial resolution of 0.1° and a linearity range of +/−35° horizontally and +/−20° vertically.

#### 2.4.4. Neuropsychological Assessment

Visual-spatial attention, sustained auditory attention and divided attention skills were assessed by a crossing test and the Childhood Attention Assessment Test (TEA-Ch [44]).

Executive functions (planning, inhibition skills, mental flexibility, working memory and verbal fluency) were assessed by the Laby 5–12 Labyrinths test [45], the NEPSY-II Categorization and Verbal Fluency subtests [39] and the Stroop’s Selective Attention Test [46].

The visual, auditory and working memory skills were assessed by the Face Memory subtest of the NEPSY-II and by a face-up and back-up number-span test (Odedys [47]).

The MDI-C Composite Childhood Depression Scale [48] assessed the emotional state of the child in order to identify a possible depressive state through eight dimensions: self-esteem, anxiety, sad mood, social introversion, pessimism, mistrust, low energy and feelings of helplessness.

#### 2.4.5. Language Skills

The regular, irregular and pseudo-word reading tests from the Odedys DYSlexia Screening Tool [47] assessed reading level and allowed researchers to rule out a diagnosis of dyslexia.

### 2.5. Statistical Analyses

The statistical analyses were carried out using R software (version 3.5.3). The degree of significance retained for all assignments was set at 0.05. Qualitative variables were described by numbers and percentages. A total of 71 binary variables (clinical variables) or tasks were considered. Tasks were scored by the psychomotor therapist as 0 (success) or 1 (failure) based on percentile or standard deviation (below 1 SD or 10th percentile, depending on the test) in accordance with standardized instructions and developmental norms. For assessments of developmental features of handwriting, we used the developmental standards published in a previous study [14]. In order to compare the frequency of failure of clinical variables between the different levels of handwriting disorder (HD not detected by BHK scale, moderate HD, dysgraphia), a Pearson chi-square test was performed. Due to the exploratory nature of our study, Bonferroni corrections were not planned in the statistical analysis (see Bender and Lange [49]).

## 3. Results

### 3.1. Characteristics of the Sample

Twenty-seven children with handwriting disorders were included in this study, four girls (15%) and twenty-three boys (85%), aged 6 years 2 months to 10 years 11 months (mean 8.15 SD 1.51). Eleven of them (41%) presented dysgraphia on the BHK scale and nine (33%) presented more moderate handwriting disorders. In contrast, seven (26%) were not identified by the BHK test as presenting any handwriting disorder (see Table 1). Among the twenty-seven children, six (22%) presented developmental coordination disorder (DCD) according to the DSM-5 criteria and two (7%) had high intellectual potential (≥130 IQ).

### 3.2. Results of the Handwriting Assessment

The detailed results of the sample about the postural and gestural organization and the spatial, temporal and kinematic features of the drawings were described in a previous study [16]. Children with handwriting disorders have poor synergistic coordination of the handwriting arm, characterized by the persistence, whatever the age, of a progression along the line consisting of moving the forearm and the elbow rather than a more mature rotation movement of the forearm at the elbow. They also have an instability of the wrist and a slow and hyper-controlled hand gesture. Moreover, the drawing is characterized by poor quality, lack of fluidity and slowness.

### 3.3. Percentage of Clinical Test Failures (Neuropsychomotor, Psychomotor, Neurovisual, Neuropsychological, Oculomotor and Language Assessments)

Figure 2 presents the percentage of failures in neuropsychomotor and neuropsychological functions assessed by standardized clinical tests and of oculomotricity disorders identified during the examination of oculomotor functions. The variables are ordered by decreasing frequency of failure.

Thus, the descriptive analysis of the results of the clinical tests highlights a set of quite varied disorders such as tone disorder, visual-motor graphic disorder, oculomotor disorder, lack of kinesthetic memory, disturbance of visual-perceptual functions and disorder of executive functions.

### 3.4. Frequency of Failures in Clinical Assessments between the Different Levels of Handwriting Disorder (HD Not Detected by BHK Scale, Moderate HD, Dysgraphia)

A more precise typology of HD was demonstrated by analyzing the distribution of failures in clinical functions according to the degree of handwriting disorder revealed by the BHK test (see Table 2).

The percentage of failures in clinical functions was significantly different (despite a risk of error associated with extended confidence intervals) depending on the level of handwriting disorder undergone by BHK for only four clinical features. Thus, the more pronounced the writing disorder, the more frequent the disorder of visuo-perceptual capacities (χ^2^(2) = 7.51, *p* = 0.016) and the disorder of balance (χ^2^(2) = 8.17, *p* = 0.016) in the population, and more particularly in the “dysgraphia” group. Bodily spatial integration disorder is absent in the “HD not detected by BHK” group and predominant in the “moderate HD” group (χ^2^(2) = 7.88, *p* = 0.019). The tendency of pessimism is strongly present in the “HD not detected by BHK” group and decreases with increasing level of HD (χ^2^(2) = 6.00, *p* = 0.046).

A trend close to statistical significance is observed for eight clinical features (*p* < 0.10) with a higher frequency for six of them in the “dysgraphia” group and for two of them in the “moderate HD” group. Their frequency is never the highest in the “HD not detected by BHK” group. Thus, when the level of HD increases, so do static balance disorders, aiming and catching difficulties, temporal identification disturbance, body diagram difficulties, rhythmic adaptation disorders, and social introversion. These disabilities are particularly common in the “dysgraphia” group. Disorders in visuomotor perceptual capacities and affirmation of manual laterality are respectively the least frequent (14%) and absent in the “HD not detected by BHK” group. They are in the majority in the “moderate HD” group.

The descriptive analysis of whole clinical features reveals a higher proportion of co-occurrences in the “dysgraphia” group than in the “moderate HD” group, whereas the children identified as “HD not detected by BHK” appear less affected.

Thus, when the level of HD increases, there is a greater proportion of neuropsychomotor, neuropsychological and oculomotricity disorders. On the other hand, psycho-affective disorders such as a sad mood, a tendency towards pessimism and low self-esteem appear in the majority of children with an HD not detected by BHK. Psycho-affective disorders are therefore a possible origin of the less pronounced writing disorders.

### 3.5. Failures Greater Than 40% in Each of the Three Groups Identified by the BHK Test (HD Not Detected by BHK, Moderate HD, Dysgraphia)

The analysis of Table 2 shows the following results.

In the “HD not detected by BHK” group, tone disorder (reduction in joint angles measured during passive tone examination), heel–ear angle reduction and graphic visual-spatial coordination disorder appear at a frequency greater than 50%. However, there is no significant difference between the groups for these clinical variables. This is explained by the presence of these difficulties in the clinical group as a whole, irrespective of the level of HD. The disorders of coordination between the upper and lower limbs and of the coordination of static balance, as well as the slowness of reading, appear at a frequency greater than 40% but not significantly because they also appear to be frequent throughout the whole clinical sample. The tendency towards pessimism appears at a frequency greater than 40% in the “HD not detected by BHK” group with a difference between groups close to significance (*p* = 0.07). In addition, a factor analysis revealed a co-occurrence of psycho-affective disorders (depression, lack of self-esteem, sad mood, feeling of helplessness, pessimism, low energy, anxiety) associated with the “HD not detected by BHK” group.

In the “moderate HD” group, tone disorder, heel–ear angle reduction, poorly asserted tonic laterality, poorly asserted manual laterality, visual memory disorder, manual and oculo-manual disorders, graphic visual-spatial coordination disorder, perceptual visual-motor disorder, kinesthetic memory disorder, visual-attentional disorder and bodily spatial integration disorder appear at a frequency greater than 50%. Among these disabilities, only trouble with spatial integration of the body appears to be significantly more frequent in the “moderate HD” group (*p* = 0.03). Once again, this seems to be because the variables appear to be mostly frequent in the whole clinical group. Bimanual coordination and praxia disorders, low energy, oculomotricity disorder and Codes test failure occur at a frequency greater than 40% but not specifically in the group “moderate HD”.

In the “dysgraphia” group, the following appear at a frequency greater than 50%: tone disorder with a heel–ear angle reduction; poorly asserted tonic laterality; disorders in visual memory, manual and oculo-manual skills, visual-motor visual coordination, perceptual visuo-motor capacities (in particular with the WISC-IV Codes test), kinesthetic memory, oculomotricity and especially smooth pursuits; disturbances in visuo-perceptual capacities, executive functions, auditory-attentional capacities, static balance and the capacities to aim and catch; and a disorder of temporal identification. Among these disorders, the static balance disorders identified with the MABC-2 and NP-MOT are significantly more frequent in the “dysgraphia” group (respectively *p* = 0.01 and *p* = 0.049), as well as disorders in aiming and catching abilities (*p* = 0.049) and visual-perceptual skills (*p* = 0.04). Neurological soft signs (synkinesis) and a disorder in unimanual and bimanual coordination appear at a frequency greater than 40% and are in the majority in the “dysgraphia” group, but they are not specific to this group. In addition, a factor analysis revealed a co-occurrence of visual-spatial/constructive and attentional disorders related to an oculomotor disorder (visual fixation) and associated with the “dysgraphia” group.

## 4. Discussion

Our whole sample of 27 children with HD included in the present study underwent a complete developmental battery of neuropsychomotor, neuropsychological and oculomotor assessments. The aim of this transdisciplinary study was to better understand the complexity of the semiology of HD because the literature is poor. The present exploratory study is an important investigation of handwriting disorders, as to our knowledge, no research has explored such a broad set of skills to better understand the etiology of HD. Our results underline high heterogeneity in the children presenting an HD and allow us to identify etiological hypotheses. A previous study highlighted the co-occurrence of difficulties associated with handwriting disorders in children [16]. However, the factor analysis carried out in this same study could only explain 28% of the variance in the sample, which is consistent with the heterogeneity of handwriting disorders highlighted in this article.

In a previous study about the influence of visual control on the quality of the graphic gesture in children with handwriting disorders [19], we hypothesized the involvement of the cortico-striatal and cortico-cerebellar pathways in HD. The hypothesis of cerebellar dysfunction in children with HDs is accepted in the literature [50]. Our clinical sample being very heterogeneous, other etiological hypotheses can be proposed.

Our findings showed an important percentage of children (26%) exhibiting an HD penalizing them at school and being notable in class notebooks but not detected by the BHK test. This lack of detection of HDs is explained by a slight degradation of the handwriting when copying a paragraph (as proposed in the BHK test). Probably, the dual and evaluative situation induced by copying with the BHK test and not by spontaneous handwriting leads to a particular concentration and application of these children, who then obtain a sufficient qualitative handwriting level. Indeed, they fail to achieve proficient handwriting at school, where handwriting times are longer and where the child is constantly double-tasking. Moreover, the mild nature of HD in these children seems to correspond to a relatively low frequency of the associated disorders identified during clinical assessments. This lower frequency would also lead to lighter consequences at the perceptual, perceptual-motor and motor control levels. Thus, HDs in these children appear mainly associated with a tone disorder characterized by a reduction in joint angles, particularly in the heel–ear angle, which may underline an abnormal strengthening of the muscle chain leading to a certain tonicity emphasizing a non-release. This can be the cause of coordination difficulties and poor or limited coordination of the graphomotor gesture. Indeed, in our sample, children with handwriting disorders have poor synergistic coordination of the handwriting arm characterized by the persistence, whatever the age, of a progression along the line consisting of moving the forearm and the elbow rather than a more mature rotation movement of the forearm at the elbow. They also have an instability of the wrist and a slow and hyper-controlled hand gesture. We can therefore assume that these children should benefit from better flexibility thanks to stretching activities and rehabilitation of coordination, and in particular from the prerequisites of the gestural tracing and segmental coordination of the graphomotor gesture by a psychomotor therapist. The slowness of handwriting in 43% of them can highlight possible deficits in phonological or metaphonological processing and in phoneme–grapheme conversion. Interestingly, the right anterior insula is strongly activated when writing letters, possibly related to phoneme–grapheme conversion [51]. It is logical to think that a child with poor recognition and phonological knowledge of a letter will have difficulty integrating the sensorimotor spatial form of the same letter. This hypothesis is confirmed by neuroimaging studies, which show stronger activation of the premotor cortex, parietal cortex, cerebellum and fusiform gyrus when typical children write an unknown letter (pseudoletter) than when they write a known letter. This is visible even though there is no difference in activation among poor writers [52]. These results support the hypothesis that the spatial shape of known letters is difficult to remember for children with HD. Even though we excluded detected speech impairments from our study, it is possible that some of the children had a phonological disturbance that would not have been detected. In the present study, we did not find any children for whom the HD could be explained exclusively by a depressive disorder. However, it is possible that an HD profile without other comorbidities may also reflect different assumptions about a psycho-affective problem. Our findings allow us to assume that depression can be the cause of co-occurring difficulties leading to HD (executive, attentional or memory disorders). This can also be the consequence of HD, which can lead to academic difficulties and remarks from adults even when the child is making significant efforts. It therefore seems that psychological care for these children could be useful in helping them improve their self-esteem and well-being, but it is not sufficient to treat the cause of the HD. It is interesting to note that 43% of the children showing an HD not detected by the BHK test are characterized as low-energy (MDI-C). This may reflect the fatigability related to the cost of handwriting for these children. The high proportion of graphic visual-spatial coordination disabilities (71%) may be related to poor coordination of the graphomotor gesture [15], which would disturb the precision of the strokes, or to visuospatial perceptual difficulties. The mental planning disorder identified using the Laby 5–12 (29% of children with HD not detected by the BHK test) is probably related to difficulties in visual-spatial perception and visual-motor coordination. Indeed, the Laby 5–12 requires significant visual-motor coordination skills, with instances of crossing the walls of the labyrinth with the pen being counted in the scoring. The association, in children with HDs not detected by the BHK, motor coordination and visual-motor graphic coordination disorders and slow reading (specifically when reading pseudo-words, which involve phonological skills) may suggest the involvement of the cerebellum. Indeed, several studies highlight a dysfunction at the level of the cerebellum in the comorbidity between DCD and dyslexia [53,54]. This hypothesis is also corroborated by Nicolson et al. [55], who demonstrate different cerebellar activity in dyslexic adults than in typical adults. In addition, several studies highlight an association between mild disorders of gestural coordination and dyslexia [56,57].

At the same time, an important proportion of children (41%) are classified by the BHK test as having dysgraphia. The more pronounced character of the HD in these children seems to occur with a high frequency of the associated disorders identified during clinical assessments. This higher frequency would also lead to higher consequences at the perceptual, perceptual-motor and motor control levels. Thus, in comparison with children presenting a less pronounced HD, children with dysgraphia would have more disabilities of motor coordination, manual skills and praxis, organization impairments of muscle tone with the presence of neurological soft signs, establishment of laterality, temporal identification, memory functions (kinesthetic and visual), visual-perceptual functions, visual-motor integration, oculomotricity, auditory-attentional capacities and executive functions. We assume that in these children, oculomotor disorders (55%) may be the cause of visual-perceptual disorders and of the static balance disorder noted in 64% of children. Indeed, vision has a proprioceptive function and participates in tonico-postural regulation [58,59]. The impairment of oculomotor functions in our sample of children suggests a possible delay in the maturation of the oculomotor system, which notably involves the cerebello-cortical and cerebellar networks [60,61]. Furthermore, the visual-perceptual difficulties noted corroborate the studies on neuroimaging, which for the most part highlight an involvement of the ventral occipito-temporal cortex in writing [62], a structure involved in visual perception. The difficulties of temporal regulation and rhythmic adaptation that are only noted in children with dysgraphia support the hypothesis of a specific dysfunction of the cortical-subcortical pathways, which involve the cortical structures, the basal ganglia and the thalamus. Indeed, these pathways would be involved in motor adaptation skills and the learning of gestural sequences [63] in the temporal regularity of writing [64]. This again signals the importance of differentiating the diagnosis of dysgraphia from that of a less severe HD because dysgraphia is a neurodevelopmental disorder for which handwriting is really difficult or impossible. Thus, it is not acceptable today to put all HDs under the same umbrella; the remediation should be different according to the HD. The preponderance of oculomotricity disorder supporting the hypothesis of the dysfunction of the cerebellum basal ganglia and superior colliculus structures being involved in oculomotor control [65] and sensorimotor functions (involving the cortico-subcortical pathways) in dysgraphic children highlights the importance of a transdisciplinary assessment of HDs. It is important that children identified as dysgraphic undergo a complete visual and neurovisual assessment, including tests for oculomotricity. The body-image disorder identified more specifically in dysgraphic children is combined with difficulties in integrating an internal representation of body segments in motion. This co-occurrence could be the result of sensory integration difficulties, especially on the proprioceptive level [66,67,68]. Our results attest to a multiplicity of functional impairments in children who are effectively dysgraphic, highlighting the need for a transdisciplinary panel of assessments, both of the graphic gesture and at the neuropsychomotor, neuropsychological and oculomotor levels. Depending on the situation, these children will need rehabilitation in psychomotricity, orthoptics, neuropsychology or occupational therapy to learn to use a computer in the classroom to compensate for his HD. It is also important to identify potential language disorders in these children, who will then need speech therapy.

Finally, 33% of children are classified by the BHK test as having a moderate HD. They have fewer co-occurrences than children with dysgraphia but more difficulties than children with milder HDs. This again highlights the importance of differentiating between different degrees of HDs that do not respond to the same semiologies. The higher frequency of body spatial integration, visuomotor perceptual disorders and poorly asserted manual laterality in these children reinforces the preponderance of the difficulties of sensorimotor integration in HD. Since sensorimotor skills are necessary for the internal representation of action, it is not surprising that impairment of these skills is involved in HD. These results are in line with the empirical data, according to which the graphic space appears as the projection of a representation of the body with an up/down, left/right organization separated by an imaginary vertical median axis reference corresponding to the axis of the body [69]. The multi-modal and redundant integration of sensory information participates in the development of the internal sensorimotor representation of the movement, which itself participates in the function of anticipation and planning of an action [70,71]. These results imply difficulties in anticipation and motor planning in children with HD who fail to correctly parameterize the spatial and temporal characteristics of the graphic trajectory. In addition, the high proportion of visuomotor perceptual disorders is congruent with the literature, which concludes with an implication of visuomotor integration skills in HD [72,73,74,75,76,77]. The lack of kinesthetic memory in 67% of children with moderate HD can lead to poor efficiency of the sensory feedback necessary for the proper anticipation and planning of strokes when writing. The high proportion of these difficulties is congruent with the studies showing an influence of the kinesthetic capacities on the grip of the writing tool [78,79] and on the graphic quality [73,80].

## 5. Conclusions

Thus, our depth clinical examination made it possible to make underlying hypotheses for the involvement of different areas of the brain in HDs. These hypotheses would require further study and brain imaging. Our findings in the present study support the interest in performing a transdisciplinary and standardized clinical examination with developmental standards (neuropsychomotor, neuropsychological and oculomotor) in children with HD. Our results also highlight the multiplicity of HDs and co-occurrences. This heterogeneity of the disorders is congruent with the neuroimaging studies, which underline the involvement of very large cortical areas as well as the parietal, temporal, frontal and occipital areas and the cerebellum [81,82]. HDs can therefore be associated with a multitude of different disorders ranging from poor coordination of the graphomotor gesture to a more general and more complex impairment affecting perceptual-motor, cognitive and/or psycho-affective functions. Our results also highlight the importance of differentiating dysgraphia from more moderate handwriting disorders. Even if the results of the analyses between the different levels of HDs would not remain statistically significant if corrected for multiple comparisons and if the low numerical strength of our population do not allow precise calculation of the percentages of success in each subgroup of handwriting disorder, these are nonetheless interesting frequency results in the context of an exploratory study. It will be important to replicate these results on a larger sample of children with post-hoc tests for greater statistical power, and the goal would be to be able to carry out analyses by school class or with smaller age groups.

## Figures and Tables

**Figure 1 children-10-01512-f001:**
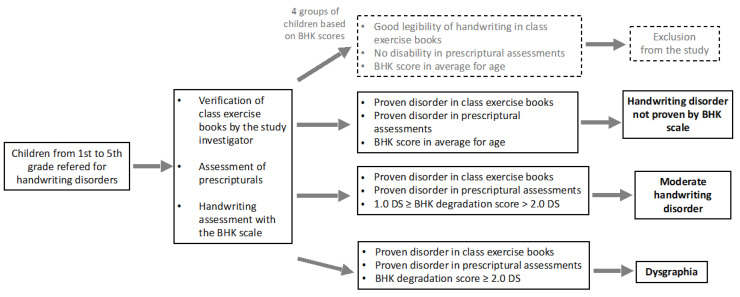
Study design.

**Figure 2 children-10-01512-f002:**
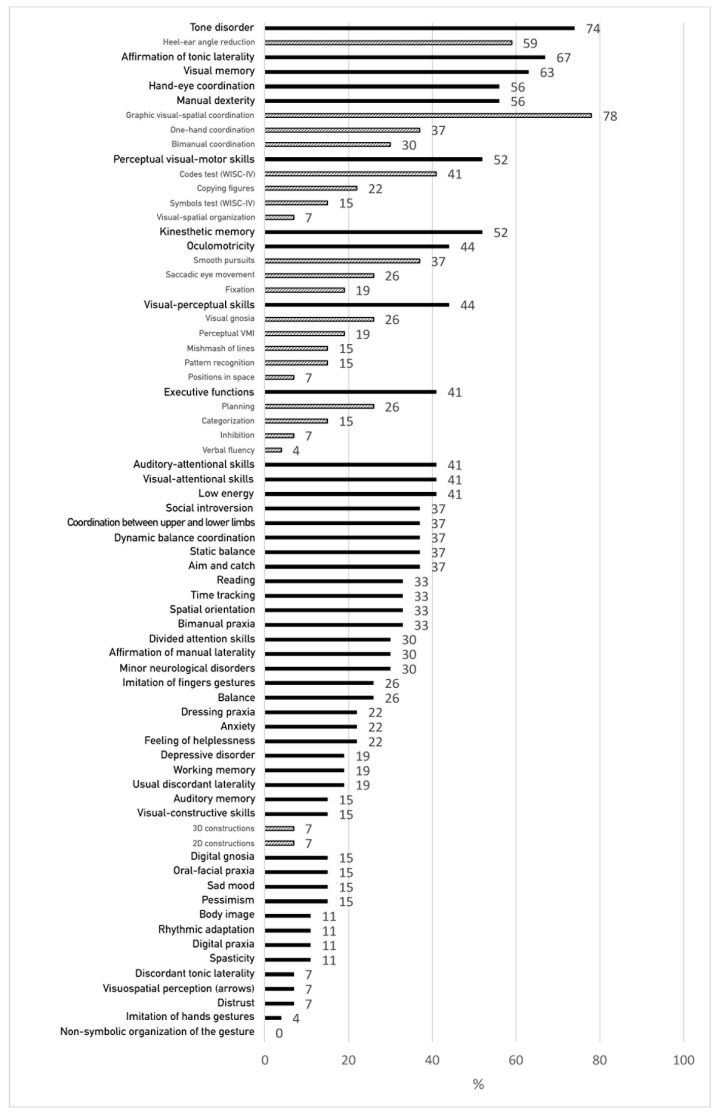
Percentage of failures in clinical functions assessed in the whole group.

**Table 1 children-10-01512-t001:** Characteristics of the children.

	Handwriting Disorder Not Identified by the BHK Test (*n* = 7)	Moderate Handwriting Disorder (*n* = 9)	Dysgraphia (*n* = 11)	Total (*n* = 27)
**Age (years) [m (SD)]**	8.30 (0.81)	7.79 (1.53)	8.36 (1.87)	8.15 (1.51)
**Gender [*n*** **(F/M)]**	0/7	1/8	3/8	4/23

*n*: number; m: mean; SD: standard deviation; F: female; M: male.

**Table 2 children-10-01512-t002:** Percentage of failures in clinical functions assessed for each of the groups classified by the BHK test.

Functions	Whole HD Group (*n* = 27)	HD Not Detected by BHK Scale (*n* = 7)	Moderate HD on BHK (*n* = 9)	Dysgraphia on BHK (*n* = 11)	*p*-Value
TONE DISORDER (NP-MOT)	74	71	78	73	0.95
Heel–ear angle reduction	59	71	56	55	0.76
AFFIRMATION OF TONIC LATERALITY (NP-MOT)	67	43	67	82	0.24
VISUAL MEMORY (REY, NEPSY-II)	63	43	78	64	0.86
HAND-EYE COORDINATION (NP-MOT, MABC-2)	56	43	67	55	0.64
MANUAL DEXTERITY (NP-MOT, MABC-2)	56	43	67	55	0.64
Graphic visual-spatial coordination	78	71	78	82	0.88
One-hand coordination	37	29	33	45	0.75
Bimanual coordination	30	14	44	27	0.43
PERCEPTUAL VISUAL-MOTOR SKILLS	52	14	67	64	0.076
Codes test (WISC-IV)	41	14	44	55	0.24
Copying figures (VMI)	22	0	22	36	0.21
Symbols test (WISC-IV)	15	0	11	27	0.28
Visual-spatial organization (Rey)	7	0	11	9	0.69
KINESTHETIC MEMORY	52	29	67	55	0.32
OCULOMOTRICITY	44	29	44	55	0.26
Smooth pursuits	37	14	33	55	0.16
Saccadic eye movements	26	29	11	36	0.25
Fixation	19	0	33	18	0.22
VISUAL-PERCEPTUAL SKILLS	44	14	33	73	0.016 *
Visual gnosis	26	14	33	27	0.69
Perceptual (VMI)	19	0	22	27	0.34
Mishmash of lines	15	0	11	27	0.28
Pattern recognition	15	0	22	18	0.44
Positions in space (Frostig)	7	0	0	18	0.22
EXECUTIVE FUNCTIONS	41	29	33	55	0.49
Planning	26	29	22	27	0.95
Categorization (Nepsy-II)	15	14	0	27	0.25
Inhibition (Laby 5–12, Stroop)	7	14	11	0	0.48
Verbal fluency (Nepsy-II)	4	0	0	9	0.48
AUDITORY-ATTENTIONAL SKILLS (TEA-CH)	41	29	33	55	0.49
VISUAL-ATTENTIONAL SKILLS (TEA-CH)	41	43	56	27	0.45
LOW ENERGY (MDI-C)	41	43	44	36	0.93
SOCIAL INTROVERSION (MDI-C)	37	14	22	64	0.06
COORDINATION BETWEEN UPPER AND LOWER LIMBS (NP-MOT)	37	43	33	36	0.93
DYNAMIC BALANCE COORDINATION (NP-MOT)	37	43	33	36	0.93
STATIC BALANCE (NP-MOT)	37	29	11	64	0.052
AIM AND CATCH (MABC-2)	37	14	22	64	0.06
READING (ODEDYS)	33	43	33	27	0.80
TIME TRACKING (NP-MOT)	33	0	33	55	0.06
BODILY SPATIAL INTEGRATION (NP-MOT)	33	0	67	27	0.019 *
BIMANUAL PRAXIS(NP-MOT)	33	0	44	45	0.10
DIVIDED ATTENTION SKILLS (TEA-CH)	30	29	22	36	0.79
AFFIRMATION OF MANUAL LATERALITY (NP-MOT)	30	0	56	27	0.053
NEUROLOGICAL SOFT SIGNS (NP-MOT)	30	0	33	45	0.12
IMITATION OF FINGER GESTURES (EMG)	26	14	22	36	0.57
BALANCE (MABC-2)	26	0	11	55	0.016 *
DRESSING PRAXIS	22	14	11	36	0.35
ANXIETY (MDI-C)	22	14	33	18	0.62
FEELING OF HELPLESSNESS (MDI-C)	22	14	11	36	0.35
DEPRESSIVE DISORDER (MDI-C)	19	14	22	18	0.92
WORKING MEMORY (ODEDYS)	19	14	22	18	0.92
USUAL DISCORDANT LATERALITY (NP-MOT)	19	0	33	18	0.25
AUDITORY MEMORY(ODEDYS)	15	0	11	27	0.28
VISUAL-CONSTRUCTIVE SKILLS	15	0	11	27	0.28
3D constructions (Nepsy-II)	7	0	0	18	0.22
2D constructions (Kohs)	7	0	11	9	0.69
DIGITAL GNOSIS (NP-MOT)	15	0	22	18	0.44
ORAL-FACIAL PRAXIS	15	0	11	27	0.29
SAD MOOD(MDI-C)	15	29	22	0	0.20
PESSIMISM (MDI-C)	15	43	11	0	0.046 *
BODY IMAGE(CORP-R)	11	0	0	27	0.09
RHYTHMIC ADAPTATION (NP-MOT)	11	0	0	27	0.09
DIGITAL PRAXIS (NP-MOT)	11	0	11	18	0.50
SPASTICITY (NP-MOT)	11	0	22	9	0.37
DISCORDANT TONIC LATERALITY (NP-MOT)	7	0	11	9	0.69
VISUAL-SPATIAL PERCEPTION (NEPSY-II)	7	14	11	0	0.48
DISTRUST (MDI-C)	7	0	11	9	0.69
IMITATION OF HAND GESTURES (EMG)	4	0	0	9	0.48
SELF-ESTEEM (MDI-C)	4	14	0	0	0.24
NON-SYMBOLIC ORGANIZATION OF THE GESTURE (NP-MOT)	0	0	0	0	na

Levels of signification: * *p* < 0.05.

## Data Availability

The data presented in this study are available on request from the corresponding author.

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
