# Peer review of "Exploratory Investigation of Handwriting Disorders in School-Aged Children from First to Fifth Grade"

_children, 2023, doi:10.3390/children10091512_

Round 1
Reviewer 1 Report (Previous Reviewer 2)
The factor analysis carried out is still not identidied in 2.5. Statistical analyses, and the procedures and statistical results are not presented.
Confidence intervals revelead that significant results found have real error type risk; so it is highly recommended that authors assume it as an internal validity threat to the study.
Author Response
The factor analysis carried out is still not identidied in 2.5. Statistical analyses, and the procedures and statistical results are not presented.
Thank you for this comment.
We removed information about factor analysis at the request of a reviewer. As these data were presented in another article (reference 16), the reviewer's question was: “If it was presented in another article, does it mean that authors have already published this study or part of it elsewhere?”
The study has not been published in another article but the factorial analysis mentioned is actually published in another article (the one to which we refered).
For greater precision, we have removed this analysis from the "Statistical analyses" paragraph. We have also made it clearer in the discussion that these are the results of another study.
The following changes have been made:
" Our whole sample of 27 children with HD included in the present study underwent a complete developmental battery of neuropsychomotor, neuropsychological and oculomotor assessments. The aim of this transdisciplinary study was to better understand the complexity of the semiology of HD because the literature is poor. The present exploratory study is an important state of handwriting disorders, as to our knowledge, no research has explored such a broad set of skills to better understand the etiology of HD. Our results underline a high heterogeneity of the children presenting a HD and allow us to identify etiological hypotheses. A previous study highlighted the co-occurrence of difficulties associated with handwriting disorders in children [16]. However, the factor analysis carried out in this same study could only explain 28% of the variance in the sample, which is consistent with the heterogeneity of handwriting disorders highlighted in this article.”
Confidence intervals revelead that significant results found have real error type risk; so it is highly recommended that authors assume it as an internal validity threat to the study.
Thank you for your comment. We have modified the text as follows:
- in the "Results" section: “The percentage of failures in clinical functions was significantly different (despite a risk of error associated with extended confidence intervals) depending on the level of handwriting disorder undergone by BHK for only 4 clinical features.”
- in the “Conclusion” section: “Our results also highlighted the importance of differentiating dysgraphia from more moderate handwriting disorders. Even if the results of the analyses between the different levels of HDs would not remain statistically significant if corrected for multiple comparisons and if our significant results have error type risk reducing the internal validity of the study, these are nonetheless interesting frequency results in the context of an exploratory study. It would be important to replicate these results on a larger sample of children with post-hoc tests for greater statistical power and the goal would be to be able to carry out analyses school class by school class or by smaller age groups.”

Reviewer 2 Report (New Reviewer)
This is a well written and interesting paper trying to better understand nature and etiology of Handwriting disorders. The results are clear and explained in detail. I have just a few comments:
1. In the abstract section, lines 25-26 (“the mild nature…”), I cannot understand what you meant with that sentence. Could you please rephrase it?
2. Introduction sounds good to me.
3. In the material and methods section, I would explain more clearly the inclusion criteria of the participants (e.g. age range), leaving data such as mean age and SD to the results section.
4. Please explain “WA” (line 110).
The level of English is good. There are only minor errors that can be solved by re-reading the text.
Author Response
This is a well written and interesting paper trying to better understand nature and etiology of Handwriting disorders. The results are clear and explained in detail.
Thank you for your positive feedback.
I have just a few comments:
- In the abstract section, lines 25-26 (“the mild nature...”), I cannot understand what you meant with that sentence. Could you please rephrase it?
Part of the summary seems to have been deleted by mistake. We have modified the text as follows:
“The mild nature of the HDs not detected by BHK appears to occur at a relatively low frequency of the associated disorders identified during clinical evaluations. On the contrary, dysgraphia appears to be associated with a high frequency of co-occurring disorders identified in the clinical assessment, with a predominance of oculomotor disorders (55% of children), leading to visual-perceptual difficulties and a high level of handwriting deterioration. Finally, children with moderate HD have fewer co-occurrences than children with dysgraphia, but have more difficulties than children with milder HD.”
- Introduction sounds good to me.
- In the material and methods section, I would explain more clearly the inclusion criteria of the participants (e.g. age range), leaving data such as mean age and SD to the results section.
To be more specific, we have changed the text as follows:
“Data from a sample of 27 children with handwriting disorders (HDs) aged 6 years 2 months to 10 years 11 months were collected from primary schools (grades 1 to 5) and from the usual out-patient consultation of Pediatrics, at Cochin Port-Royal Hospital, and of Child Psychiatry department, at Necker University Hospital, in Paris, France. We have chosen children in 1st to 5th grade, as this is the elementary school cycle in France. Our sample of children was drawn in such a way as to exclude as much as possible any comorbid disorders that might have an impact on our results, notably oral and written language disorders. 14% of the children were recruited from the Necker-Enfants Malades University Hospital, and 86% from an elementary school, which allowed us to obtain a sample of handwriting disorders representative of those encountered in a population of general school children.”
4.Please explain “WA” (line 110).
We have modified the text as follows:
“(birth < 37 weeks of amenorrhea)”.
The level of English is good. There are only minor errors that can be solved by re-reading the text.
Thanks for your comments, we have corrected the minor errors. We apologize. We have checked the English writing of all the manuscript.

Reviewer 3 Report (New Reviewer)
The article shows a good structure and a detailed analysis of the sample.
In future publications the sample should be larger and as a proposal, perhaps it is better to eliminate those study subjects with clear prior assessments leading to dysgraphia and to provide data from subjects whose dysgraphia is of unknown origin.
Author Response
The article shows a good structure and a detailed analysis of the sample.
Thank you for your positive feedback.
In future publications the sample should be larger and as a proposal, perhaps it is better to eliminate those study subjects with clear prior assessments leading to dysgraphia and to provide data from subjects whose dysgraphia is of unknown origin.
We are aware that the study needs to be replicated with a larger sample. We have made this clear in the study's limitations.
Thanks for your suggestion about exclusion of subjects with clearly identified dysgraphia. However, this seems complex given that handwriting disorders appear to be particularly multifactorial, and part of the goal of our study is to understand what dysgraphia actually is and how it differs from a more moderate handwriting disorder. However, once the semiology of dysgraphia is clearer, it would be interesting to refine the results by excluding children for whom dysgraphia is clearly identified.

Round 2
Reviewer 1 Report (Previous Reviewer 2)
Although text corrections made and the recognition that statistical results constraints to more secure conclusions, authors have not adequatly identified confidence intervals information, more than interval amplitude is the signs difference that alerts to significance frailty, have not identified what kind of error type may occur. We recognize the strong investment made during the study, but is important to "ear" the results and what they are "telling" us.
Author Response
Thank you for this comment.
You're right. We checked with our statistician, who pointed out an error in the formulation of our confidence intervals in this context, which was inconsistent. With his recommendation and to avoid any confusion, we removed the confidence intervals.
We have also modified the following sentence in the section on study limitations:
“Our results also highlighted the importance of differentiating dysgraphia from more moderate handwriting disorders. Even if the results of the analyses between the different levels of HDs would not remain statistically significant if corrected for multiple comparisons and if the low numerical strength of our population do not allow precise calculation of the percentages of success in each sub-group of handwriting disorder, these are nonetheless interesting frequency results in the context of an exploratory study. It would be important to replicate these results on a larger sample of children with post-hoc tests for greater statistical power and the goal would be to be able to carry out analyses school class by school class or by smaller age groups.”

This manuscript is a resubmission of an earlier submission. The following is a list of the peer review reports and author responses from that submission.
Round 1
Reviewer 1 Report
The study describes a interdisciplinary approach to gain insights into the heterogenous expression of handwriting disorders. The authors present a battery of 71 tests related to the assessment of psychomotor, visual and psychological abilities. The results suggest a difference between children with or without handwriting disorders correlated with the level of impairment.
The manuscript contains a lot of information, but lacks coherence and sufficient detail at some points. The findings are very relevant to the audience of the special issue and I would therefore recommend mayor revisions of the manuscript to explain the research to a broad audience.
I present my concerns and comments in order of the paper:
1. Abstract: The sentence in lines 26-27 is confusing.
2. Introduction: The authors note that "the definition of dysgraphia is unclear" (line 57). How would the authors themselves define dysgraphia or handwriting disorder in the study?
Motivation of the study (page 3): The authors list three studies related to brain studies and handwriting disorders and write in line 103f "few studies offer an analysis of the underlying clinical functions involved in handwriting disorders and the literature is poor on this stubject". Actually, research is quite time-consuming and expensive in this area, hence I would not expect a multitude of studies. Still, three is quite a good number. What were other aims or motivations of the study?
3. Participants: The age-range of the children is quite big. These children had between 1 and 4 years of handwriting acquisition already. I am not surprised that they are a heterogenous group. Why did the authors decide on these children specifically? What treatment did the children get (apart from school training)?
line 121: Since the participants were minors, I hope that the parents/legal guardians signed the consent form. Please clarify.
There is no Procedures part in the paper. It would be interesting to know how long the children took for each test and how long the whole data collection took place. Who did the data collection? In what order did testing take place? Over what period of time did children get tested? These parameters are important for the interpretation of results and for replication of the study (or parts of it).
p. 3: It is not possible to read Figure 1.
p. 5 lines 192-198: there seems to be a copy-paste error
p. 6 Statistical analyses: Who or how many persons rated a task as successful or failed? Some of the assessments are testing developmental features of handwriting. How or when do you fail on these tests? Where did the authors set the threshold for failure?
Results: Characteristics of the sample: Please provide the age range and gender for each of the three groups.
Please add H-values to the statistical analyses. Further, if Kruskal-Wallis is significant (p below 0.05) it would require post-hoc testing to figure out where the difference between the groups is exactly. Did the authors do this post-hoc testing?
The authors test for a significant difference with Kruskal-Wallis (one-way ANOVA), but write about correlations in the results. Please be concise in wording here or report correlational results.
What kind of factor analysis (line 332) was performed? This is not mentioned in the statistical analyses.
In line 326-327 the authors interpret the data as "This is because these variables mostly appear to be insuccessful in the whole clinical group." What does this mean? Please explain.
A general point: The authors performed 71 tests in total. About 10-12 showed significant results and distinguished between the three groups with varying handwriting competences or impairments. It is quite overwhelming for a reader to follow the methods part. If the amount of tests and results is reduced, it would be easier to interpret and discuss the data, I suppose. Why report all of these assessments in one paper?
The heterogeneity in the data could not only be due to the different level in handwriting disorders, but could also be related to age or development of handwriting. In my view, the wide age range is a limitation of the study and needs to be addressed in the discussion.
Minor points:
Typo: line 64 "of of"
line 67: "a following study" -- follow-up
Some of the sentences were hard to understand. I mentioned these in my review above.
Author Response
Response to the reviewer 1
" Exploratory investigation of handwriting disorders in school-aged children from 1st to 5th grade"
- Abstract: The sentence in lines 26-27 is confusing.
Thank you for this comment. The sentence is: «The current study offers a transdisciplinary approach with the use of normed and standardized clinical assessments of neuropsychomotor, neuropsychological and oculomotor functions to provide objective data to better understand the nature and etiology of HDs.» We have modified it as follows: «The current study offers a transdisciplinary approach with the use of normed and standardized clinical assessments of neuropsychomotor, neuropsychological and oculomotor functions. The aim is to provide objective data for a better understanding of the nature and the etiology of HDs.»
- Introduction: The authors note that "the definition of dysgraphia is unclear" (line 57). How would the authors themselves define dysgraphia or handwriting disorder in the study?
We used the following classification (as shown in the figure 1):
- Handwriting disorder not identified by the BHK test but identified by their teachers and by an analysis of their notebooks as having writing difficulties: handwriting degradation score lower than + 1.0 SD on the BHK scale and speed score higher than - 1.0 SD ;
- Moderate handwriting disorder: handwriting degradation score between + 1.0 SD and + 2.0 SD on the BHK or speed score less than or equal to - 1.0 SD on the BHK scale;
- Dysgraphia: handwriting degradation score greater than or equal to + 2.0 SD on the BHK scale.
However, the aim of this study is to propose a more precise definition of dysgraphia.
Motivation of the study (page 3): The authors list three studies related to brain studies and handwriting disorders and write in line 103f "few studies offer an analysis of the underlying clinical functions involved in handwriting disorders and the literature is poor on this stubject". Actually, research is quite time-consuming and expensive in this area, hence I would not expect a multitude of studies. Still, three is quite a good number. What were other aims or motivations of the study?
We have modified the sentence as follows: «In addition to these few studies, it is important to enrich the literature concerning the analysis of the underlying clinical functions involved in handwriting disorders. In the present study, we offer a transdisciplinary approach in depth with the use of normed and standardized clinical assessments of neuropsychomotor, neuropsychological and oculomotor functions. We aimed to provide objective data to better understand the nature and etiology of HDs.»
- Participants: The age-range of the children is quite big. These children had between 1 and 4 years of handwriting acquisition already. I am not surprised that they are a heterogenous group. Why did the authors decide on these children specifically? What treatment did the children get (apart from school training)?
We've chosen children in 1st to 5th grade, as this is the elementary school cycle in France. Our sample of children was drawn in such a way as to exclude as far as possible any comorbid disorders that might have an impact on our results, notably oral and written language disorders. As 14% of the children were recruited from the Necker-Enfants Malades University Hospital, and 86% from an elementary school, we obtained a sample of handwriting disorders representative of those encountered in a population of general schoolchildren.
In fact, it would be important to propose a study with a larger sample allowing analysis by school class. However, as you mentioned earlier, research is quite time-consuming and expensive in this area and recruiting children with a handwriting disability and offering them a protocol as comprehensive as that of our study is very time-consuming. As a result, we were unable to increase our sample size for this study.
It would also be interesting to carry out an analysis separating children from 1st to 2th grade and those from 3th to 5th grade. Indeed, the 3th grade or the age of 8 years old is emphasized in several studies of handwriting, indicating more mature patterns of gestural organization and a plateau in trace speed (Vaivre-Douret et al., 2020) and a stabilization of handwriting quality at this developmental period (Overvelde & Hulstijn, 2011) or a stabilization of the evolution of temporal variables (Hamstra-Bletz & Blote, 1990; Mojet, 1991; Thibon et al., 2018).
In order not to influence the results, the children in our sample had not undergone any handwriting retraining at the time of the study. We have included this information in the Participants section.
line 121: Since the participants were minors, I hope that the parents/legal guardians signed the consent form. Please clarify.
Indeed the parents/legal guardians signed the consent form. Our methodology currently reads as follows: «All participants provided written informed consent». We modified it as follows: «All parents/legal guardians of participants provided written informed consent».
There is no Procedures part in the paper. It would be interesting to know how long the children took for each test and how long the whole data collection took place. Who did the data collection? In what order did testing take place? Over what period of time did children get tested? These parameters are important for the interpretation of results and for replication of the study (or parts of it).
Thank you for your questions. We have added some information as following in Material and Methods:
Design and Measures
Handwriting disorders were detected by the teachers and considered to be objective by an analysis of their class exercise books by an experienced psychomotor therapist. In order to assess their handwriting level, each child began to underwent the French adaptation of the standardized assessment of handwriting, the BHK scale [25] adapted from the Concise Evaluation Scale for children’s handwriting [26]. Fig 1 shows the study design previously published with the permission of the editor to reproduce it [15]. Then, a neuropsychomotor assesment (NP-MOT) was administrated and folowed by other tests (psychomotor, neurovisual, neuropsychological) proposed in different orders according to each child's motivation and time constraints. All the assesments were administrated on a single day, with breaks with a total around six hours of testing. The examination of oculomotor functions was recorded about twenty minutes at a second appointment on a different day. The psychomotor therapist investigator in the study administered all the tests.
Moreover, we have added in neurovisual assessment the examination of oculomotor functions which was in separate part Oculomotriciy 2.4.5 (removed)
- 3: It is not possible to read Figure 1.
We have changed the figure 1.
- 5 lines 192-198: there seems to be a copy-paste error
We've removed this part.
- 6 Statistical analyses: Who or how many persons rated a task as successful or failed? Some of the assessments are testing developmental features of handwriting. How or when do you fail on these tests? Where did the authors set the threshold for failure?
The psychomotor therapist investigator in the study rated all tasks as successful or failed. About assessments testing developmental features of handwriting, we have based ourselves on the developmental standards published in the previous article : Vaivre-Douret, L., Lopez, C., Dutruel, A., Vaivre, S. Phenotyping features in the genesis of pre-scriptural gestures in children to assess handwriting developmental levels. Sci Rep. 2021, 11, 731.
Thus, we have added these precisions: The statistical analyses were carried out on R software (version 3.5.3). The degree of significance retained for all assignments was set at 0.05. Qualitative variables are described by numbers and percentages. A total of 71 binary variables (clinical variables) or tasks were considered. Tasks were scored by the psychomotor therapist 0 (success) and 1 (failure) based on percentile or standard deviation (below 1 SD or 10th percentile, depending on the test) in accordance with standardized instructions and developmental norms. About assessments testing developmental features of handwriting, we have used the developmental standard set published in a previous study [14]. In order to compare the frequency of failures in clinical variables between the different levels of handwriting disorder (HD not detected by BHK scale, moderate HD, dysgraphia), a statistical test of Kruskal-Wallis was carried out. In addition, a Bartlett's sphericity test and then a factorial correspondence analysis was carried out to observe whether there are any similarities between the individuals in our sample in terms of their clinical test results.
Results: Characteristics of the sample: Please provide the age range and gender for each of the three groups.
We have added the following table in the article:
Table 1. Characteristics of the children.
|
|
Handwriting disorder not identified by the BHK test (n=7) |
Moderate handwriting disorder (n=9) |
Dysgraphia (n=11) |
Total (n=27) |
|
Age (years) [m (SD)] |
8.30 (0.81) |
7.79 (1.53) |
8.36 (1.87) |
8.15 (1.51) |
|
Gender [n(F/M)] |
0/7 |
1/8 |
3/8 |
4/23 |
n: number; m: mean; SD: standard deviation; F: female; M: male.
Please add H-values to the statistical analyses.
We have added the interval confidence in the document.
Further, if Kruskal-Wallis is significant (p below 0.05) it would require post-hoc testing to figure out where the difference between the groups is exactly. Did the authors do this post-hoc testing?
We have not performed any post-hoc tests.
The authors test for a significant difference with Kruskal- Wallis (one-way ANOVA), but write about correlations in the results. Please be concise in wording here or report correlational results.
We have modified our wording.
What kind of factor analysis (line 332) was performed? This is not mentioned in the statistical analyses.
We performed a Bartlett's sphericity test and then a factorial correspondence analysis. We have added this information in the document.
In line 326-327 the authors interpret the data as "This is because these variables mostly appear to be insuccessful in the whole clinical group." What does this mean? Please explain.
We have modified the text as follows: "This is explained by the presence of these difficulties in the clinical group as a whole, irrespective of the level of HD".
A general point: The authors performed 71 tests in total. About 10-12 showed significant results and distinguished between the three groups with varying handwriting competences or impairments. It is quite overwhelming for a reader to follow the methods part. If the amount of tests and results is reduced, it would be easier to interpret and discuss the data, I suppose. Why report all of these assessments in one paper?
We present all the assessments in a single document, as the aim of our study is to highlight, as exhaustively as possible and from a multidisciplinary perspective, the clinical functions involved in handwriting disorders.
The heterogeneity in the data could not only be due to the different level in handwriting disorders, but could also be related to age or development of handwriting. In my view, the wide age range is a limitation of the study and needs to be addressed in the discussion.
We have added this limitation to the conclusion as follows: "However, it would be interesting to replicate our results with a larger sample of children in order to be able to carry out analyses school class by school class or by smaller age groups".
Minor points:
Typo: line 64 "of of"
line 67: "a following study" -- follow-up
Thank you for this comment. We have corrected the document.
Please find the new revised manuscript :
Article
Exploratory investigation of handwriting disorders in school-aged children from 1st to 5th grade
Abstract: Handwriting disorders (HDs) are prevalent in school-aged children with significant interference with academic performances. The current study offers a transdisciplinary approach with the use of normed and standardized clinical assessments of neuropsychomotor, neuropsychological and oculomotor functions. The aim is to provide objective data for a better understanding of the nature and the etiology of HDs. Data from these clinical assessments were analysed for 27 school-aged children with HD (1st to 5th grade). The results underline a high heterogeneity of the children presenting HDs with many co-occurrences often unknown. However, it was possible to highlight three levels of HDs based on BHK scores: mild HD not detected by the BHK test (26% of children), moderate HD (33%), dysgraphia (41% of children). The mild nature of HDs not detected by BHK seems to co-occurrences than children with dysgraphia but more difficulties than children with milder HDs. This highlights the importance of differentiating between different degrees of HDs that do not respond to the same semiologies. Our findings support the interest of performing a transdisciplinary and standardized clinical examination with developmental standards (neuropsychomotor, neuropsychological and oculomotor) in children with HD. Indeed, HDs can therefore be associated with a multitude of disorders of different nature ranging from poor coordination of the graphomotor gesture to a more general and more complex impairment affecting perceptual-motor, cognitive and/or psycho-affective functions.
Keywords: handwriting disorders; dysgraphia; children; semiology; neuropsychomotor assessment; neuropsychological assessment; oculomotricity
1. Introduction
As children spend 31-60% of their school day writing and performing other fine motor tasks [1], the development of handwriting skills is necessary for academic success [2,3] and the proper development of self-esteem [4,5]. According to the old version of the Diagnostic and Statistical Manual of Mental Disorders (DSM-IV-TR [6]), handwriting disorders (HDs) can be diagnosed in the case of "writing skills significantly lower than expected given chronological age, of measured intelligence and of an appropriate education”. The DSM-5 [7] described HDs as “Impairment in written expression” and dysgraphia is not described in the DSM-5. However, handwriting disorders (HD) affect between 10 and 30 % of school-aged children [8-10]. This is observed both by health professionals in clinical consultations and by teachers who struggle to adapt to the differences in the individual rhythms of these children and their learning difficulties. In this context, current studies on handwriting and its disorders attempt to provide new fundamental knowledge on the different processes involved in the development of handwriting, while clinical and therapeutic aspects remain little explored. Thus, handwriting disorder appears as an umbrella term defining a heterogeneous class of children exhibiting graphic impairments. The study of these disorders is complex as their understanding, both on the semiological level and on the etiological level, is still in the literature only in its early stages, and the definition of dysgraphia is unclear. Since the 1960s, it has been characterized by poor writing quality without any neurological or intellectual disorder being able to explain it [11]. This definition has been clarified by other authors who define dysgraphia as a disorder in written language partly linked to a lack of fine motor control in the execution of motor programs [12,13]. Recently, a relevant study [14] has shown phenotyping features in the genesis of pre-scriptural gestures in children to assess handwriting developmental levels because no recent research has previously think to study the developmental prerequisites of the handwriting organization. The better the quality of the handwriting gesture, the less variation there is in the inter-segmental organization coordinated during the writing task. This makes it possible to assess handwriting development levels in the context of screening for handwriting disorders [14]. Hence, another study was able to demonstrate for the first time the immaturity of the graphomotor gesture in children with a handwriting disorder, characterized both by a lack of synergistic coordination of the different segments of the writing arm and by an impairment of the temporal and kinematic characteristics of prescriptural traces (decrease in fluidity characterized by an increase in the number of strokes and velocity peaks, and an increase in drawing time and in-air pauses) [15,16]. The results about the impairment of the temporal and kinematic characteristics of handwriting are also corroborated by Asselborn et al. [17]). Moreover, generally, the authors highlight a lack in motor programming or in motor execution. Wann [18] suggests a motor programming defect characterized by altered temporal organization of writing (dysfunction, high pause times) due to the child's over-reliance on visual feedback. Lopez & Vaivre-Douret [19] suggest both proprioceptive/kinesthetic feedback deficits and a disruptive effect of visual control on the quality of pre-script drawings in these children, many of whom have kinaesthetic memory and visuo-spatial deficits. Thus, the ability to direct strokes would remain dependent on sensory feedback, itself insufficiently effective, leading to difficulties in achieving proactive control of handwriting. Other authors [20-22] suppose an impairment of the motor execution processes, which is characterized by a spatial, temporal, and kinematic irregularity of the writing characteristic in dysgraphic children. This would be the consequence of excessive neuromotor noises [21,23]. However, only one study has proposed a transdisciplinary investigation of handwriting disorders (HDs) [16]. The results highlighted a typology of three groups of HDs (mild ; mild to moderate; dysgraphia), each being associated with co-occurrences of specific neurodevelopmental dysfunctions: a co-occurrence of psycho-affective disorders that can be considered as a predictor of mild and moderate HDs; a co-occurrence of tone disorders and gross coordination that can be considered as a predictor of mild HDs; a co-occurrence of visual-spatial/constructive and attentional disorders which can be considered as a predictor of the most severe (dysgraphia) and moderate HDs. More specifically, a recent study proposed a transdisciplinary investigation of HDs in a cohort of children with a Developmental Coordination Disorder (DCD) [24]. This highlighted a significant association between neurological soft signs and the presence of dysgraphia in a sample of 65 children with DCD [24]. The dysgraphia appeared to be closely related to several specific dysfunctions of the laterality, to a minor neurological dysfunction of the pyramidal tract manifested by a distal phasic stretch reflex in the lower limb, and to a slowness in digital praxia. In addition to these few studies, it is important to enrich the literature concerning the analysis of the underlying clinical functions involved in handwriting disorders.
In the present study, we offer a transdisciplinary approach in depth with the use of normed and standardized clinical assessments of neuropsychomotor, neuropsychological and oculomotor functions. We aimed to provide objective data to better understand the nature and etiology of HDs.
2. Material and Methods
2.1. Participants
Data from a sample of 27 children with handwriting disorders (HDs) aged 6 years 2 months to 10 years 11 months (mean 8.15 SD 1.51) were collected from primary schools (grades 1 to 5) and in the usual out-patient consultation of Pediatrics, Cochin Port-Royal Hospital, and of Child Psychiatry department, Necker University Hospital, in Paris, France. Children were excluded from the study in case of prematurity (birth <37 WA), sensory, visual, neurological or genetic disorders, dyslexia and severe language disorder, ADHD (according to the DSM-5 criteria [7]), autism spectrum disorder, psychopathology, or motor disorder caused by injury or accident. None of them had repeated or skipped a grade or undergone any handwriting retraining at the time of the study. The institutional research ethics committee of Paris Descartes University approved the study procedures (CER·2018-72) conducted in accordance with the Declaration of Helsinki. All parents/legal guardians of participants provided written informed consent.
2.2. Design and Measures
Handwriting disorders were detected by the teachers and considered to be objective by an analysis of their class exercise books by an experienced psychomotor therapist. In order to assess their handwriting level, each child began to underwent the French adaptation of the standardized assessment of handwriting, the BHK scale [25] adapted from the Concise Evaluation Scale for children’s handwriting [26]. Fig 1 shows the study design previously published with the permission of the editor to reproduce it [15]. Then, a neuropsychomotor assesment (NP-MOT) was administrated and folowed by other tests (psychomotor, neurovisual, neuropsychological) proposed in different orders according to each child's motivation and time constraints. All the assesments were administrated on a single day, with breaks with a total around six hours of testing . The examination of oculomotor functions was recorded about twenty minutes at a second appointment on a different day. The psychomotor therapist investigator in the study administered all the tests.
Figure 1. Study design.
2.3. Handwriting assessment.
In addition to the BHK test, the children performed a previously validated cycloid loop line copying test[14,15]. Data on postural organization and inter-segmental coordination of the writing arm were systematically collected by video recording as described in previous studies [14,15]. Features about the proximal (head, trunk axis, shoulder, elbow, and forearm) and distal (wrist and fingers) gestural organization of the drawing process are collected. Variables relating the material (sheet, drawing line, pen) positioning and observational clinical variables related to the semiology of the motor characteristics of the gesture (control, pressure, synkinesis). In addition, spatio-temporal and kinematic measures were recorded using an Anoto digital pen with Elian Research software (Version 4.2, http://www.seldage.com, accessed on 24 September 2022) for which we have developed specific algorithms to record the measures above.
2.4. Clinical assessments
2.4.1. Neuropsychomotor assessment
All children performed neuropsychomotor physical tasks with the NP-MOT battery [27], including assessment of minor neurological dysfunctions (MND) exploring neurological soft signs like synkinesis (NSS). The age-standardized child assessment using the French NPMOT test battery is applicable to children as young as 4 years old. It has been found to have adequate test-retest reliability and internal consistency. Correlation coefficients of the NP-MOT with the BOTMP [28] range from 0.72 to 0.84, for motor coordination and balance. The NP-MOT battery enables physical assessment of passive/active muscular tone of limbs and axial tone (dangling and extensibility of wrist, shoulder, foot, heel-ear angle, popliteal angles, adductor angles, trunk), highlighting NSS denoting the existence of MND, such as limb pyramidal dysfunction. This is complemented by the assessment of basic motor function, control and regulation in gross motor tasks, gait, balance, coordination, manual dexterity, praxis, gnosopraxis (non-meaningful hand and finger imitation of gestures), digital perception, laterality, bodily spatial integration, rhythmic, and auditory attention tasks. The standardized NP-MOT battery is a developmental assessment because each subtest and milestone is scored from qualitative and quantitative viewpoints according to age, with each score converted to a standard deviation vs. mean, based on normative data for age and applicable to children as young as 4 years old [29]. There is a saturation of the maturation scores between 8 and 10 years, allowing to assess with the NP-MOT older children or adults.
2.4.2. Psychomotor assessment
The MABC-2 children's movement assessment battery (2nd edition) [30], adapted from the American battery [31] was used to assess psychomotor skills. It aims to assess motor impairments and is divided into three categories: manual dexterity (uni-manual, bi-manual test and visual-motor graphic tasks), target and catch (to throw a weighted bag/ball on a target and to catch a weighted bag or a ball), balance (static balance, walking and jumping tasks).
The gnosopraxis imitation of gestures assessment EMG [32,33] was used to assess the distal and digital gnosopraxic efficiency and to measure the child's ideomotor adaptation skills. It consists of performing imitations of arbitrary simple (with the hands) and complex (with the fingers) gestures in the absence of verbal command. This is an adaptation of the Bergès-Lézine assessment [34], paying particular attention to the gesture programming in the notation.
The Body Schema Test – Revised [35] was used to assess the child's representation of his own body and the relationships between different parts of his body. The task consists of a puzzle (non-contiguous pieces) of the body and face from the front (for children aged 3 to 8) and/or from a side view (for children aged 8 and over).
Spatial and temporal identification questions were asked in order to assess the knowledge and mastery of the spatio-temporal vocabulary.
A kinesthetic memorization test consisting of a reproduction test of asymbolic postures which had previously been printed and felt with the eyes closed, has been proposed in order to assess the body's perceptual skills [24].
2.4.3. Neurovisual assessment
Neurovisual aspects including visual gnosias, visual-perceptual, perceptual visual-motor, visuospatial, visuo-constructive skills and oculomotricity were assessed.
Visual perception was assessed using form recognition tasks [36], tangled lines and visual gnosia with outlines of animals, outlines of muddled fruits.
The KABC-II Shape Recognition subtest [37] consists in recognizing and naming drawings of various objects whose image has been altered (some lines of the drawing appear while others have been erased). This item assesses the child's ability to mentally represent the missing parts of the drawing to form a complete mental image, making it possible to name the represented object.
The Developmental Test of Visual-Motor Integration (VMI) (6th ed) [38] assessed pure perceptual abilities (the perceptual subtest of the test consisting of visual recognition of identical insignificant geometric shapes) and visuomotor integration abilities (subtest copy of the test figure consisting of the reproduction of simple and more complex insignificant geometric figures).
The NEPSY-II Arrows subtest [39] consisted of judging the direction, orientation and angles of different lines.
Rey’s complex geometric figure [40] allows the evaluation of aptitudes for perception, structuring and spatial organization (visual-spatial and visuo-constructive praxies). By copying and then reproducing from memory a complex geometric figure, it studies the ability to structure different elements in a graphic space.
The Code and Symbols subtests of the Wechsler intelligence scale for children and adolescents WISC-IV [41] for measuring a mental processing speed index (IVT) in connection with graphomotor capacities consist in analyzing and distinguishing non-significant signs.
The NEPSY-II Cubes subtest [39] and the Kohs block design [42] respectively assessed 3D visual-constructive skills and visual-spatial/constructive skills.
The examination of oculomotor functions was performed using an eye-tracking device made up of two infrared cameras positioned at the level of the inner corner of each eye (Ober consulting eye-Tracker Eyefant® [48]) recording the movements of fixation, smooth visual pursuit, and horizontal and vertical eye saccades. The device records in the horizontal and vertical planes at a sampling rate of 1000 Hz, a spatial resolution of 0.1 ° and a linearity range of +/- 35 ° horizontally and +/- 20 ° vertically.
2.4.4. Neuropsychological assessment
Visual-spatial attention, sustained auditory attention, and divided attention skills were assessed by a crossing test and the Childhood Attention Assessment Test (TEA-Ch [43]).
Executive functions (planning, inhibition skills, mental flexibility, working memory, and verbal fluency) were assessed by the Laby 5-12 labyrinths test [44], the NEPSY-II Categorization and Verbal Fluency subtests [42] and the Stroop's Selective Attention Test [45].
The visual, auditory and working memory skills were assessed by the Face Memory subtest of the NEPSY-II and by a face-up and back-up number span test (Odedys [46]).
The MDI-C Composite Childhood Depression Scale [47] assessed the emotional state of the child in order to identify a possible depressive state, this through 8 dimensions: self-esteem, anxiety, sad mood, social introversion, pessimism, mistrust, low energy and feelings of helplessness.
2.4.5. Language skills
The regular, irregular and pseudo-word reading test from the Odedys DYSlexia Screening Tool [46] assessed reading level and allowed to ruling out a diagnosis of dyslexia.
2.5. Statistical analyses
The statistical analyses were carried out on R software (version 3.5.3). The degree of significance retained for all assignments was set at 0.05. Qualitative variables are described by numbers and percentages. A total of 71 binary variables (clinical variables) or tasks were considered. Tasks were scored by the psychomotor therapist 0 (success) and 1 (failure) based on percentile or standard deviation (below 1 SD or 10th percentile, depending on the test) in accordance with standardized instructions and developmental norms. About assessments testing developmental features of handwriting, we have used the developmental standards published in a previous study [14]. In order to compare the frequency of failures in clinical variables between the different levels of handwriting disorder (HD not detected by BHK scale, moderate HD, dysgraphia), a statistical test of Kruskal-Wallis was carried out. In addition, a Bartlett's sphericity test and then a factorial correspondence analysis was carried out to observe whether there are any similarities between the individuals in our sample in terms of their clinical test results.
3. Results
3.1. Characteristics of the sample
Twenty-seven children with handwriting disorders were included in this study, 4 girls (15%) and 23 boys (85%), aged 6 years 2 months to 10 years 11 months (mean 8.15 SD 1.51). Eleven of them (41%) presented dysgraphia on the BHK scale and nine (33%) presented more moderate handwriting disorders. In contrast, seven (26%) were not identified by the BHK test as presenting any handwriting disorder (see Table 1). Among the 27 children, six (22%) presented developmental coordination disorder (DCD) according to the DSM-5 criteria, and two (7%) had high intellectual potential (>130 IQ).
Table 1. Characteristics of the children.
|
|
Handwriting disorder not identified by the BHK test (n=7) |
Moderate handwriting disorder (n=9) |
Dysgraphia (n=11) |
Total (n=27) |
|
Age (years) [m (SD)] |
8.30 (0.81) |
7.79 (1.53) |
8.36 (1.87) |
8.15 (1.51) |
|
Gender [n(F/M)] |
0/7 |
1/8 |
3/8 |
4/23 |
n: number; m: mean; SD: standard deviation; F: female; M: male.
3.2. Results of the handwriting assessment
The detailed results of the sample about the postural and gestural organization and their spatial, temporal, and kinematic features of the drawings were described in a previous study [16]. Children with handwriting disorders have poor synergistic coordination of the handwriting arm characterised by the persistence, whatever the age, of a progression along the line, consisting in moving the forearm and the elbow rather than a more mature rotation movement of the forearm at the elbow. They have also an instability of the wrist and a slow and hyper-controlled hand gesture. Alongside, the drawing is characterised by poor quality, lack of fluidity, and slowness.
3.3. Percentage of clinical test failures (neuropsychomotor, psychomotor, neurovisual, neuropsychological, oculomotor, and language assessements)
Fig 2 presents the percentage of failures in neuropsychomotor and neuropsychological functions assessed by standardized clinical tests and of oculomotricity disorders identified during the examination of oculomotor functions. The variables are ordered by decreasing frequency of failure.
Figure 2. Percentage of failures in clinical functions assessed in the whole group.
Thus, the descriptive analysis of the results of the clinical tests highlights a set of quite varied disorders such as tone disorder, visual-motor graphic disorder, oculomotor disorder, lack of kinesthetic memory, disturbance of visual perceptual functions, and disorder of executive functions.
3.4. Frequency of failures in clinical assessments between the different levels of handwriting disorder (HD not detected by BHK scale, moderate HD, dysgraphia)
A more precise typology of HD was demonstrated by analyzing the distribution of failures in clinical functions according to the degree of writing disorder revealed by the BHK (see Table 2).
Table 2. Percentage of failures in clinical functions assessed for each of the groups classified by the BHK test.
|
Fonctions |
Whole HD group (n=27) |
HD not detected by BHK scale (n=7) |
Moderate HD on BHK (n=9) |
Dysgraphia on BHK (n=11) |
p-value |
|
Tone disorder (NP-MOT) |
74 |
71 |
78 |
73 |
0.95 |
|
Heel-ear angle reduction |
59 |
71 |
56 |
55 |
0.76 |
|
Affirmation of tonic laterality (NP-MOT) |
67 |
43 |
67 |
82 |
0.24 |
|
Visual memory (Rey, NEPSY-II) |
63 |
43 |
78 |
64 |
0.86 |
|
Hand-eye coordination (NP-MOT, MABC-2) |
56 |
43 |
67 |
55 |
0.64 |
|
Manual dexterity (NP-MOT , MABC-2) |
56 |
43 |
67 |
55 |
0.64 |
|
Graphic visual-spatial coordination |
78 |
71 |
78 |
82 |
0.88 |
|
One-hand coordination |
37 |
29 |
33 |
45 |
0.75 |
|
Bimanual coordination |
30 |
14 |
44 |
27 |
0.43 |
|
Perceptual visual-motor skills |
52 |
14 |
67 |
64 |
0.076 |
|
Codes test (WISC-IV) |
41 |
14 |
44 |
55 |
0.24 |
|
Copying figures (VMI) |
22 |
0 |
22 |
36 |
0.21 |
|
Symbols test (WISC-IV) |
15 |
0 |
11 |
27 |
0.28 |
|
Visual-spatial organization (Rey) |
7 |
0 |
11 |
9 |
0.69 |
|
Kinesthetic memory |
52 |
29 |
67 |
55 |
0.32 |
|
Oculomotricity |
44 |
29 |
44 |
55 |
0.26 |
|
Smooth pursuits |
37 |
14 |
33 |
55 |
0.16 |
|
Saccadic aye movement |
26 |
29 |
11 |
36 |
0.25 |
|
Fixation |
19 |
0 |
33 |
18 |
0.22 |
|
Visual-perceptual skills |
44 |
14 |
33 |
73 |
0.016 * |
|
Visual gnosia |
26 |
14 |
33 |
27 |
0.69 |
|
Perceptual (VMI) |
19 |
0 |
22 |
27 |
0.34 |
|
Mishmash of lines |
15 |
0 |
11 |
27 |
0.28 |
|
Pattern recognition |
15 |
0 |
22 |
18 |
0.44 |
|
Positions in space (Frostig) |
7 |
0 |
0 |
18 |
0.22 |
|
Exécutive functions |
41 |
29 |
33 |
55 |
0.49 |
|
Planning |
26 |
29 |
22 |
27 |
0.95 |
|
Categorization (Nepsy-II) |
15 |
14 |
0 |
27 |
0.25 |
|
Inhibition (Laby 5-12, Stroop) |
7 |
14 |
11 |
0 |
0.48 |
|
Verbal fluency (Nepsy-II) |
4 |
0 |
0 |
9 |
0.48 |
|
Auditory-attentional skills (TEA-Ch) |
41 |
29 |
33 |
55 |
0.49 |
|
Visual-attentional skills (TEA-Ch) |
41 |
43 |
56 |
27 |
0.45 |
|
Low énergy (MDI-C) |
41 |
43 |
44 |
36 |
0.93 |
|
Social introversion (MDI-C) |
37 |
14 |
22 |
64 |
0.06 |
|
Coordination between upper and lower limbs (NP-MOT) |
37 |
43 |
33 |
36 |
0.93 |
|
Dynamic balance coordination (NP-MOT) |
37 |
43 |
33 |
36 |
0.93 |
|
Static balance (NP-MOT) |
37 |
29 |
11 |
64 |
0.052 |
|
Aim and catch (MABC-2) |
37 |
14 |
22 |
64 |
0.06 |
|
Reading (Odedys) |
33 |
43 |
33 |
27 |
0.80 |
|
Time tracking (NP-MOT) |
33 |
0 |
33 |
55 |
0.06 |
|
Bodily spatial integration (NP-MOT) |
33 |
0 |
67 |
27 |
0.019 * |
|
Bimanual praxia (NP-MOT) |
33 |
0 |
44 |
45 |
0.10 |
|
Divided attention skills (TEA-Ch) |
30 |
29 |
22 |
36 |
0.79 |
|
Affirmation of manual laterality (NP-MOT) |
30 |
0 |
56 |
27 |
0.053 |
|
Neurological soft signs (NP-MOT) |
30 |
0 |
33 |
45 |
0.12 |
|
Imitation of fingers gestures (EMG) |
26 |
14 |
22 |
36 |
0.57 |
|
Balance (MABC-2) |
26 |
0 |
11 |
55 |
0.02 * |
|
Dressing praxia |
22 |
14 |
11 |
36 |
0.35 |
|
Anxiety (MDI-C) |
22 |
14 |
33 |
18 |
0.62 |
|
Feeling of helplessness (MDI-C) |
22 |
14 |
11 |
36 |
0.35 |
|
Dépressive disorder (MDI-C) |
19 |
14 |
22 |
18 |
0.92 |
|
Working memory (Odedys) |
19 |
14 |
22 |
18 |
0.92 |
|
usual discordant laterality (NP-MOT) |
19 |
0 |
33 |
18 |
0.25 |
|
Auditory memory (Odedys) |
15 |
0 |
11 |
27 |
0.28 |
|
Visual-constructive skills |
15 |
0 |
11 |
27 |
0.28 |
|
3D constructions (Nepsy-II) |
7 |
0 |
0 |
18 |
0.22 |
|
2D constructions (Kohs) |
7 |
0 |
11 |
9 |
0.69 |
|
Digital gnosia (NP-MOT) |
15 |
0 |
22 |
18 |
0.44 |
|
Oral-facial praxia |
15 |
0 |
11 |
27 |
0.29 |
|
Sad mood (MDI-C) |
15 |
29 |
22 |
0 |
0.20 |
|
Pessimism (MDI-C) |
15 |
43 |
11 |
0 |
0.046 * |
|
Body image (CORP-R) |
11 |
0 |
0 |
27 |
0.09 |
|
Rhythmic adaptation (NP-MOT) |
11 |
0 |
0 |
27 |
0.09 |
|
Digital praxia (NP-MOT) |
11 |
0 |
11 |
18 |
0.50 |
|
Spasticity (NP-MOT) |
11 |
0 |
22 |
9 |
0.37 |
|
Discordant tonic latérality (NP-MOT) |
7 |
0 |
11 |
9 |
0.69 |
|
Visual-spatial perception (Nepsy-II) |
7 |
14 |
11 |
0 |
0.48 |
|
Distrust (MDI-C) |
7 |
0 |
11 |
9 |
0.69 |
|
Imitation of hands gestures (EMG) |
4 |
0 |
0 |
9 |
0.48 |
|
Self estim (MDI-C) |
4 |
14 |
0 |
0 |
0.24 |
|
Non symbolic organisation of the gesture (NP-MOT) |
0 |
0 |
0 |
0 |
na |
The percentage of failures in clinical functions was significantly different depending on the level of handwriting disorder underwent by BHK for only 4 clinical features. the. Thus, the more pronounced the writing disorder, the more frequent the disorder of visuo-perceptual capacities (CI 95% 26%-64%;p = 0.016) and the disorder of balance (CI 95% 12%-47%;p = 0.02) in the population, and more particularly in the "dysgraphia" group. Bodily spatial integration disorder is absent in the "HD not detected by BHK" group and predominant in the "moderate HD" group (CI 95% 17%-54%;p = 0.019). The tendency to pessimism is strongly present in the "HD not detected by BHK" group and decreases with increasing level of HD (CI 95% 5%-35%;p = 0.046).
A trend close to statistical significance is observed for 8 clinical features (p <0.10) with a higher frequency of 6 of them in the "dysgraphia" group and of 2 of them in the "moderate HD" group. Their frequency is never the highest in the "HD not detected by BHK" group. Thus, when the level of HD increases on the one hand, so do static balance disorders, aiming and catching difficulties, temporal identification, body diagram, rhythmic adaptation disorders, and social introversion. These disabilities are particularly common in the "dysgraphia" group. Disorders of visuomotor perceptual capacities and affirmation of manual laterality are respectively the least frequent (14%) and absent in the "HD not detected by BHK" group. They are in the majority in the "moderate HD" group.
The descriptive analysis of the whole clinical features reveals a higher proportion of co-occurrences in the "dysgraphia" group than in the "moderate HD" group, whereas the children identified as having a " HD not detected by BHK "appear less affected”.
Thus, when the level of HD increases, there is a greater proportion of neuro-psychomotor, neuropsychological and oculomotricity disorders. On the other hand, psycho-affective disorders such as a sad mood, a tendency to pessimism, and low self-esteem appear in the majority of children with a HD not detected by BHK. Psycho-affective disorders are therefore a possible origin of the less pronounced writing disorders.
3.5. Failures greater than 40% in each of the three groups identified by the BHK test (HD not detected by BHK, moderate HD, dysgraphia)
The analysis of Table 2 shows the following results.
In the "HD not detected by BHK" group, the tone disorder (reduction in joint angles measured during passive tone examination), the heel-ear angle reduction, and the graphic visual-spatial coordination disorder appear at a frequency greater than 50%. However, there is no significant difference between groups for these clinical variables. This is explained by the presence of these difficulties in the clinical group as a whole, irrespective of the level of HD. The disorders of the coordination between the upper and lower limbs, of the coordination of static balance and a slowness of reading appear at a frequency greater than 40% but not significantly. Because they also appear to be highly unsuccessful throughout the whole clinical sample. The tendency to pessimism appears at a frequency greater than 40% in the "HD not detected by BHK" group with a difference between groups close to significance (p=0.07). In addition, a factor analysis revealed a co-occurrence of psycho-affective disorders (depression, lack of self-esteem, sad mood, feeling of helplessness, pessimism, low energy, anxiety) associated with the "HD not detected by BHK" group.
In the "moderate HD" group, tone disorder, heel-ear angle reduction, poorly asserted tonic laterality, poorly asserted manual laterality, visual memory disorder, manual and oculo-manual disorders, graphic visual-spatial coordination disorder, perceptual visual-motor disorder, kinesthetic memory disorder, visual-attentional disorder, and bodily spatial integration disorder appear at a frequency greater than 50%. Among these disabilities, only a trouble of spatial integration of the body appears to be significantly more frequent in the "moderate HD" group (p=0.03). Once again this seems to be because the variables appear to be mostly unsuccessful in the whole clinical group. Bimanual coordination and praxia disorders, low energy, oculomotricity disorder and Codes test failure occur at a frequency greater than 40% but not specifically in the group "moderate HD".
In the "Dysgraphia" group, appear at a frequency greater than 50% : a tone disorder with a heel-ear angle reduction, a poorly asserted tonic laterality, disorders of visual memory, manual and oculo-manual skills, visual-motor visual coordination, perceptual visuo-motor capacities (in particular with the test of the WISC-IV Codes), kinesthetic memory, oculomotricity and especially in smooth pursuits, disturbances of visuo-perceptual capacities, of executive functions, of auditory-attentional capacities, of static balance and of the capacities to aim and catch, and a disorder of temporal identification. Among these disorders, the static balance disorders identified with MABC-2 and NP-MOT is significantly more frequent in the "dysgraphia" group (respectively p=0.01 and p=0.049), as well as the disorder of aiming and catching abilities (p=0.049) and visual-perceptual skills disorder (p=0.04). Neurological soft signs (synkinesis), a disorder of uni-manual and bimanual coordinations appear at a frequency greater than 40% and are in the majority in the "Dysgraphia" group, but they are not specific to this group. In addition, a factor analysis revealed a co-occurrence of visual-spatial/constructive and attentional disorders related to an oculomotor disorder (visual fixation) and associated with the "dysgraphia" group.
4. Discussion
Our whole sample of 27 children with HD included in the present study underwent a complete developmental battery of neuropsychomotor, neuropsychological and oculomotor assessments. The aim of this transdisciplinary study was to better understand the complexity of the semiology of HD because the literature is poor. The present study is an important state of handwriting disorders, as to our knowledge, no research has explored such a broad set of skills to better understand the etiology of HD. Our results underline a high heterogeneity of the children presenting a HD with many co-occurrences often unknown. This is notably highlighted by a factorial analysis which can only explain 28% of the sample variance [16]. The study of the percentage of failures in the whole clinical functions assessed allows us to identify major clinical profiles and etiological hypotheses.
In a previous study about the influence of visual control on the quality of the graphic gesture in children with handwriting disorders [19], we hypothesized the involvement of the cortico-striatal and cortico-cerebellar pathways in HD. The hypothesis of cerebellar dysfunction in children with HDs is accepted in the literature [49]. Our clinical sample being very heterogeneous, other etiological hypotheses can be proposed.
Our findings showed an important percentage of children (26%) exhibiting a HD penalizing them at school and being notable in class notebooks but not detected by the BHK test. This lack of detection of HDs is explained by a slight degradation of the handwriting when copying a paragraph (like proposed in the BHK test). Probably, the dual and evaluative situation induced by a copy with the BHK and not by a spontaneous handwriting leads to a particular concentration and application of these children, who then obtain a sufficient qualitative handwriting. Indeed, they fail to achieve proficient handwriting at school, where handwriting times are longer and where the child is constantly in double-tasking. Moreover, the mild nature of HD in these children seems to occur to a relatively low frequency of the associated disorders identified during clinical assessments. This lower frequency would also lead to lighter consequences at the perceptual, perceptual-motor and motor control levels. Thus, HDs in these children appears mainly associated with a tone disorder characterized by a reduction in joint angles, particularly in the heel-ear angle which may underline an abnormal strengthening of the muscle chain leading to a certain tonicity emphasizing a non-release. This can be the cause of coordination difficulties and poor or limited coordination of the graphomotor gesture. Indeed, in our sample, children with handwriting disorders have poor synergistic coordination of the handwriting arm characterized by the persistence, whatever the age, of a progression along the line, consisting in moving the forearm and the elbow rather than a more mature rotation movement of the forearm at the elbow. They have also an instability of the wrist and a slow and hyper-controlled hand gesture. We can therefore assume that these children should benefit from better flexibility thanks to stretching activities and rehabilitation of coordination, and in particular about the prerequisites of the gestural tracing and segmental coordination of the graphomotor gesture by a psychomotor therapist. The slowness of handwriting in 43% of them can highlight possible deficits in phonological or metaphonological processing and in phoneme-grapheme conversion. Interestingly, the right anterior insula is strongly activated when writing letters, possibly related to phoneme-grapheme conversion [50]. It is logical to think that a child with poor recognition and phonological knowledge of the letter will have difficulty integrating the sensorimotor spatial form of the same letter. This hypothesis is confirmed by neuroimaging studies which show stronger activation of the premotor cortex, parietal cortex, cerebellum, and fusiform gyrus when typical children write an unknown letter (pseudoletter) than when they write a known letter. This is visible even though there is no difference in activation among poor writers [51]. These results support the hypothesis that the spatial shape of known letters is difficult to remember for children with HD. Even though we excluded detected speech impairments from our study, it is possible that some of the children had a phonological disturbance that would not have been detected. In the present study, we did not find any children for whom the HD could be explained exclusively by a depressive disorder. However, it is possible that a HD profile without other comorbidities may also reflect different assumptions about a psycho-affective problem. Our findings allow to assume that depression can be the cause of co-occurring difficulties leading to HD (executive, attentional or memory disorders). This can also be the consequence of HD which can lead to academic difficulties and remarks from adults even when the child is making significant efforts. It therefore seems that psychological care for these children could be useful in helping them improve their self-esteem and well-being, but it is not sufficient to treat the cause of the HD. It is interesting to note that 43% of the children showing a HD not detected by BHK are characterized by low energy (MDI-C). This may reflect the fatigability related to the cost of handwriting for these children. The high proportion of graphic visual-spatial coordination disabilities (71%) may be related to poor coordination of the graphomotor gesture [15], which would disturb the precision of the strokes, or to visuospatial perceptual difficulties. The mental planning disorder identified at Laby 5-12 (29% of children with HD not detected by BHK) is probably related to difficulties in visual-spatial perception and visual-motor coordination. Indeed, the Laby 5-12 requires significant visual-motor coordination skills, the fact of crossing the walls of the labyrinth with the pen being counted in the scoring. The association, in children with HDs not detected by BHK, between motor coordination and visual-motor graphic coordination disorders and slow reading (specifically when reading pseudo-words, which involve phonological skills) may suggest the involvement of the cerebellum. Indeed, several studies highlight a dysfunction at the level of the cerebellum in the comorbidity between DCD and dyslexia [52,53]. This hypothesis is also corroborated by Nicolson et al. [54] who demonstrate different cerebellar activity in dyslexic adults compared to typical adults. In addition, several studies highlight an association between mild disorders of gestural coordination and dyslexia [55,56].
At the same time, a significant proportion of children (41%) are classified by BHK test as having dysgraphia. The more pronounced character of the HD in these children seems to occur to a high frequency of the associated disorders identified during clinical assessments. This higher frequency would also lead to higher consequences at the perceptual, perceptual-motor, and motor control levels. Thus, children with dysgraphia have disabilities in motor coordination, manual skills and praxis, organization impairments of muscle tone with the presence of neurological soft signs, establishment of laterality, temporal identification, memory functions (kinesthetic and visual), visual perceptual functions, visual-motor integration, oculomotricity, auditory-attentional capacities and executive functions. They differ significantly (p <0.05) or almost significantly (p <0.10) from children presenting a less pronounced HD. We assume in these children that the oculomotor disorder (55%) may be the cause of visual-perceptual disorders and of the static balance disorder noted in 64% of children. Indeed, vision has a proprioceptive function and participates in tonico-postural regulation [57,58]. The impairment of oculomotor functions in our sample of children suggests a possible delay in the maturation of the oculomotor system, which notably involves the cerebello-cortical and cerebellar networks [59,60]. Furthermore, the visual-perceptual difficulties noted corroborate the studies in neuroimaging, which for the most part highlight an involvement of the ventral occipito-temporal cortex in writing [61], a structure involved in visual perception. The difficulties of temporal regulation and rhythmic adaptation that are only noted in children with dysgraphia support the hypothesis of a specific dysfunction of the cortical-subcortical pathways, which involve the cortical structures, the basal ganglia, and the thalamus. Indeed, these pathways would be involved in the motor adaptation skills and the learning of gestural sequences [62] in the temporal regularity of writing [63]. This again signals the importance of differentiating the diagnosis of dysgraphia from that of a less severe HD because dysgraphia is a neurodevelopmental disorder for which handwriting is really difficult or impossible. Thus, it is not acceptable today to put all HDs under the same umbrella, the remediation should be different according to the HD. The preponderance of oculomotricity disorder supporting the hypothesis of dysfunction of the cerebellum basal ganglia and superior colliculus structures being involved in oculomotor control [64] and sensorimotor functions (involving the cortico-subcortical pathways) in dysgraphic children highlights the importance of a transdisciplinary assessment of HDs. It is important that children identified as dysgraphic could undergo a complete visual and neurovisual assessment including oculomotricity. The body image disorder identified more specifically in dysgraphic children is combined with difficulties in integrating an internal representation of body segments in motion. This co-occurrence could be the result of sensory integration difficulties, especially on the proprioceptive level [65-67]. Our results attest to a multiplicity of functional impairments in children who are effectively dysgraphic, highlighting the need for a transdisciplinary panel of assessments, both of the graphic gesture, and at the neuropsychomotor, neuropsychological and oculomotor level. Depending on the situation, these children will need rehabilitation in psychomotricity, orthoptics, neuropsychology, or an occupational therapy to learn to use a computer in the classroom to compensate for his HD. It is also important to identify potential language disorders in these children, who will then need speech therapy.
Finally, 33% of children are classified by BHK test as having a moderate HD. They have fewer co-occurrences than children with dysgraphia but more difficulties than children with milder HDs. This again highlights the importance of differentiating between different degrees of HDs that do not respond to the same semiologies. The significantly higher frequency (p<0.05) of body spatial integration, visuomotor perceptual disorders, and of poorly asserted manual laterality in these children, reinforces the preponderance of the difficulties of sensorimotor integration in HD. Since sensorimotor skills are necessary for the internal representation of action, it is not surprising that impairment of these skills is involved in HD. These results are in line with the empirical data according to which the graphic space appears as the projection of a representation of the body with an up/down, left/right organization separated by an imaginary vertical median axis reference corresponding to the axis of the body [68]. The multi-modal and redundant integration of sensory information participates in the development of the internal sensorimotor representation of movement, which itself participates in the function of anticipation and planning of the action [69,70]. These results imply difficulties in anticipation and motor planning in children with HD who fail to correctly parameterize the spatial and temporal characteristics of the graphic trajectory. In addition, the high proportion of visuomotor perceptual disorders is congruent with the literature which concludes to an implication of visuomotor integration skills in HD [71-76]. The lack of kinesthetic memory in 67% of children with a moderate HD can lead to poor efficiency of the sensory feedbacksnecessary for the proper anticipation and planning of strokes when writing. The high proportion of these difficulties is congruent with the studies showing an influence of the kinesthetic capacities on the grip of the writing tool [77-78] and on the graphic quality [72,79].
5. Conclusions
Thus, our depth clinical examination made it possible to make underlying hypotheses for the involvement of different areas of the brain in HDs. These hypotheses would require further study in brain imaging. Our findings in the present study support the interest of performing a transdisciplinary and standardized clinical examination with developmental standards (neuropsychomotor, neuropsychological and oculomotor) in children with HD. Our results also highlighted the multiplicity of HDs and co-occurrences. This heterogeneity of the disorders is congruent with the neuroimaging studies, which underline the involvement of very large cortical areas as well as the parietal, temporal, frontal, occipital areas, and cerebellum [80,81]. HDs can therefore be associated with a multitude of different disorders ranging from a poor coordination of the graphomotor gesture to a more general and more complex impairment affecting perceptual-motor, cognitive and/or psycho-affective functions. However, it would be interesting to replicate our results with a larger sample of children in order to be able to carry out analyses school class by school class or by smaller age groups.

Reviewer 2 Report
Because Kruskal-Wallis test is traditionally classified has a non parametric comparison test for ordinal measures, please explain with more accuracy how and why it was used to find associtations.
3.4. Statistical associations between HD level underwent by BHK test and the results of clinical 285 assessments
We do not understand how you used Kruskal-Wallis test. If you have compared the four groups, the probability presented recquires Bonferroni correction; and based on it, no statistical significance was obtained. Additionally, you do not know among what groups this difference actually may have occurred, because no paired Mann-Whitney tests were made. If you really want to verify if association exist we recommend you Spearman correlation test or, if necessary contingency coefficient (for nominal data). In fact, sometimes you refer differences and not associations (e.g., 449-450)
lines 309-12- rewrite paragraph, to be more understandable what you want to express.
lines 313-319- your study design does not allow to take such conclusion, even if you consider that there is an association, results can also be considered the other way: HD as the origin of psycho-affective problems; or even equivalence among variables (e.g., lines 415-422).
lines 332-335- no factor analysis is mentioned "Statistical analyses"and no statistical results are presented
Author Response
Because Kruskal-Wallis test is traditionally classified has a non parametric comparison test for ordinal measures, please explain with more accuracy how and why it was used to find associations.
The wording was wrong. The test was used to compare different samples (HD not identified by the BHK test, moderate HD, dysgraphia) and to determine whether there was a difference between these samples for each of the clinical variables. We modified the titles and the Statistical analysis section as follows: "In order to compare the frequency of failures in clinical variables between the different levels of handwriting disorder (HD not detected by BHK scale, moderate HD, dysgraphia), a statistical test of Kruskal-Wallis was carried out."
3.4. Statistical associations between HD level underwent by BHK test and the results of clinical assessments
3.4. Frequency of failures in clinical assessments between the different levels of handwriting disorder (HD not detected by BHK scale, moderate HD, dysgraphia)
We do not understand how you used Kruskal-Wallis test. If you have compared the four groups, the probability presented recquires Bonferroni correction; and based on it, no statistical significance was obtained. Additionally, you do not know among what groups this difference actually may have occurred, because no paired Mann-Whitney tests were made. If you really want to verify if association exist we recommend you Spearman correlation test or, if necessary contingency coefficient (for nominal data). In fact, sometimes you refer differences and not associations (e.g., 449-450)
We compared the frequency of failures in clinical variables between the three groups (HD not detected by BHK scale, moderate HD, dysgraphia). We have modified the wording in the text to be more precise. Thanks
lines 309-12- rewrite paragraph, to be more understandable what you want to express.
OK
lines 313-319- your study design does not allow to take such conclusion, even if you consider that there is an association, results can also be considered the other way: HD as the origin of psycho-affective problems; or even equivalence among variables (e.g., lines 415-422).
We removed psychological weakness in the sentence
we did not find any children for whom the HD could be explained exclusively by a depressive disorder or a psychological weakness.
We completed by : In the present study, we did not find any children for whom the HD could be explained exclusively by a depressive disorder. However, it is possible that a HD profile without other comorbidities may also reflect different assumptions about a psycho-affective problem. Our findings allow to assume that depression can be the cause of co-occurring difficulties leading to HD (executive, attentional or memory disorders). This can also be the consequence of HD which can lead to academic difficulties and remarks from adults even when the child is making significant efforts.
lines 332-335- no factor analysis is mentioned "Statistical analyses"and no statistical results are presented.
We performed a Bartlett's sphericity test and then a factorial correspondence analysis. We have added this information in the "Statistical analysis". The results are not presented in this document so as not to weigh down the results, but they are presented in the referenced article (reference 17).
There is no Procedures part in the paper. It would be interesting to know how long the children took for each test and how long the whole data collection took place. Who did the data collection? In what order did testing take place? Over what period of time did children get tested? These parameters are important for the interpretation of results and for replication of the study (or parts of it).
Thank you for your questions. We have added some information as following in Material and Methods:
Design and Measures
Handwriting disorders were detected by the teachers and considered to be objective by an analysis of their class exercise books by an experienced psychomotor therapist. In order to assess their handwriting level, each child began to underwent the French adaptation of the standardized assessment of handwriting, the BHK scale [25] adapted from the Concise Evaluation Scale for children’s handwriting [26]. Fig 1 shows the study design previously published with the permission of the editor to reproduce it [15]. Then, a neuropsychomotor assesment (NP-MOT) was administrated and folowed by other tests (psychomotor, neurovisual, neuropsychological) proposed in different orders according to each child's motivation and time constraints. All the assesments were administrated on a single day, with breaks with a total around six hours of testing. The examination of oculomotor functions was recorded about twenty minutes at a second appointment on a different day. The psychomotor therapist investigator in the study administered all the tests.
Moreover, we have added in neurovisual assessment the examination of oculomotor functions which was in separate part Oculomotriciy 2.4.5 (removed)
Further, if Kruskal-Wallis is significant (p below 0.05) it would require post-hoc testing to figure out where the difference between the groups is exactly. Did the authors do this post-hoc testing?
We have not performed any post-hoc tests.
The authors test for a significant difference with Kruskal- Wallis (one-way ANOVA), but write about correlations in the results. Please be concise in wording here or report correlational results.
We have modified our wording.
What kind of factor analysis (line 332) was performed? This is not mentioned in the statistical analyses.
We performed a Bartlett's sphericity test and then a factorial correspondence analysis. We have added this information in the document.
In line 326-327 the authors interpret the data as "This is because these variables mostly appear to be insuccessful in the whole clinical group." What does this mean? Please explain.
We have modified the text as follows: "This is explained by the presence of these difficulties in the clinical group as a whole, irrespective of the level of HD".
A general point: The authors performed 71 tests in total. About 10-12 showed significant results and distinguished between the three groups with varying handwriting competences or impairments. It is quite overwhelming for a reader to follow the methods part. If the amount of tests and results is reduced, it would be easier to interpret and discuss the data, I suppose. Why report all of these assessments in one paper?
Thanks; We presented all the assessments in a single document, as the aim of our study is to highlight, as exhaustively as possible and from a multidisciplinary perspective, the clinical functions involved in handwriting disorders.
The heterogeneity in the data could not only be due to the different level in handwriting disorders, but could also be related to age or development of handwriting. In my view, the wide age range is a limitation of the study and needs to be addressed in the discussion.
We have added this limitation to the conclusion as follows: "However, it would be interesting to replicate our results with a larger sample of children in order to be able to carry out analyses school class by school class or by smaller age groups".
Minor points:
Typo: line 64 "of of"
line 67: "a following study" -- follow-up
Thank you for this comment. We have corrected the document.
Please find the new revised manuscript :
Article
Exploratory investigation of handwriting disorders in school-aged children from 1st to 5th grade
Abstract: Handwriting disorders (HDs) are prevalent in school-aged children with significant interference with academic performances. The current study offers a transdisciplinary approach with the use of normed and standardized clinical assessments of neuropsychomotor, neuropsychological and oculomotor functions. The aim is to provide objective data for a better understanding of the nature and the etiology of HDs. Data from these clinical assessments were analysed for 27 school-aged children with HD (1st to 5th grade). The results underline a high heterogeneity of the children presenting HDs with many co-occurrences often unknown. However, it was possible to highlight three levels of HDs based on BHK scores: mild HD not detected by the BHK test (26% of children), moderate HD (33%), dysgraphia (41% of children). The mild nature of HDs not detected by BHK seems to co-occurrences than children with dysgraphia but more difficulties than children with milder HDs. This highlights the importance of differentiating between different degrees of HDs that do not respond to the same semiologies. Our findings support the interest of performing a transdisciplinary and standardized clinical examination with developmental standards (neuropsychomotor, neuropsychological and oculomotor) in children with HD. Indeed, HDs can therefore be associated with a multitude of disorders of different nature ranging from poor coordination of the graphomotor gesture to a more general and more complex impairment affecting perceptual-motor, cognitive and/or psycho-affective functions.
Keywords: handwriting disorders; dysgraphia; children; semiology; neuropsychomotor assessment; neuropsychological assessment; oculomotricity
1. Introduction
As children spend 31-60% of their school day writing and performing other fine motor tasks [1], the development of handwriting skills is necessary for academic success [2,3] and the proper development of self-esteem [4,5]. According to the old version of the Diagnostic and Statistical Manual of Mental Disorders (DSM-IV-TR [6]), handwriting disorders (HDs) can be diagnosed in the case of "writing skills significantly lower than expected given chronological age, of measured intelligence and of an appropriate education”. The DSM-5 [7] described HDs as “Impairment in written expression” and dysgraphia is not described in the DSM-5. However, handwriting disorders (HD) affect between 10 and 30 % of school-aged children [8-10]. This is observed both by health professionals in clinical consultations and by teachers who struggle to adapt to the differences in the individual rhythms of these children and their learning difficulties. In this context, current studies on handwriting and its disorders attempt to provide new fundamental knowledge on the different processes involved in the development of handwriting, while clinical and therapeutic aspects remain little explored. Thus, handwriting disorder appears as an umbrella term defining a heterogeneous class of children exhibiting graphic impairments. The study of these disorders is complex as their understanding, both on the semiological level and on the etiological level, is still in the literature only in its early stages, and the definition of dysgraphia is unclear. Since the 1960s, it has been characterized by poor writing quality without any neurological or intellectual disorder being able to explain it [11]. This definition has been clarified by other authors who define dysgraphia as a disorder in written language partly linked to a lack of fine motor control in the execution of motor programs [12,13]. Recently, a relevant study [14] has shown phenotyping features in the genesis of pre-scriptural gestures in children to assess handwriting developmental levels because no recent research has previously think to study the developmental prerequisites of the handwriting organization. The better the quality of the handwriting gesture, the less variation there is in the inter-segmental organization coordinated during the writing task. This makes it possible to assess handwriting development levels in the context of screening for handwriting disorders [14]. Hence, another study was able to demonstrate for the first time the immaturity of the graphomotor gesture in children with a handwriting disorder, characterized both by a lack of synergistic coordination of the different segments of the writing arm and by an impairment of the temporal and kinematic characteristics of prescriptural traces (decrease in fluidity characterized by an increase in the number of strokes and velocity peaks, and an increase in drawing time and in-air pauses) [15,16]. The results about the impairment of the temporal and kinematic characteristics of handwriting are also corroborated by Asselborn et al. [17]). Moreover, generally, the authors highlight a lack in motor programming or in motor execution. Wann [18] suggests a motor programming defect characterized by altered temporal organization of writing (dysfunction, high pause times) due to the child's over-reliance on visual feedback. Lopez & Vaivre-Douret [19] suggest both proprioceptive/kinesthetic feedback deficits and a disruptive effect of visual control on the quality of pre-script drawings in these children, many of whom have kinaesthetic memory and visuo-spatial deficits. Thus, the ability to direct strokes would remain dependent on sensory feedback, itself insufficiently effective, leading to difficulties in achieving proactive control of handwriting. Other authors [20-22] suppose an impairment of the motor execution processes, which is characterized by a spatial, temporal, and kinematic irregularity of the writing characteristic in dysgraphic children. This would be the consequence of excessive neuromotor noises [21,23]. However, only one study has proposed a transdisciplinary investigation of handwriting disorders (HDs) [16]. The results highlighted a typology of three groups of HDs (mild ; mild to moderate; dysgraphia), each being associated with co-occurrences of specific neurodevelopmental dysfunctions: a co-occurrence of psycho-affective disorders that can be considered as a predictor of mild and moderate HDs; a co-occurrence of tone disorders and gross coordination that can be considered as a predictor of mild HDs; a co-occurrence of visual-spatial/constructive and attentional disorders which can be considered as a predictor of the most severe (dysgraphia) and moderate HDs. More specifically, a recent study proposed a transdisciplinary investigation of HDs in a cohort of children with a Developmental Coordination Disorder (DCD) [24]. This highlighted a significant association between neurological soft signs and the presence of dysgraphia in a sample of 65 children with DCD [24]. The dysgraphia appeared to be closely related to several specific dysfunctions of the laterality, to a minor neurological dysfunction of the pyramidal tract manifested by a distal phasic stretch reflex in the lower limb, and to a slowness in digital praxia. In addition to these few studies, it is important to enrich the literature concerning the analysis of the underlying clinical functions involved in handwriting disorders.
In the present study, we offer a transdisciplinary approach in depth with the use of normed and standardized clinical assessments of neuropsychomotor, neuropsychological and oculomotor functions. We aimed to provide objective data to better understand the nature and etiology of HDs.
2. Material and Methods
2.1. Participants
Data from a sample of 27 children with handwriting disorders (HDs) aged 6 years 2 months to 10 years 11 months (mean 8.15 SD 1.51) were collected from primary schools (grades 1 to 5) and in the usual out-patient consultation of Pediatrics, Cochin Port-Royal Hospital, and of Child Psychiatry department, Necker University Hospital, in Paris, France. Children were excluded from the study in case of prematurity (birth <37 WA), sensory, visual, neurological or genetic disorders, dyslexia and severe language disorder, ADHD (according to the DSM-5 criteria [7]), autism spectrum disorder, psychopathology, or motor disorder caused by injury or accident. None of them had repeated or skipped a grade or undergone any handwriting retraining at the time of the study. The institutional research ethics committee of Paris Descartes University approved the study procedures (CER·2018-72) conducted in accordance with the Declaration of Helsinki. All parents/legal guardians of participants provided written informed consent.
2.2. Design and Measures
Handwriting disorders were detected by the teachers and considered to be objective by an analysis of their class exercise books by an experienced psychomotor therapist. In order to assess their handwriting level, each child began to underwent the French adaptation of the standardized assessment of handwriting, the BHK scale [25] adapted from the Concise Evaluation Scale for children’s handwriting [26]. Fig 1 shows the study design previously published with the permission of the editor to reproduce it [15]. Then, a neuropsychomotor assesment (NP-MOT) was administrated and folowed by other tests (psychomotor, neurovisual, neuropsychological) proposed in different orders according to each child's motivation and time constraints. All the assesments were administrated on a single day, with breaks with a total around six hours of testing . The examination of oculomotor functions was recorded about twenty minutes at a second appointment on a different day. The psychomotor therapist investigator in the study administered all the tests.
Figure 1. Study design.
2.3. Handwriting assessment.
In addition to the BHK test, the children performed a previously validated cycloid loop line copying test[14,15]. Data on postural organization and inter-segmental coordination of the writing arm were systematically collected by video recording as described in previous studies [14,15]. Features about the proximal (head, trunk axis, shoulder, elbow, and forearm) and distal (wrist and fingers) gestural organization of the drawing process are collected. Variables relating the material (sheet, drawing line, pen) positioning and observational clinical variables related to the semiology of the motor characteristics of the gesture (control, pressure, synkinesis). In addition, spatio-temporal and kinematic measures were recorded using an Anoto digital pen with Elian Research software (Version 4.2, http://www.seldage.com, accessed on 24 September 2022) for which we have developed specific algorithms to record the measures above.
2.4. Clinical assessments
2.4.1. Neuropsychomotor assessment
All children performed neuropsychomotor physical tasks with the NP-MOT battery [27], including assessment of minor neurological dysfunctions (MND) exploring neurological soft signs like synkinesis (NSS). The age-standardized child assessment using the French NPMOT test battery is applicable to children as young as 4 years old. It has been found to have adequate test-retest reliability and internal consistency. Correlation coefficients of the NP-MOT with the BOTMP [28] range from 0.72 to 0.84, for motor coordination and balance. The NP-MOT battery enables physical assessment of passive/active muscular tone of limbs and axial tone (dangling and extensibility of wrist, shoulder, foot, heel-ear angle, popliteal angles, adductor angles, trunk), highlighting NSS denoting the existence of MND, such as limb pyramidal dysfunction. This is complemented by the assessment of basic motor function, control and regulation in gross motor tasks, gait, balance, coordination, manual dexterity, praxis, gnosopraxis (non-meaningful hand and finger imitation of gestures), digital perception, laterality, bodily spatial integration, rhythmic, and auditory attention tasks. The standardized NP-MOT battery is a developmental assessment because each subtest and milestone is scored from qualitative and quantitative viewpoints according to age, with each score converted to a standard deviation vs. mean, based on normative data for age and applicable to children as young as 4 years old [29]. There is a saturation of the maturation scores between 8 and 10 years, allowing to assess with the NP-MOT older children or adults.
2.4.2. Psychomotor assessment
The MABC-2 children's movement assessment battery (2nd edition) [30], adapted from the American battery [31] was used to assess psychomotor skills. It aims to assess motor impairments and is divided into three categories: manual dexterity (uni-manual, bi-manual test and visual-motor graphic tasks), target and catch (to throw a weighted bag/ball on a target and to catch a weighted bag or a ball), balance (static balance, walking and jumping tasks).
The gnosopraxis imitation of gestures assessment EMG [32,33] was used to assess the distal and digital gnosopraxic efficiency and to measure the child's ideomotor adaptation skills. It consists of performing imitations of arbitrary simple (with the hands) and complex (with the fingers) gestures in the absence of verbal command. This is an adaptation of the Bergès-Lézine assessment [34], paying particular attention to the gesture programming in the notation.
The Body Schema Test – Revised [35] was used to assess the child's representation of his own body and the relationships between different parts of his body. The task consists of a puzzle (non-contiguous pieces) of the body and face from the front (for children aged 3 to 8) and/or from a side view (for children aged 8 and over).
Spatial and temporal identification questions were asked in order to assess the knowledge and mastery of the spatio-temporal vocabulary.
A kinesthetic memorization test consisting of a reproduction test of asymbolic postures which had previously been printed and felt with the eyes closed, has been proposed in order to assess the body's perceptual skills [24].
2.4.3. Neurovisual assessment
Neurovisual aspects including visual gnosias, visual-perceptual, perceptual visual-motor, visuospatial, visuo-constructive skills and oculomotricity were assessed.
Visual perception was assessed using form recognition tasks [36], tangled lines and visual gnosia with outlines of animals, outlines of muddled fruits.
The KABC-II Shape Recognition subtest [37] consists in recognizing and naming drawings of various objects whose image has been altered (some lines of the drawing appear while others have been erased). This item assesses the child's ability to mentally represent the missing parts of the drawing to form a complete mental image, making it possible to name the represented object.
The Developmental Test of Visual-Motor Integration (VMI) (6th ed) [38] assessed pure perceptual abilities (the perceptual subtest of the test consisting of visual recognition of identical insignificant geometric shapes) and visuomotor integration abilities (subtest copy of the test figure consisting of the reproduction of simple and more complex insignificant geometric figures).
The NEPSY-II Arrows subtest [39] consisted of judging the direction, orientation and angles of different lines.
Rey’s complex geometric figure [40] allows the evaluation of aptitudes for perception, structuring and spatial organization (visual-spatial and visuo-constructive praxies). By copying and then reproducing from memory a complex geometric figure, it studies the ability to structure different elements in a graphic space.
The Code and Symbols subtests of the Wechsler intelligence scale for children and adolescents WISC-IV [41] for measuring a mental processing speed index (IVT) in connection with graphomotor capacities consist in analyzing and distinguishing non-significant signs.
The NEPSY-II Cubes subtest [39] and the Kohs block design [42] respectively assessed 3D visual-constructive skills and visual-spatial/constructive skills.
The examination of oculomotor functions was performed using an eye-tracking device made up of two infrared cameras positioned at the level of the inner corner of each eye (Ober consulting eye-Tracker Eyefant® [48]) recording the movements of fixation, smooth visual pursuit, and horizontal and vertical eye saccades. The device records in the horizontal and vertical planes at a sampling rate of 1000 Hz, a spatial resolution of 0.1 ° and a linearity range of +/- 35 ° horizontally and +/- 20 ° vertically.
2.4.4. Neuropsychological assessment
Visual-spatial attention, sustained auditory attention, and divided attention skills were assessed by a crossing test and the Childhood Attention Assessment Test (TEA-Ch [43]).
Executive functions (planning, inhibition skills, mental flexibility, working memory, and verbal fluency) were assessed by the Laby 5-12 labyrinths test [44], the NEPSY-II Categorization and Verbal Fluency subtests [42] and the Stroop's Selective Attention Test [45].
The visual, auditory and working memory skills were assessed by the Face Memory subtest of the NEPSY-II and by a face-up and back-up number span test (Odedys [46]).
The MDI-C Composite Childhood Depression Scale [47] assessed the emotional state of the child in order to identify a possible depressive state, this through 8 dimensions: self-esteem, anxiety, sad mood, social introversion, pessimism, mistrust, low energy and feelings of helplessness.
2.4.5. Language skills
The regular, irregular and pseudo-word reading test from the Odedys DYSlexia Screening Tool [46] assessed reading level and allowed to ruling out a diagnosis of dyslexia.
2.5. Statistical analyses
The statistical analyses were carried out on R software (version 3.5.3). The degree of significance retained for all assignments was set at 0.05. Qualitative variables are described by numbers and percentages. A total of 71 binary variables (clinical variables) or tasks were considered. Tasks were scored by the psychomotor therapist 0 (success) and 1 (failure) based on percentile or standard deviation (below 1 SD or 10th percentile, depending on the test) in accordance with standardized instructions and developmental norms. About assessments testing developmental features of handwriting, we have used the developmental standards published in a previous study [14]. In order to compare the frequency of failures in clinical variables between the different levels of handwriting disorder (HD not detected by BHK scale, moderate HD, dysgraphia), a statistical test of Kruskal-Wallis was carried out. In addition, a Bartlett's sphericity test and then a factorial correspondence analysis was carried out to observe whether there are any similarities between the individuals in our sample in terms of their clinical test results.
3. Results
3.1. Characteristics of the sample
Twenty-seven children with handwriting disorders were included in this study, 4 girls (15%) and 23 boys (85%), aged 6 years 2 months to 10 years 11 months (mean 8.15 SD 1.51). Eleven of them (41%) presented dysgraphia on the BHK scale and nine (33%) presented more moderate handwriting disorders. In contrast, seven (26%) were not identified by the BHK test as presenting any handwriting disorder (see Table 1). Among the 27 children, six (22%) presented developmental coordination disorder (DCD) according to the DSM-5 criteria, and two (7%) had high intellectual potential (>130 IQ).
Table 1. Characteristics of the children.
|
|
Handwriting disorder not identified by the BHK test (n=7) |
Moderate handwriting disorder (n=9) |
Dysgraphia (n=11) |
Total (n=27) |
|
Age (years) [m (SD)] |
8.30 (0.81) |
7.79 (1.53) |
8.36 (1.87) |
8.15 (1.51) |
|
Gender [n(F/M)] |
0/7 |
1/8 |
3/8 |
4/23 |
n: number; m: mean; SD: standard deviation; F: female; M: male.
3.2. Results of the handwriting assessment
The detailed results of the sample about the postural and gestural organization and their spatial, temporal, and kinematic features of the drawings were described in a previous study [16]. Children with handwriting disorders have poor synergistic coordination of the handwriting arm characterised by the persistence, whatever the age, of a progression along the line, consisting in moving the forearm and the elbow rather than a more mature rotation movement of the forearm at the elbow. They have also an instability of the wrist and a slow and hyper-controlled hand gesture. Alongside, the drawing is characterised by poor quality, lack of fluidity, and slowness.
3.3. Percentage of clinical test failures (neuropsychomotor, psychomotor, neurovisual, neuropsychological, oculomotor, and language assessements)
Fig 2 presents the percentage of failures in neuropsychomotor and neuropsychological functions assessed by standardized clinical tests and of oculomotricity disorders identified during the examination of oculomotor functions. The variables are ordered by decreasing frequency of failure.
Figure 2. Percentage of failures in clinical functions assessed in the whole group.
Thus, the descriptive analysis of the results of the clinical tests highlights a set of quite varied disorders such as tone disorder, visual-motor graphic disorder, oculomotor disorder, lack of kinesthetic memory, disturbance of visual perceptual functions, and disorder of executive functions.
3.4. Frequency of failures in clinical assessments between the different levels of handwriting disorder (HD not detected by BHK scale, moderate HD, dysgraphia)
A more precise typology of HD was demonstrated by analyzing the distribution of failures in clinical functions according to the degree of writing disorder revealed by the BHK (see Table 2).
Table 2. Percentage of failures in clinical functions assessed for each of the groups classified by the BHK test.
|
Fonctions |
Whole HD group (n=27) |
HD not detected by BHK scale (n=7) |
Moderate HD on BHK (n=9) |
Dysgraphia on BHK (n=11) |
p-value |
|
Tone disorder (NP-MOT) |
74 |
71 |
78 |
73 |
0.95 |
|
Heel-ear angle reduction |
59 |
71 |
56 |
55 |
0.76 |
|
Affirmation of tonic laterality (NP-MOT) |
67 |
43 |
67 |
82 |
0.24 |
|
Visual memory (Rey, NEPSY-II) |
63 |
43 |
78 |
64 |
0.86 |
|
Hand-eye coordination (NP-MOT, MABC-2) |
56 |
43 |
67 |
55 |
0.64 |
|
Manual dexterity (NP-MOT , MABC-2) |
56 |
43 |
67 |
55 |
0.64 |
|
Graphic visual-spatial coordination |
78 |
71 |
78 |
82 |
0.88 |
|
One-hand coordination |
37 |
29 |
33 |
45 |
0.75 |
|
Bimanual coordination |
30 |
14 |
44 |
27 |
0.43 |
|
Perceptual visual-motor skills |
52 |
14 |
67 |
64 |
0.076 |
|
Codes test (WISC-IV) |
41 |
14 |
44 |
55 |
0.24 |
|
Copying figures (VMI) |
22 |
0 |
22 |
36 |
0.21 |
|
Symbols test (WISC-IV) |
15 |
0 |
11 |
27 |
0.28 |
|
Visual-spatial organization (Rey) |
7 |
0 |
11 |
9 |
0.69 |
|
Kinesthetic memory |
52 |
29 |
67 |
55 |
0.32 |
|
Oculomotricity |
44 |
29 |
44 |
55 |
0.26 |
|
Smooth pursuits |
37 |
14 |
33 |
55 |
0.16 |
|
Saccadic aye movement |
26 |
29 |
11 |
36 |
0.25 |
|
Fixation |
19 |
0 |
33 |
18 |
0.22 |
|
Visual-perceptual skills |
44 |
14 |
33 |
73 |
0.016 * |
|
Visual gnosia |
26 |
14 |
33 |
27 |
0.69 |
|
Perceptual (VMI) |
19 |
0 |
22 |
27 |
0.34 |
|
Mishmash of lines |
15 |
0 |
11 |
27 |
0.28 |
|
Pattern recognition |
15 |
0 |
22 |
18 |
0.44 |
|
Positions in space (Frostig) |
7 |
0 |
0 |
18 |
0.22 |
|
Exécutive functions |
41 |
29 |
33 |
55 |
0.49 |
|
Planning |
26 |
29 |
22 |
27 |
0.95 |
|
Categorization (Nepsy-II) |
15 |
14 |
0 |
27 |
0.25 |
|
Inhibition (Laby 5-12, Stroop) |
7 |
14 |
11 |
0 |
0.48 |
|
Verbal fluency (Nepsy-II) |
4 |
0 |
0 |
9 |
0.48 |
|
Auditory-attentional skills (TEA-Ch) |
41 |
29 |
33 |
55 |
0.49 |
|
Visual-attentional skills (TEA-Ch) |
41 |
43 |
56 |
27 |
0.45 |
|
Low énergy (MDI-C) |
41 |
43 |
44 |
36 |
0.93 |
|
Social introversion (MDI-C) |
37 |
14 |
22 |
64 |
0.06 |
|
Coordination between upper and lower limbs (NP-MOT) |
37 |
43 |
33 |
36 |
0.93 |
|
Dynamic balance coordination (NP-MOT) |
37 |
43 |
33 |
36 |
0.93 |
|
Static balance (NP-MOT) |
37 |
29 |
11 |
64 |
0.052 |
|
Aim and catch (MABC-2) |
37 |
14 |
22 |
64 |
0.06 |
|
Reading (Odedys) |
33 |
43 |
33 |
27 |
0.80 |
|
Time tracking (NP-MOT) |
33 |
0 |
33 |
55 |
0.06 |
|
Bodily spatial integration (NP-MOT) |
33 |
0 |
67 |
27 |
0.019 * |
|
Bimanual praxia (NP-MOT) |
33 |
0 |
44 |
45 |
0.10 |
|
Divided attention skills (TEA-Ch) |
30 |
29 |
22 |
36 |
0.79 |
|
Affirmation of manual laterality (NP-MOT) |
30 |
0 |
56 |
27 |
0.053 |
|
Neurological soft signs (NP-MOT) |
30 |
0 |
33 |
45 |
0.12 |
|
Imitation of fingers gestures (EMG) |
26 |
14 |
22 |
36 |
0.57 |
|
Balance (MABC-2) |
26 |
0 |
11 |
55 |
0.02 * |
|
Dressing praxia |
22 |
14 |
11 |
36 |
0.35 |
|
Anxiety (MDI-C) |
22 |
14 |
33 |
18 |
0.62 |
|
Feeling of helplessness (MDI-C) |
22 |
14 |
11 |
36 |
0.35 |
|
Dépressive disorder (MDI-C) |
19 |
14 |
22 |
18 |
0.92 |
|
Working memory (Odedys) |
19 |
14 |
22 |
18 |
0.92 |
|
usual discordant laterality (NP-MOT) |
19 |
0 |
33 |
18 |
0.25 |
|
Auditory memory (Odedys) |
15 |
0 |
11 |
27 |
0.28 |
|
Visual-constructive skills |
15 |
0 |
11 |
27 |
0.28 |
|
3D constructions (Nepsy-II) |
7 |
0 |
0 |
18 |
0.22 |
|
2D constructions (Kohs) |
7 |
0 |
11 |
9 |
0.69 |
|
Digital gnosia (NP-MOT) |
15 |
0 |
22 |
18 |
0.44 |
|
Oral-facial praxia |
15 |
0 |
11 |
27 |
0.29 |
|
Sad mood (MDI-C) |
15 |
29 |
22 |
0 |
0.20 |
|
Pessimism (MDI-C) |
15 |
43 |
11 |
0 |
0.046 * |
|
Body image (CORP-R) |
11 |
0 |
0 |
27 |
0.09 |
|
Rhythmic adaptation (NP-MOT) |
11 |
0 |
0 |
27 |
0.09 |
|
Digital praxia (NP-MOT) |
11 |
0 |
11 |
18 |
0.50 |
|
Spasticity (NP-MOT) |
11 |
0 |
22 |
9 |
0.37 |
|
Discordant tonic latérality (NP-MOT) |
7 |
0 |
11 |
9 |
0.69 |
|
Visual-spatial perception (Nepsy-II) |
7 |
14 |
11 |
0 |
0.48 |
|
Distrust (MDI-C) |
7 |
0 |
11 |
9 |
0.69 |
|
Imitation of hands gestures (EMG) |
4 |
0 |
0 |
9 |
0.48 |
|
Self estim (MDI-C) |
4 |
14 |
0 |
0 |
0.24 |
|
Non symbolic organisation of the gesture (NP-MOT) |
0 |
0 |
0 |
0 |
na |
The percentage of failures in clinical functions was significantly different depending on the level of handwriting disorder underwent by BHK for only 4 clinical features. the. Thus, the more pronounced the writing disorder, the more frequent the disorder of visuo-perceptual capacities (CI 95% 26%-64%;p = 0.016) and the disorder of balance (CI 95% 12%-47%;p = 0.02) in the population, and more particularly in the "dysgraphia" group. Bodily spatial integration disorder is absent in the "HD not detected by BHK" group and predominant in the "moderate HD" group (CI 95% 17%-54%;p = 0.019). The tendency to pessimism is strongly present in the "HD not detected by BHK" group and decreases with increasing level of HD (CI 95% 5%-35%;p = 0.046).
A trend close to statistical significance is observed for 8 clinical features (p <0.10) with a higher frequency of 6 of them in the "dysgraphia" group and of 2 of them in the "moderate HD" group. Their frequency is never the highest in the "HD not detected by BHK" group. Thus, when the level of HD increases on the one hand, so do static balance disorders, aiming and catching difficulties, temporal identification, body diagram, rhythmic adaptation disorders, and social introversion. These disabilities are particularly common in the "dysgraphia" group. Disorders of visuomotor perceptual capacities and affirmation of manual laterality are respectively the least frequent (14%) and absent in the "HD not detected by BHK" group. They are in the majority in the "moderate HD" group.
The descriptive analysis of the whole clinical features reveals a higher proportion of co-occurrences in the "dysgraphia" group than in the "moderate HD" group, whereas the children identified as having a " HD not detected by BHK "appear less affected”.
Thus, when the level of HD increases, there is a greater proportion of neuro-psychomotor, neuropsychological and oculomotricity disorders. On the other hand, psycho-affective disorders such as a sad mood, a tendency to pessimism, and low self-esteem appear in the majority of children with a HD not detected by BHK. Psycho-affective disorders are therefore a possible origin of the less pronounced writing disorders.
3.5. Failures greater than 40% in each of the three groups identified by the BHK test (HD not detected by BHK, moderate HD, dysgraphia)
The analysis of Table 2 shows the following results.
In the "HD not detected by BHK" group, the tone disorder (reduction in joint angles measured during passive tone examination), the heel-ear angle reduction, and the graphic visual-spatial coordination disorder appear at a frequency greater than 50%. However, there is no significant difference between groups for these clinical variables. This is explained by the presence of these difficulties in the clinical group as a whole, irrespective of the level of HD. The disorders of the coordination between the upper and lower limbs, of the coordination of static balance and a slowness of reading appear at a frequency greater than 40% but not significantly. Because they also appear to be highly unsuccessful throughout the whole clinical sample. The tendency to pessimism appears at a frequency greater than 40% in the "HD not detected by BHK" group with a difference between groups close to significance (p=0.07). In addition, a factor analysis revealed a co-occurrence of psycho-affective disorders (depression, lack of self-esteem, sad mood, feeling of helplessness, pessimism, low energy, anxiety) associated with the "HD not detected by BHK" group.
In the "moderate HD" group, tone disorder, heel-ear angle reduction, poorly asserted tonic laterality, poorly asserted manual laterality, visual memory disorder, manual and oculo-manual disorders, graphic visual-spatial coordination disorder, perceptual visual-motor disorder, kinesthetic memory disorder, visual-attentional disorder, and bodily spatial integration disorder appear at a frequency greater than 50%. Among these disabilities, only a trouble of spatial integration of the body appears to be significantly more frequent in the "moderate HD" group (p=0.03). Once again this seems to be because the variables appear to be mostly unsuccessful in the whole clinical group. Bimanual coordination and praxia disorders, low energy, oculomotricity disorder and Codes test failure occur at a frequency greater than 40% but not specifically in the group "moderate HD".
In the "Dysgraphia" group, appear at a frequency greater than 50% : a tone disorder with a heel-ear angle reduction, a poorly asserted tonic laterality, disorders of visual memory, manual and oculo-manual skills, visual-motor visual coordination, perceptual visuo-motor capacities (in particular with the test of the WISC-IV Codes), kinesthetic memory, oculomotricity and especially in smooth pursuits, disturbances of visuo-perceptual capacities, of executive functions, of auditory-attentional capacities, of static balance and of the capacities to aim and catch, and a disorder of temporal identification. Among these disorders, the static balance disorders identified with MABC-2 and NP-MOT is significantly more frequent in the "dysgraphia" group (respectively p=0.01 and p=0.049), as well as the disorder of aiming and catching abilities (p=0.049) and visual-perceptual skills disorder (p=0.04). Neurological soft signs (synkinesis), a disorder of uni-manual and bimanual coordinations appear at a frequency greater than 40% and are in the majority in the "Dysgraphia" group, but they are not specific to this group. In addition, a factor analysis revealed a co-occurrence of visual-spatial/constructive and attentional disorders related to an oculomotor disorder (visual fixation) and associated with the "dysgraphia" group.
4. Discussion
Our whole sample of 27 children with HD included in the present study underwent a complete developmental battery of neuropsychomotor, neuropsychological and oculomotor assessments. The aim of this transdisciplinary study was to better understand the complexity of the semiology of HD because the literature is poor. The present study is an important state of handwriting disorders, as to our knowledge, no research has explored such a broad set of skills to better understand the etiology of HD. Our results underline a high heterogeneity of the children presenting a HD with many co-occurrences often unknown. This is notably highlighted by a factorial analysis which can only explain 28% of the sample variance [16]. The study of the percentage of failures in the whole clinical functions assessed allows us to identify major clinical profiles and etiological hypotheses.
In a previous study about the influence of visual control on the quality of the graphic gesture in children with handwriting disorders [19], we hypothesized the involvement of the cortico-striatal and cortico-cerebellar pathways in HD. The hypothesis of cerebellar dysfunction in children with HDs is accepted in the literature [49]. Our clinical sample being very heterogeneous, other etiological hypotheses can be proposed.
Our findings showed an important percentage of children (26%) exhibiting a HD penalizing them at school and being notable in class notebooks but not detected by the BHK test. This lack of detection of HDs is explained by a slight degradation of the handwriting when copying a paragraph (like proposed in the BHK test). Probably, the dual and evaluative situation induced by a copy with the BHK and not by a spontaneous handwriting leads to a particular concentration and application of these children, who then obtain a sufficient qualitative handwriting. Indeed, they fail to achieve proficient handwriting at school, where handwriting times are longer and where the child is constantly in double-tasking. Moreover, the mild nature of HD in these children seems to occur to a relatively low frequency of the associated disorders identified during clinical assessments. This lower frequency would also lead to lighter consequences at the perceptual, perceptual-motor and motor control levels. Thus, HDs in these children appears mainly associated with a tone disorder characterized by a reduction in joint angles, particularly in the heel-ear angle which may underline an abnormal strengthening of the muscle chain leading to a certain tonicity emphasizing a non-release. This can be the cause of coordination difficulties and poor or limited coordination of the graphomotor gesture. Indeed, in our sample, children with handwriting disorders have poor synergistic coordination of the handwriting arm characterized by the persistence, whatever the age, of a progression along the line, consisting in moving the forearm and the elbow rather than a more mature rotation movement of the forearm at the elbow. They have also an instability of the wrist and a slow and hyper-controlled hand gesture. We can therefore assume that these children should benefit from better flexibility thanks to stretching activities and rehabilitation of coordination, and in particular about the prerequisites of the gestural tracing and segmental coordination of the graphomotor gesture by a psychomotor therapist. The slowness of handwriting in 43% of them can highlight possible deficits in phonological or metaphonological processing and in phoneme-grapheme conversion. Interestingly, the right anterior insula is strongly activated when writing letters, possibly related to phoneme-grapheme conversion [50]. It is logical to think that a child with poor recognition and phonological knowledge of the letter will have difficulty integrating the sensorimotor spatial form of the same letter. This hypothesis is confirmed by neuroimaging studies which show stronger activation of the premotor cortex, parietal cortex, cerebellum, and fusiform gyrus when typical children write an unknown letter (pseudoletter) than when they write a known letter. This is visible even though there is no difference in activation among poor writers [51]. These results support the hypothesis that the spatial shape of known letters is difficult to remember for children with HD. Even though we excluded detected speech impairments from our study, it is possible that some of the children had a phonological disturbance that would not have been detected. In the present study, we did not find any children for whom the HD could be explained exclusively by a depressive disorder. However, it is possible that a HD profile without other comorbidities may also reflect different assumptions about a psycho-affective problem. Our findings allow to assume that depression can be the cause of co-occurring difficulties leading to HD (executive, attentional or memory disorders). This can also be the consequence of HD which can lead to academic difficulties and remarks from adults even when the child is making significant efforts. It therefore seems that psychological care for these children could be useful in helping them improve their self-esteem and well-being, but it is not sufficient to treat the cause of the HD. It is interesting to note that 43% of the children showing a HD not detected by BHK are characterized by low energy (MDI-C). This may reflect the fatigability related to the cost of handwriting for these children. The high proportion of graphic visual-spatial coordination disabilities (71%) may be related to poor coordination of the graphomotor gesture [15], which would disturb the precision of the strokes, or to visuospatial perceptual difficulties. The mental planning disorder identified at Laby 5-12 (29% of children with HD not detected by BHK) is probably related to difficulties in visual-spatial perception and visual-motor coordination. Indeed, the Laby 5-12 requires significant visual-motor coordination skills, the fact of crossing the walls of the labyrinth with the pen being counted in the scoring. The association, in children with HDs not detected by BHK, between motor coordination and visual-motor graphic coordination disorders and slow reading (specifically when reading pseudo-words, which involve phonological skills) may suggest the involvement of the cerebellum. Indeed, several studies highlight a dysfunction at the level of the cerebellum in the comorbidity between DCD and dyslexia [52,53]. This hypothesis is also corroborated by Nicolson et al. [54] who demonstrate different cerebellar activity in dyslexic adults compared to typical adults. In addition, several studies highlight an association between mild disorders of gestural coordination and dyslexia [55,56].
At the same time, a significant proportion of children (41%) are classified by BHK test as having dysgraphia. The more pronounced character of the HD in these children seems to occur to a high frequency of the associated disorders identified during clinical assessments. This higher frequency would also lead to higher consequences at the perceptual, perceptual-motor, and motor control levels. Thus, children with dysgraphia have disabilities in motor coordination, manual skills and praxis, organization impairments of muscle tone with the presence of neurological soft signs, establishment of laterality, temporal identification, memory functions (kinesthetic and visual), visual perceptual functions, visual-motor integration, oculomotricity, auditory-attentional capacities and executive functions. They differ significantly (p <0.05) or almost significantly (p <0.10) from children presenting a less pronounced HD. We assume in these children that the oculomotor disorder (55%) may be the cause of visual-perceptual disorders and of the static balance disorder noted in 64% of children. Indeed, vision has a proprioceptive function and participates in tonico-postural regulation [57,58]. The impairment of oculomotor functions in our sample of children suggests a possible delay in the maturation of the oculomotor system, which notably involves the cerebello-cortical and cerebellar networks [59,60]. Furthermore, the visual-perceptual difficulties noted corroborate the studies in neuroimaging, which for the most part highlight an involvement of the ventral occipito-temporal cortex in writing [61], a structure involved in visual perception. The difficulties of temporal regulation and rhythmic adaptation that are only noted in children with dysgraphia support the hypothesis of a specific dysfunction of the cortical-subcortical pathways, which involve the cortical structures, the basal ganglia, and the thalamus. Indeed, these pathways would be involved in the motor adaptation skills and the learning of gestural sequences [62] in the temporal regularity of writing [63]. This again signals the importance of differentiating the diagnosis of dysgraphia from that of a less severe HD because dysgraphia is a neurodevelopmental disorder for which handwriting is really difficult or impossible. Thus, it is not acceptable today to put all HDs under the same umbrella, the remediation should be different according to the HD. The preponderance of oculomotricity disorder supporting the hypothesis of dysfunction of the cerebellum basal ganglia and superior colliculus structures being involved in oculomotor control [64] and sensorimotor functions (involving the cortico-subcortical pathways) in dysgraphic children highlights the importance of a transdisciplinary assessment of HDs. It is important that children identified as dysgraphic could undergo a complete visual and neurovisual assessment including oculomotricity. The body image disorder identified more specifically in dysgraphic children is combined with difficulties in integrating an internal representation of body segments in motion. This co-occurrence could be the result of sensory integration difficulties, especially on the proprioceptive level [65-67]. Our results attest to a multiplicity of functional impairments in children who are effectively dysgraphic, highlighting the need for a transdisciplinary panel of assessments, both of the graphic gesture, and at the neuropsychomotor, neuropsychological and oculomotor level. Depending on the situation, these children will need rehabilitation in psychomotricity, orthoptics, neuropsychology, or an occupational therapy to learn to use a computer in the classroom to compensate for his HD. It is also important to identify potential language disorders in these children, who will then need speech therapy.
Finally, 33% of children are classified by BHK test as having a moderate HD. They have fewer co-occurrences than children with dysgraphia but more difficulties than children with milder HDs. This again highlights the importance of differentiating between different degrees of HDs that do not respond to the same semiologies. The significantly higher frequency (p<0.05) of body spatial integration, visuomotor perceptual disorders, and of poorly asserted manual laterality in these children, reinforces the preponderance of the difficulties of sensorimotor integration in HD. Since sensorimotor skills are necessary for the internal representation of action, it is not surprising that impairment of these skills is involved in HD. These results are in line with the empirical data according to which the graphic space appears as the projection of a representation of the body with an up/down, left/right organization separated by an imaginary vertical median axis reference corresponding to the axis of the body [68]. The multi-modal and redundant integration of sensory information participates in the development of the internal sensorimotor representation of movement, which itself participates in the function of anticipation and planning of the action [69,70]. These results imply difficulties in anticipation and motor planning in children with HD who fail to correctly parameterize the spatial and temporal characteristics of the graphic trajectory. In addition, the high proportion of visuomotor perceptual disorders is congruent with the literature which concludes to an implication of visuomotor integration skills in HD [71-76]. The lack of kinesthetic memory in 67% of children with a moderate HD can lead to poor efficiency of the sensory feedbacksnecessary for the proper anticipation and planning of strokes when writing. The high proportion of these difficulties is congruent with the studies showing an influence of the kinesthetic capacities on the grip of the writing tool [77-78] and on the graphic quality [72,79].
5. Conclusions
Thus, our depth clinical examination made it possible to make underlying hypotheses for the involvement of different areas of the brain in HDs. These hypotheses would require further study in brain imaging. Our findings in the present study support the interest of performing a transdisciplinary and standardized clinical examination with developmental standards (neuropsychomotor, neuropsychological and oculomotor) in children with HD. Our results also highlighted the multiplicity of HDs and co-occurrences. This heterogeneity of the disorders is congruent with the neuroimaging studies, which underline the involvement of very large cortical areas as well as the parietal, temporal, frontal, occipital areas, and cerebellum [80,81]. HDs can therefore be associated with a multitude of different disorders ranging from a poor coordination of the graphomotor gesture to a more general and more complex impairment affecting perceptual-motor, cognitive and/or psycho-affective functions. However, it would be interesting to replicate our results with a larger sample of children in order to be able to carry out analyses school class by school class or by smaller age groups.

Round 2
Reviewer 1 Report
none
There are a few typos on page 7 in the new paragraph Design and Measures
assesment - assessment
folowed - followed
Author Response
Response to the reviewer 1
" Exploratory investigation of handwriting disorders in school-aged children from 1st to 5th grade"
There are a few typos on page 7 in the new paragraph
Thank you for this comment. We have corrected the few typos.
Additional information :
Concerning the description of statistical analysis :
We had made a vocabulary error concerning the statistical method. We had performed a Pearson chi-square test and not a Kruskal-Wallis test. We have modified this information in the document as follows and what is important is that there is no impact on the results and interpretation.
"The statistical analysis were carried out on R software (version 3.5.3). The degree of significance retained for all assignments was set at 0.05. Qualitative variables are described by numbers and percentages. A total of 71 binary variables (clinical variables) or tasks were considered. Tasks were scored by the psychomotor therapist 0 (success) and 1 (failure) based on percentile or standard deviation (below 1 SD or 10th percentile, depending on the test) in accordance with standardized instructions and developmental norms. For assessments of developmental features of handwriting, we used the developmental standards published in a previous study [14]. In order to compare the frequency of failure of clinical variables between the different levels of handwriting disorder (HD not detected by BHK scale, moderate HD, dysgraphia), a Pearson chi-square test was performed."
We added the results of Pearson chi-square test as follows:
"The percentage of failures in clinical functions was significantly different depending on the level of handwriting disorder underwent by BHK for only 4 clinical features. the. Thus, the more pronounced the writing disorder, the more frequent the disorder of visuo-perceptual capacities (χ2(2)=7.51, p=0.016 [95% CI, 26%-64%]) and the disorder of balance (χ2(2)= 8.17, p=0.016 [95% CI, 12%-47%]) in the population, and more particularly in the "dysgraphia" group. Bodily spatial integration disorder is absent in the "HD not detected by BHK" group and predominant in the "moderate HD" group (χ2(2)=7.88, p=0.019 [95% CI, 17%-54%]). The tendency to pessimism is strongly present in the "HD not detected by BHK" group and decreases with increasing level of HD (χ2(2)=6.00, p=0.046 [95% CI, 5%-35%])."
Article
Exploratory investigation of handwriting disorders in school-aged children from 1st to 5th grade
Abstract: Handwriting disorders (HDs) are prevalent in school-aged children with significant interference with academic performances. The current study offers a transdisciplinary approach with the use of normed and standardized clinical assessments of neuropsychomotor, neuropsychological and oculomotor functions. The aim is to provide objective data for a better understanding of the nature and the etiology of HDs. Data from these clinical assessments were analysed for 27 school-aged children with HD (1st to 5th grade). The results underline a high heterogeneity of the children presenting HDs with many co-occurrences often unknown. However, it was possible to highlight three levels of HDs based on BHK scores: mild HD not detected by the BHK test (26% of children), moderate HD (33%), dysgraphia (41% of children). The mild nature of HDs not detected by BHK seems to co-occurrences than children with dysgraphia but more difficulties than children with milder HDs. This highlights the importance of differentiating between different degrees of HDs that do not respond to the same semiologies. Our findings support the interest of performing a transdisciplinary and standardized clinical examination with developmental standards (neuropsychomotor, neuropsychological and oculomotor) in children with HD. Indeed, HDs can therefore be associated with a multitude of disorders of different nature ranging from poor coordination of the graphomotor gesture to a more general and more complex impairment affecting perceptual-motor, cognitive and/or psycho-affective functions.
Keywords: handwriting disorders; dysgraphia; children; semiology; neuropsychomotor assessment; neuropsychological assessment; oculomotricity
1. Introduction
As children spend 31-60% of their school day writing and performing other fine motor tasks [1], the development of handwriting skills is necessary for academic success [2,3] and the proper development of self-esteem [4,5]. According to the old version of the Diagnostic and Statistical Manual of Mental Disorders (DSM-IV-TR [6]), handwriting disorders (HDs) can be diagnosed in the case of "writing skills significantly lower than expected given chronological age, of measured intelligence and of an appropriate education”. The DSM-5 [7] described HDs as “Impairment in written expression” and dysgraphia is not described in the DSM-5. However, handwriting disorders (HD) affect between 10 and 30 % of school-aged children [8-10]. This is observed both by health professionals in clinical consultations and by teachers who struggle to adapt to the differences in the individual rhythms of these children and their learning difficulties. In this context, current studies on handwriting and its disorders attempt to provide new fundamental knowledge on the different processes involved in the development of handwriting, while clinical and therapeutic aspects remain little explored. Thus, handwriting disorder appears as an umbrella term defining a heterogeneous class of children exhibiting graphic impairments. The study of these disorders is complex as their understanding, both on the semiological level and on the etiological level, is still in the literature only in its early stages, and the definition of dysgraphia is unclear. Since the 1960s, it has been characterized by poor writing quality without any neurological or intellectual disorder being able to explain it [11]. This definition has been clarified by other authors who define dysgraphia as a disorder in written language partly linked to a lack of fine motor control in the execution of motor programs [12,13]. Recently, a relevant study [14] has shown phenotyping features in the genesis of pre-scriptural gestures in children to assess handwriting developmental levels because no recent research has previously think to study the developmental prerequisites of the handwriting organization. The better the quality of the handwriting gesture, the less variation there is in the inter-segmental organization coordinated during the writing task. This makes it possible to assess handwriting development levels in the context of screening for handwriting disorders [14]. Hence, another study was able to demonstrate for the first time the immaturity of the graphomotor gesture in children with a handwriting disorder, characterized both by a lack of synergistic coordination of the different segments of the writing arm and by an impairment of the temporal and kinematic characteristics of prescriptural traces (decrease in fluidity characterized by an increase in the number of strokes and velocity peaks, and an increase in drawing time and in-air pauses) [15,16]. The results about the impairment of the temporal and kinematic characteristics of handwriting are also corroborated by Asselborn et al. [17]). Moreover, generally, the authors highlight a lack in motor programming or in motor execution. Wann [18] suggests a motor programming defect characterized by altered temporal organization of writing (dysfunction, high pause times) due to the child's over-reliance on visual feedback. Lopez & Vaivre-Douret [19] suggest both proprioceptive/kinesthetic feedback deficits and a disruptive effect of visual control on the quality of pre-script drawings in these children, many of whom have kinaesthetic memory and visuo-spatial deficits. Thus, the ability to direct strokes would remain dependent on sensory feedback, itself insufficiently effective, leading to difficulties in achieving proactive control of handwriting. Other authors [20-22] suppose an impairment of the motor execution processes, which is characterized by a spatial, temporal, and kinematic irregularity of the writing characteristic in dysgraphic children. This would be the consequence of excessive neuromotor noises [21,23]. However, only one study has proposed a transdisciplinary investigation of handwriting disorders (HDs) [16]. The results highlighted a typology of three groups of HDs (mild ; mild to moderate; dysgraphia), each being associated with co-occurrences of specific neurodevelopmental dysfunctions: a co-occurrence of psycho-affective disorders that can be considered as a predictor of mild and moderate HDs; a co-occurrence of tone disorders and gross coordination that can be considered as a predictor of mild HDs; a co-occurrence of visual-spatial/constructive and attentional disorders which can be considered as a predictor of the most severe (dysgraphia) and moderate HDs. More specifically, a recent study proposed a transdisciplinary investigation of HDs in a cohort of children with a Developmental Coordination Disorder (DCD) [24]. This highlighted a significant association between neurological soft signs and the presence of dysgraphia in a sample of 65 children with DCD [24]. The dysgraphia appeared to be closely related to several specific dysfunctions of the laterality, to a minor neurological dysfunction of the pyramidal tract manifested by a distal phasic stretch reflex in the lower limb, and to a slowness in digital praxia. In addition to these few studies, it is important to enrich the literature concerning the analysis of the underlying clinical functions involved in handwriting disorders.
In the present study, we offer a transdisciplinary approach in depth with the use of normed and standardized clinical assessments of neuropsychomotor, neuropsychological and oculomotor functions. We aimed to provide objective data to better understand the nature and etiology of HDs.
2. Material and Methods
2.1. Participants
Data from a sample of 27 children with handwriting disorders (HDs) aged 6 years 2 months to 10 years 11 months (mean 8.15 SD 1.51) were collected from primary schools (grades 1 to 5) and in the usual out-patient consultation of Pediatrics, Cochin Port-Royal Hospital, and of Child Psychiatry department, Necker University Hospital, in Paris, France. Children were excluded from the study in case of prematurity (birth <37 WA), sensory, visual, neurological or genetic disorders, dyslexia and severe language disorder, ADHD (according to the DSM-5 criteria [7]), autism spectrum disorder, psychopathology, or motor disorder caused by injury or accident. None of them had repeated or skipped a grade or undergone any handwriting retraining at the time of the study. The institutional research ethics committee of Paris Descartes University approved the study procedures (CER·2018-72) conducted in accordance with the Declaration of Helsinki. All parents/legal guardians of participants provided written informed consent.
2.2. Design and Measures
Handwriting disorders were detected by the teachers and considered to be objective by an analysis of their class exercise books by an experienced psychomotor therapist. In order to assess their handwriting level, each child began to underwent the French adaptation of the standardized assessment of handwriting, the BHK scale [25] adapted from the Concise Evaluation Scale for children’s handwriting [26]. Fig 1 shows the study design previously published with the permission of the editor to reproduce it [15]. Then, a neuropsychomotor assesment (NP-MOT) was administrated and folowed by other tests (psychomotor, neurovisual, neuropsychological) proposed in different orders according to each child's motivation and time constraints. All the assesments were administrated on a single day, with breaks with a total around six hours of testing . The examination of oculomotor functions was recorded about twenty minutes at a second appointment on a different day. The psychomotor therapist investigator in the study administered all the tests.
Figure 1. Study design.
2.3. Handwriting assessment.
In addition to the BHK test, the children performed a previously validated cycloid loop line copying test[14,15]. Data on postural organization and inter-segmental coordination of the writing arm were systematically collected by video recording as described in previous studies [14,15]. Features about the proximal (head, trunk axis, shoulder, elbow, and forearm) and distal (wrist and fingers) gestural organization of the drawing process are collected. Variables relating the material (sheet, drawing line, pen) positioning and observational clinical variables related to the semiology of the motor characteristics of the gesture (control, pressure, synkinesis). In addition, spatio-temporal and kinematic measures were recorded using an Anoto digital pen with Elian Research software (Version 4.2, http://www.seldage.com, accessed on 24 September 2022) for which we have developed specific algorithms to record the measures above.
2.4. Clinical assessments
2.4.1. Neuropsychomotor assessment
All children performed neuropsychomotor physical tasks with the NP-MOT battery [27], including assessment of minor neurological dysfunctions (MND) exploring neurological soft signs like synkinesis (NSS). The age-standardized child assessment using the French NPMOT test battery is applicable to children as young as 4 years old. It has been found to have adequate test-retest reliability and internal consistency. Correlation coefficients of the NP-MOT with the BOTMP [28] range from 0.72 to 0.84, for motor coordination and balance. The NP-MOT battery enables physical assessment of passive/active muscular tone of limbs and axial tone (dangling and extensibility of wrist, shoulder, foot, heel-ear angle, popliteal angles, adductor angles, trunk), highlighting NSS denoting the existence of MND, such as limb pyramidal dysfunction. This is complemented by the assessment of basic motor function, control and regulation in gross motor tasks, gait, balance, coordination, manual dexterity, praxis, gnosopraxis (non-meaningful hand and finger imitation of gestures), digital perception, laterality, bodily spatial integration, rhythmic, and auditory attention tasks. The standardized NP-MOT battery is a developmental assessment because each subtest and milestone is scored from qualitative and quantitative viewpoints according to age, with each score converted to a standard deviation vs. mean, based on normative data for age and applicable to children as young as 4 years old [29]. There is a saturation of the maturation scores between 8 and 10 years, allowing to assess with the NP-MOT older children or adults.
2.4.2. Psychomotor assessment
The MABC-2 children's movement assessment battery (2nd edition) [30], adapted from the American battery [31] was used to assess psychomotor skills. It aims to assess motor impairments and is divided into three categories: manual dexterity (uni-manual, bi-manual test and visual-motor graphic tasks), target and catch (to throw a weighted bag/ball on a target and to catch a weighted bag or a ball), balance (static balance, walking and jumping tasks).
The gnosopraxis imitation of gestures assessment EMG [32,33] was used to assess the distal and digital gnosopraxic efficiency and to measure the child's ideomotor adaptation skills. It consists of performing imitations of arbitrary simple (with the hands) and complex (with the fingers) gestures in the absence of verbal command. This is an adaptation of the Bergès-Lézine assessment [34], paying particular attention to the gesture programming in the notation.
The Body Schema Test – Revised [35] was used to assess the child's representation of his own body and the relationships between different parts of his body. The task consists of a puzzle (non-contiguous pieces) of the body and face from the front (for children aged 3 to 8) and/or from a side view (for children aged 8 and over).
Spatial and temporal identification questions were asked in order to assess the knowledge and mastery of the spatio-temporal vocabulary.
A kinesthetic memorization test consisting of a reproduction test of asymbolic postures which had previously been printed and felt with the eyes closed, has been proposed in order to assess the body's perceptual skills [24].
2.4.3. Neurovisual assessment
Neurovisual aspects including visual gnosias, visual-perceptual, perceptual visual-motor, visuospatial, visuo-constructive skills and oculomotricity were assessed.
Visual perception was assessed using form recognition tasks [36], tangled lines and visual gnosia with outlines of animals, outlines of muddled fruits.
The KABC-II Shape Recognition subtest [37] consists in recognizing and naming drawings of various objects whose image has been altered (some lines of the drawing appear while others have been erased). This item assesses the child's ability to mentally represent the missing parts of the drawing to form a complete mental image, making it possible to name the represented object.
The Developmental Test of Visual-Motor Integration (VMI) (6th ed) [38] assessed pure perceptual abilities (the perceptual subtest of the test consisting of visual recognition of identical insignificant geometric shapes) and visuomotor integration abilities (subtest copy of the test figure consisting of the reproduction of simple and more complex insignificant geometric figures).
The NEPSY-II Arrows subtest [39] consisted of judging the direction, orientation and angles of different lines.
Rey’s complex geometric figure [40] allows the evaluation of aptitudes for perception, structuring and spatial organization (visual-spatial and visuo-constructive praxies). By copying and then reproducing from memory a complex geometric figure, it studies the ability to structure different elements in a graphic space.
The Code and Symbols subtests of the Wechsler intelligence scale for children and adolescents WISC-IV [41] for measuring a mental processing speed index (IVT) in connection with graphomotor capacities consist in analyzing and distinguishing non-significant signs.
The NEPSY-II Cubes subtest [39] and the Kohs block design [42] respectively assessed 3D visual-constructive skills and visual-spatial/constructive skills.
The examination of oculomotor functions was performed using an eye-tracking device made up of two infrared cameras positioned at the level of the inner corner of each eye (Ober consulting eye-Tracker Eyefant® [48]) recording the movements of fixation, smooth visual pursuit, and horizontal and vertical eye saccades. The device records in the horizontal and vertical planes at a sampling rate of 1000 Hz, a spatial resolution of 0.1 ° and a linearity range of +/- 35 ° horizontally and +/- 20 ° vertically.
2.4.4. Neuropsychological assessment
Visual-spatial attention, sustained auditory attention, and divided attention skills were assessed by a crossing test and the Childhood Attention Assessment Test (TEA-Ch [43]).
Executive functions (planning, inhibition skills, mental flexibility, working memory, and verbal fluency) were assessed by the Laby 5-12 labyrinths test [44], the NEPSY-II Categorization and Verbal Fluency subtests [42] and the Stroop's Selective Attention Test [45].
The visual, auditory and working memory skills were assessed by the Face Memory subtest of the NEPSY-II and by a face-up and back-up number span test (Odedys [46]).
The MDI-C Composite Childhood Depression Scale [47] assessed the emotional state of the child in order to identify a possible depressive state, this through 8 dimensions: self-esteem, anxiety, sad mood, social introversion, pessimism, mistrust, low energy and feelings of helplessness.
2.4.5. Language skills
The regular, irregular and pseudo-word reading test from the Odedys DYSlexia Screening Tool [46] assessed reading level and allowed to ruling out a diagnosis of dyslexia.
2.5. Statistical analyses
The statistical analysis were carried out on R software (version 3.5.3). The degree of significance retained for all assignments was set at 0.05. Qualitative variables are described by numbers and percentages. A total of 71 binary variables (clinical variables) or tasks were considered. Tasks were scored by the psychomotor therapist 0 (success) and 1 (failure) based on percentile or standard deviation (below 1 SD or 10th percentile, depending on the test) in accordance with standardized instructions and developmental norms. For assessments of developmental features of handwriting, we used the developmental standards published in a previous study [14]. In order to compare the frequency of failure of clinical variables between the different levels of handwriting disorder (HD not detected by BHK scale, moderate HD, dysgraphia), a Pearson chi-square test was performed.
3. Results
3.1. Characteristics of the sample
Twenty-seven children with handwriting disorders were included in this study, 4 girls (15%) and 23 boys (85%), aged 6 years 2 months to 10 years 11 months (mean 8.15 SD 1.51). Eleven of them (41%) presented dysgraphia on the BHK scale and nine (33%) presented more moderate handwriting disorders. In contrast, seven (26%) were not identified by the BHK test as presenting any handwriting disorder (see Table 1). Among the 27 children, six (22%) presented developmental coordination disorder (DCD) according to the DSM-5 criteria, and two (7%) had high intellectual potential (>130 IQ).
Table 1. Characteristics of the children.
|
|
Handwriting disorder not identified by the BHK test (n=7) |
Moderate handwriting disorder (n=9) |
Dysgraphia (n=11) |
Total (n=27) |
|
Age (years) [m (SD)] |
8.30 (0.81) |
7.79 (1.53) |
8.36 (1.87) |
8.15 (1.51) |
|
Gender [n(F/M)] |
0/7 |
1/8 |
3/8 |
4/23 |
n: number; m: mean; SD: standard deviation; F: female; M: male.
3.2. Results of the handwriting assessment
The detailed results of the sample about the postural and gestural organization and their spatial, temporal, and kinematic features of the drawings were described in a previous study [16]. Children with handwriting disorders have poor synergistic coordination of the handwriting arm characterised by the persistence, whatever the age, of a progression along the line, consisting in moving the forearm and the elbow rather than a more mature rotation movement of the forearm at the elbow. They have also an instability of the wrist and a slow and hyper-controlled hand gesture. Alongside, the drawing is characterised by poor quality, lack of fluidity, and slowness.
3.3. Percentage of clinical test failures (neuropsychomotor, psychomotor, neurovisual, neuropsychological, oculomotor, and language assessements)
Fig 2 presents the percentage of failures in neuropsychomotor and neuropsychological functions assessed by standardized clinical tests and of oculomotricity disorders identified during the examination of oculomotor functions. The variables are ordered by decreasing frequency of failure.
Figure 2. Percentage of failures in clinical functions assessed in the whole group.
Thus, the descriptive analysis of the results of the clinical tests highlights a set of quite varied disorders such as tone disorder, visual-motor graphic disorder, oculomotor disorder, lack of kinesthetic memory, disturbance of visual perceptual functions, and disorder of executive functions.
3.4. Frequency of failures in clinical assessments between the different levels of handwriting disorder (HD not detected by BHK scale, moderate HD, dysgraphia)
A more precise typology of HD was demonstrated by analyzing the distribution of failures in clinical functions according to the degree of writing disorder revealed by the BHK (see Table 2).
Table 2. Percentage of failures in clinical functions assessed for each of the groups classified by the BHK test.
|
Fonctions |
Whole HD group (n=27) |
HD not detected by BHK scale (n=7) |
Moderate HD on BHK (n=9) |
Dysgraphia on BHK (n=11) |
p-value |
|
Tone disorder (NP-MOT) |
74 |
71 |
78 |
73 |
0.95 |
|
Heel-ear angle reduction |
59 |
71 |
56 |
55 |
0.76 |
|
Affirmation of tonic laterality (NP-MOT) |
67 |
43 |
67 |
82 |
0.24 |
|
Visual memory (Rey, NEPSY-II) |
63 |
43 |
78 |
64 |
0.86 |
|
Hand-eye coordination (NP-MOT, MABC-2) |
56 |
43 |
67 |
55 |
0.64 |
|
Manual dexterity (NP-MOT , MABC-2) |
56 |
43 |
67 |
55 |
0.64 |
|
Graphic visual-spatial coordination |
78 |
71 |
78 |
82 |
0.88 |
|
One-hand coordination |
37 |
29 |
33 |
45 |
0.75 |
|
Bimanual coordination |
30 |
14 |
44 |
27 |
0.43 |
|
Perceptual visual-motor skills |
52 |
14 |
67 |
64 |
0.076 |
|
Codes test (WISC-IV) |
41 |
14 |
44 |
55 |
0.24 |
|
Copying figures (VMI) |
22 |
0 |
22 |
36 |
0.21 |
|
Symbols test (WISC-IV) |
15 |
0 |
11 |
27 |
0.28 |
|
Visual-spatial organization (Rey) |
7 |
0 |
11 |
9 |
0.69 |
|
Kinesthetic memory |
52 |
29 |
67 |
55 |
0.32 |
|
Oculomotricity |
44 |
29 |
44 |
55 |
0.26 |
|
Smooth pursuits |
37 |
14 |
33 |
55 |
0.16 |
|
Saccadic aye movement |
26 |
29 |
11 |
36 |
0.25 |
|
Fixation |
19 |
0 |
33 |
18 |
0.22 |
|
Visual-perceptual skills |
44 |
14 |
33 |
73 |
0.016 * |
|
Visual gnosia |
26 |
14 |
33 |
27 |
0.69 |
|
Perceptual (VMI) |
19 |
0 |
22 |
27 |
0.34 |
|
Mishmash of lines |
15 |
0 |
11 |
27 |
0.28 |
|
Pattern recognition |
15 |
0 |
22 |
18 |
0.44 |
|
Positions in space (Frostig) |
7 |
0 |
0 |
18 |
0.22 |
|
Exécutive functions |
41 |
29 |
33 |
55 |
0.49 |
|
Planning |
26 |
29 |
22 |
27 |
0.95 |
|
Categorization (Nepsy-II) |
15 |
14 |
0 |
27 |
0.25 |
|
Inhibition (Laby 5-12, Stroop) |
7 |
14 |
11 |
0 |
0.48 |
|
Verbal fluency (Nepsy-II) |
4 |
0 |
0 |
9 |
0.48 |
|
Auditory-attentional skills (TEA-Ch) |
41 |
29 |
33 |
55 |
0.49 |
|
Visual-attentional skills (TEA-Ch) |
41 |
43 |
56 |
27 |
0.45 |
|
Low énergy (MDI-C) |
41 |
43 |
44 |
36 |
0.93 |
|
Social introversion (MDI-C) |
37 |
14 |
22 |
64 |
0.06 |
|
Coordination between upper and lower limbs (NP-MOT) |
37 |
43 |
33 |
36 |
0.93 |
|
Dynamic balance coordination (NP-MOT) |
37 |
43 |
33 |
36 |
0.93 |
|
Static balance (NP-MOT) |
37 |
29 |
11 |
64 |
0.052 |
|
Aim and catch (MABC-2) |
37 |
14 |
22 |
64 |
0.06 |
|
Reading (Odedys) |
33 |
43 |
33 |
27 |
0.80 |
|
Time tracking (NP-MOT) |
33 |
0 |
33 |
55 |
0.06 |
|
Bodily spatial integration (NP-MOT) |
33 |
0 |
67 |
27 |
0.019 * |
|
Bimanual praxia (NP-MOT) |
33 |
0 |
44 |
45 |
0.10 |
|
Divided attention skills (TEA-Ch) |
30 |
29 |
22 |
36 |
0.79 |
|
Affirmation of manual laterality (NP-MOT) |
30 |
0 |
56 |
27 |
0.053 |
|
Neurological soft signs (NP-MOT) |
30 |
0 |
33 |
45 |
0.12 |
|
Imitation of fingers gestures (EMG) |
26 |
14 |
22 |
36 |
0.57 |
|
Balance (MABC-2) |
26 |
0 |
11 |
55 |
0.016 * |
|
Dressing praxia |
22 |
14 |
11 |
36 |
0.35 |
|
Anxiety (MDI-C) |
22 |
14 |
33 |
18 |
0.62 |
|
Feeling of helplessness (MDI-C) |
22 |
14 |
11 |
36 |
0.35 |
|
Dépressive disorder (MDI-C) |
19 |
14 |
22 |
18 |
0.92 |
|
Working memory (Odedys) |
19 |
14 |
22 |
18 |
0.92 |
|
usual discordant laterality (NP-MOT) |
19 |
0 |
33 |
18 |
0.25 |
|
Auditory memory (Odedys) |
15 |
0 |
11 |
27 |
0.28 |
|
Visual-constructive skills |
15 |
0 |
11 |
27 |
0.28 |
|
3D constructions (Nepsy-II) |
7 |
0 |
0 |
18 |
0.22 |
|
2D constructions (Kohs) |
7 |
0 |
11 |
9 |
0.69 |
|
Digital gnosia (NP-MOT) |
15 |
0 |
22 |
18 |
0.44 |
|
Oral-facial praxia |
15 |
0 |
11 |
27 |
0.29 |
|
Sad mood (MDI-C) |
15 |
29 |
22 |
0 |
0.20 |
|
Pessimism (MDI-C) |
15 |
43 |
11 |
0 |
0.046 * |
|
Body image (CORP-R) |
11 |
0 |
0 |
27 |
0.09 |
|
Rhythmic adaptation (NP-MOT) |
11 |
0 |
0 |
27 |
0.09 |
|
Digital praxia (NP-MOT) |
11 |
0 |
11 |
18 |
0.50 |
|
Spasticity (NP-MOT) |
11 |
0 |
22 |
9 |
0.37 |
|
Discordant tonic latérality (NP-MOT) |
7 |
0 |
11 |
9 |
0.69 |
|
Visual-spatial perception (Nepsy-II) |
7 |
14 |
11 |
0 |
0.48 |
|
Distrust (MDI-C) |
7 |
0 |
11 |
9 |
0.69 |
|
Imitation of hands gestures (EMG) |
4 |
0 |
0 |
9 |
0.48 |
|
Self estim (MDI-C) |
4 |
14 |
0 |
0 |
0.24 |
|
Non symbolic organisation of the gesture (NP-MOT) |
0 |
0 |
0 |
0 |
na |
The percentage of failures in clinical functions was significantly different depending on the level of handwriting disorder underwent by BHK for only 4 clinical features. the. Thus, the more pronounced the writing disorder, the more frequent the disorder of visuo-perceptual capacities (χ2(2)=7.51, p=0.016 [95% CI, 26%-64%])and the disorder of balance (χ2(2)= 8.17, p=0.016 [95% CI, 12%-47%]) in the population, and more particularly in the "dysgraphia" group. Bodily spatial integration disorder is absent in the "HD not detected by BHK" group and predominant in the "moderate HD" group (χ2(2)=7.88, p=0.019 [95% CI, 17%-54%]). The tendency to pessimism is strongly present in the "HD not detected by BHK" group and decreases with increasing level of HD (χ2(2)=6.00, p=0.046 [95% CI, 5%-35%]).
A trend close to statistical significance is observed for 8 clinical features (p <0.10) with a higher frequency of 6 of them in the "dysgraphia" group and of 2 of them in the "moderate HD" group. Their frequency is never the highest in the "HD not detected by BHK" group. Thus, when the level of HD increases on the one hand, so do static balance disorders, aiming and catching difficulties, temporal identification, body diagram, rhythmic adaptation disorders, and social introversion. These disabilities are particularly common in the "dysgraphia" group. Disorders of visuomotor perceptual capacities and affirmation of manual laterality are respectively the least frequent (14%) and absent in the "HD not detected by BHK" group. They are in the majority in the "moderate HD" group.
The descriptive analysis of the whole clinical features reveals a higher proportion of co-occurrences in the "dysgraphia" group than in the "moderate HD" group, whereas the children identified as having a " HD not detected by BHK "appear less affected”.
Thus, when the level of HD increases, there is a greater proportion of neuro-psychomotor, neuropsychological and oculomotricity disorders. On the other hand, psycho-affective disorders such as a sad mood, a tendency to pessimism, and low self-esteem appear in the majority of children with a HD not detected by BHK. Psycho-affective disorders are therefore a possible origin of the less pronounced writing disorders.
3.5. Failures greater than 40% in each of the three groups identified by the BHK test (HD not detected by BHK, moderate HD, dysgraphia)
The analysis of Table 2 shows the following results.
In the "HD not detected by BHK" group, the tone disorder (reduction in joint angles measured during passive tone examination), the heel-ear angle reduction, and the graphic visual-spatial coordination disorder appear at a frequency greater than 50%. However, there is no significant difference between groups for these clinical variables. This is explained by the presence of these difficulties in the clinical group as a whole, irrespective of the level of HD. The disorders of the coordination between the upper and lower limbs, of the coordination of static balance and a slowness of reading appear at a frequency greater than 40% but not significantly. Because they also appear to be highly unsuccessful throughout the whole clinical sample. The tendency to pessimism appears at a frequency greater than 40% in the "HD not detected by BHK" group with a difference between groups close to significance (p=0.07). In addition, a factor analysis revealed a co-occurrence of psycho-affective disorders (depression, lack of self-esteem, sad mood, feeling of helplessness, pessimism, low energy, anxiety) associated with the "HD not detected by BHK" group.
In the "moderate HD" group, tone disorder, heel-ear angle reduction, poorly asserted tonic laterality, poorly asserted manual laterality, visual memory disorder, manual and oculo-manual disorders, graphic visual-spatial coordination disorder, perceptual visual-motor disorder, kinesthetic memory disorder, visual-attentional disorder, and bodily spatial integration disorder appear at a frequency greater than 50%. Among these disabilities, only a trouble of spatial integration of the body appears to be significantly more frequent in the "moderate HD" group (p=0.03). Once again this seems to be because the variables appear to be mostly unsuccessful in the whole clinical group. Bimanual coordination and praxia disorders, low energy, oculomotricity disorder and Codes test failure occur at a frequency greater than 40% but not specifically in the group "moderate HD".
In the "Dysgraphia" group, appear at a frequency greater than 50% : a tone disorder with a heel-ear angle reduction, a poorly asserted tonic laterality, disorders of visual memory, manual and oculo-manual skills, visual-motor visual coordination, perceptual visuo-motor capacities (in particular with the test of the WISC-IV Codes), kinesthetic memory, oculomotricity and especially in smooth pursuits, disturbances of visuo-perceptual capacities, of executive functions, of auditory-attentional capacities, of static balance and of the capacities to aim and catch, and a disorder of temporal identification. Among these disorders, the static balance disorders identified with MABC-2 and NP-MOT is significantly more frequent in the "dysgraphia" group (respectively p=0.01 and p=0.049), as well as the disorder of aiming and catching abilities (p=0.049) and visual-perceptual skills disorder (p=0.04). Neurological soft signs (synkinesis), a disorder of uni-manual and bimanual coordinations appear at a frequency greater than 40% and are in the majority in the "Dysgraphia" group, but they are not specific to this group. In addition, a factor analysis revealed a co-occurrence of visual-spatial/constructive and attentional disorders related to an oculomotor disorder (visual fixation) and associated with the "dysgraphia" group.
4. Discussion
Our whole sample of 27 children with HD included in the present study underwent a complete developmental battery of neuropsychomotor, neuropsychological and oculomotor assessments. The aim of this transdisciplinary study was to better understand the complexity of the semiology of HD because the literature is poor. The present study is an important state of handwriting disorders, as to our knowledge, no research has explored such a broad set of skills to better understand the etiology of HD. Our results underline a high heterogeneity of the children presenting a HD and allows us to identify etiological hypotheses. A previous study highlighted the co-occurrence of difficulties associated with handwriting disorders in children [16]. However, the factor analysis carried out in this same study could only explain 28% of the variance in the sample, which is consistent with the heterogeneity of handwriting disorders highlighted in this article.
In a previous study about the influence of visual control on the quality of the graphic gesture in children with handwriting disorders [19], we hypothesized the involvement of the cortico-striatal and cortico-cerebellar pathways in HD. The hypothesis of cerebellar dysfunction in children with HDs is accepted in the literature [49]. Our clinical sample being very heterogeneous, other etiological hypotheses can be proposed.
Our findings showed an important percentage of children (26%) exhibiting a HD penalizing them at school and being notable in class notebooks but not detected by the BHK test. This lack of detection of HDs is explained by a slight degradation of the handwriting when copying a paragraph (like proposed in the BHK test). Probably, the dual and evaluative situation induced by a copy with the BHK and not by a spontaneous handwriting leads to a particular concentration and application of these children, who then obtain a sufficient qualitative handwriting. Indeed, they fail to achieve proficient handwriting at school, where handwriting times are longer and where the child is constantly in double-tasking. Moreover, the mild nature of HD in these children seems to occur to a relatively low frequency of the associated disorders identified during clinical assessments. This lower frequency would also lead to lighter consequences at the perceptual, perceptual-motor and motor control levels. Thus, HDs in these children appears mainly associated with a tone disorder characterized by a reduction in joint angles, particularly in the heel-ear angle which may underline an abnormal strengthening of the muscle chain leading to a certain tonicity emphasizing a non-release. This can be the cause of coordination difficulties and poor or limited coordination of the graphomotor gesture. Indeed, in our sample, children with handwriting disorders have poor synergistic coordination of the handwriting arm characterized by the persistence, whatever the age, of a progression along the line, consisting in moving the forearm and the elbow rather than a more mature rotation movement of the forearm at the elbow. They have also an instability of the wrist and a slow and hyper-controlled hand gesture. We can therefore assume that these children should benefit from better flexibility thanks to stretching activities and rehabilitation of coordination, and in particular about the prerequisites of the gestural tracing and segmental coordination of the graphomotor gesture by a psychomotor therapist. The slowness of handwriting in 43% of them can highlight possible deficits in phonological or metaphonological processing and in phoneme-grapheme conversion. Interestingly, the right anterior insula is strongly activated when writing letters, possibly related to phoneme-grapheme conversion [50]. It is logical to think that a child with poor recognition and phonological knowledge of the letter will have difficulty integrating the sensorimotor spatial form of the same letter. This hypothesis is confirmed by neuroimaging studies which show stronger activation of the premotor cortex, parietal cortex, cerebellum, and fusiform gyrus when typical children write an unknown letter (pseudoletter) than when they write a known letter. This is visible even though there is no difference in activation among poor writers [51]. These results support the hypothesis that the spatial shape of known letters is difficult to remember for children with HD. Even though we excluded detected speech impairments from our study, it is possible that some of the children had a phonological disturbance that would not have been detected. In the present study, we did not find any children for whom the HD could be explained exclusively by a depressive disorder. However, it is possible that a HD profile without other comorbidities may also reflect different assumptions about a psycho-affective problem. Our findings allow to assume that depression can be the cause of co-occurring difficulties leading to HD (executive, attentional or memory disorders). This can also be the consequence of HD which can lead to academic difficulties and remarks from adults even when the child is making significant efforts. It therefore seems that psychological care for these children could be useful in helping them improve their self-esteem and well-being, but it is not sufficient to treat the cause of the HD. It is interesting to note that 43% of the children showing a HD not detected by BHK are characterized by low energy (MDI-C). This may reflect the fatigability related to the cost of handwriting for these children. The high proportion of graphic visual-spatial coordination disabilities (71%) may be related to poor coordination of the graphomotor gesture [15], which would disturb the precision of the strokes, or to visuospatial perceptual difficulties. The mental planning disorder identified at Laby 5-12 (29% of children with HD not detected by BHK) is probably related to difficulties in visual-spatial perception and visual-motor coordination. Indeed, the Laby 5-12 requires significant visual-motor coordination skills, the fact of crossing the walls of the labyrinth with the pen being counted in the scoring. The association, in children with HDs not detected by BHK, between motor coordination and visual-motor graphic coordination disorders and slow reading (specifically when reading pseudo-words, which involve phonological skills) may suggest the involvement of the cerebellum. Indeed, several studies highlight a dysfunction at the level of the cerebellum in the comorbidity between DCD and dyslexia [52,53]. This hypothesis is also corroborated by Nicolson et al. [54] who demonstrate different cerebellar activity in dyslexic adults compared to typical adults. In addition, several studies highlight an association between mild disorders of gestural coordination and dyslexia [55,56].
At the same time, a significant proportion of children (41%) are classified by BHK test as having dysgraphia. The more pronounced character of the HD in these children seems to occur to a high frequency of the associated disorders identified during clinical assessments. This higher frequency would also lead to higher consequences at the perceptual, perceptual-motor, and motor control levels. Thus, children with dysgraphia have disabilities in motor coordination, manual skills and praxis, organization impairments of muscle tone with the presence of neurological soft signs, establishment of laterality, temporal identification, memory functions (kinesthetic and visual), visual perceptual functions, visual-motor integration, oculomotricity, auditory-attentional capacities and executive functions. They differ significantly (p <0.05) or almost significantly (p <0.10) from children presenting a less pronounced HD. We assume in these children that the oculomotor disorder (55%) may be the cause of visual-perceptual disorders and of the static balance disorder noted in 64% of children. Indeed, vision has a proprioceptive function and participates in tonico-postural regulation [57,58]. The impairment of oculomotor functions in our sample of children suggests a possible delay in the maturation of the oculomotor system, which notably involves the cerebello-cortical and cerebellar networks [59,60]. Furthermore, the visual-perceptual difficulties noted corroborate the studies in neuroimaging, which for the most part highlight an involvement of the ventral occipito-temporal cortex in writing [61], a structure involved in visual perception. The difficulties of temporal regulation and rhythmic adaptation that are only noted in children with dysgraphia support the hypothesis of a specific dysfunction of the cortical-subcortical pathways, which involve the cortical structures, the basal ganglia, and the thalamus. Indeed, these pathways would be involved in the motor adaptation skills and the learning of gestural sequences [62] in the temporal regularity of writing [63]. This again signals the importance of differentiating the diagnosis of dysgraphia from that of a less severe HD because dysgraphia is a neurodevelopmental disorder for which handwriting is really difficult or impossible. Thus, it is not acceptable today to put all HDs under the same umbrella, the remediation should be different according to the HD. The preponderance of oculomotricity disorder supporting the hypothesis of dysfunction of the cerebellum basal ganglia and superior colliculus structures being involved in oculomotor control [64] and sensorimotor functions (involving the cortico-subcortical pathways) in dysgraphic children highlights the importance of a transdisciplinary assessment of HDs. It is important that children identified as dysgraphic could undergo a complete visual and neurovisual assessment including oculomotricity. The body image disorder identified more specifically in dysgraphic children is combined with difficulties in integrating an internal representation of body segments in motion. This co-occurrence could be the result of sensory integration difficulties, especially on the proprioceptive level [65-67]. Our results attest to a multiplicity of functional impairments in children who are effectively dysgraphic, highlighting the need for a transdisciplinary panel of assessments, both of the graphic gesture, and at the neuropsychomotor, neuropsychological and oculomotor level. Depending on the situation, these children will need rehabilitation in psychomotricity, orthoptics, neuropsychology, or an occupational therapy to learn to use a computer in the classroom to compensate for his HD. It is also important to identify potential language disorders in these children, who will then need speech therapy.
Finally, 33% of children are classified by BHK test as having a moderate HD. They have fewer co-occurrences than children with dysgraphia but more difficulties than children with milder HDs. This again highlights the importance of differentiating between different degrees of HDs that do not respond to the same semiologies. The significantly higher frequency (p<0.05) of body spatial integration, visuomotor perceptual disorders, and of poorly asserted manual laterality in these children, reinforces the preponderance of the difficulties of sensorimotor integration in HD. Since sensorimotor skills are necessary for the internal representation of action, it is not surprising that impairment of these skills is involved in HD. These results are in line with the empirical data according to which the graphic space appears as the projection of a representation of the body with an up/down, left/right organization separated by an imaginary vertical median axis reference corresponding to the axis of the body [68]. The multi-modal and redundant integration of sensory information participates in the development of the internal sensorimotor representation of movement, which itself participates in the function of anticipation and planning of the action [69,70]. These results imply difficulties in anticipation and motor planning in children with HD who fail to correctly parameterize the spatial and temporal characteristics of the graphic trajectory. In addition, the high proportion of visuomotor perceptual disorders is congruent with the literature which concludes to an implication of visuomotor integration skills in HD [71-76]. The lack of kinesthetic memory in 67% of children with a moderate HD can lead to poor efficiency of the sensory feedbacksnecessary for the proper anticipation and planning of strokes when writing. The high proportion of these difficulties is congruent with the studies showing an influence of the kinesthetic capacities on the grip of the writing tool [77-78] and on the graphic quality [72,79].
5. Conclusions
Thus, our depth clinical examination made it possible to make underlying hypotheses for the involvement of different areas of the brain in HDs. These hypotheses would require further study in brain imaging. Our findings in the present study support the interest of performing a transdisciplinary and standardized clinical examination with developmental standards (neuropsychomotor, neuropsychological and oculomotor) in children with HD. Our results also highlighted the multiplicity of HDs and co-occurrences. This heterogeneity of the disorders is congruent with the neuroimaging studies, which underline the involvement of very large cortical areas as well as the parietal, temporal, frontal, occipital areas, and cerebellum [80,81]. HDs can therefore be associated with a multitude of different disorders ranging from a poor coordination of the graphomotor gesture to a more general and more complex impairment affecting perceptual-motor, cognitive and/or psycho-affective functions. However, it would be interesting to replicate our results with a larger sample of children in order to be able to carry out analyses school class by school class or by smaller age groups.
Author Contributions: C.L. and L.V.-D. conceived the study and designed the methodology; C.L. recruited the participants and collected the data; C.L. and L.V.-D. analyzed the data; C.L. and L.V.-D. performed the interpretation of results; C.L. prepared the tables and figures; C.L. and L.V.-D. wrote the manuscript. All authors have read and agreed to the published version of the manuscript.
Funding: This research was funded Institut Universitaire de France (IUF), Grant n° L17P99-IUF004, Paris, France, and partly by the Fondation de France, with the Fondation pour la Recherche en Psychomotricité et Maladies de Civilisation (FRPMC).
Institutional Review Board Statement: The institutional research ethics committee of Paris Descartes University approved the study procedures (CER·2018-72, date 18 June 2018) conducted in accordance with the Declaration of Helsinki.
Informed Consent Statement: All the parents and the children provided written informed consent. The data collected was anonymised.
Data Availability Statement: The data presented in this study are available on request from the corresponding author.
Acknowledgments: We are grateful to the children and their families who took part in the study. The authors thank Lucie Coussement for her help to check the English writing of the manuscript.
Conflicts of Interest: The authors have stated that they had no interest which might be perceived as posing a conflict or bias.
References
- McHale, K.; Cermak, S.A. Fine motor activities in elementary school: Preliminary findings and provisional implications for children with fine motor problems. J. Occ. Ther. 1992, 46, 898–903.
- Sassoon, R. Handwriting : A New Perspective; Stanley Thornes: Chelthenham, UK, 1990.
- Stewart, S.R. Developement of written language proficiency: Methods for teaching text structure. In Communication Skills and Classroom Success; Simon, C.C., Ed.; Taylor & Francis: United Kingdom, 1992.
- Chang, S.H., Yu, N.Y. Handwriting movement analyses comparing first and second graders with normal or dysgraphic characteristics. Dev. Disabil. 2013, 34, 2433-2441.
- Feder, K.P., Majnemer, A. Handwriting development, competency, and intervention. Med. Child Neurol. 2007, 49, 312-317.
- Diagnostic and Statistical Manual of Mental Disorders, DSM IV-TR, 4th ed.; American Psychiatric Publishing: Arlington, MA, USA, 2000.
- Diagnostic and Statistical Manual of Mental Disorders, DSM-5, 5th ed.; American Psychiatric Publishing: Arlington, MA, USA, 2013.
- Karlsdottir, R., Stefansson, T. Problems in developing functional handwriting. Mot. Ski. 2002, 94, 623-662.
- Maeland, A.E. Handwriting and perceptual motor skills in clumsy, dysgraphic, and normal children. Mot. Ski. 1992, 75, 1207-1217.
- Naider-Steinhart, S., Katz-Leurer, M. Analysis of proximal and distal muscle activity during handwriting tasks. J. Occ. Ther. 2007, 61, 392–398.
- Ajuriaguerra (de), J., Auzias, M., Denner, A. L’écriture de l’enfant. L’évolution de l’écriture et ses difficultés, 4e , Vol. 1 [The child's writing. The evolution of writing and its difficulties, 4e ed., Vol. 1]; Delachaux et Niestlé: Paris, France, 1964.
- Hamstra-Bletz, L., Blöte, A.W. A Longitudinal Study on Dysgraphic Handwriting in Primary School. J Learn Disabil. 1993, 26, 689–699.
- Van Dorn, R.R.A., Keuss, P.J.G. Dysfluency in children’s handwriting. In The Development of Graphic Skills; Wann, J., Wing, A.M., Sovik, N., Eds.; Academic Press; Cambridge, MA, USA, 1991.
- Vaivre-Douret, L., Lopez, C., Dutruel, A., Vaivre, S. Phenotyping features in the genesis of pre-scriptural gestures in children to assess handwriting developmental levels. Sci Rep. 2021, 11, 731.
- Lopez, C., Vaivre-Douret, L. Concurrent and predictive validity of a cycloid loops copy task to assess handwriting disorders in children. Children 2023, 10 (305), 1-14
- Lopez, C., Vaivre-Douret, L. Investigations exploratoires des troubles de l’écriture chez des enfants scolarisés du CP au CM2 [Exploratory investigations of handwriting disorders in children from 1st to 5th grade]. ANAE 2021, 170, 77-89.
- Asselborn, T., Chapatte, M., Dillenbourg, P. Extending the Spectrum of Dysgraphia: A Data Driven Strategy to Estimate Handwriting Quality. Rep.2020, 10, 31-40.
- Wann, J.P. Handwriting disturbance: Developmental trends. In Themes in motor development; Whiting, H.T.A., Wade, M.G., Eds.; Martinus Nijhoff Publishers: Leyde, The Netherlands, 1986.
- Lopez, C., Vaivre-Douret, L. Influence of visual control on the quality of graphic gesture in children with handwriting disorders. Rep. 2021, 11, 23537.
- Smits-Engelsman, B.C., van Galen, G.P. Dysgraphia in Children: Lasting Psychomotor Deficiency or Transient Developmental Delay? Exp. Child Psychol. 1997, 67, 164-184.
- van Galen, G.P., Portier, S.J., Smits-Engelsman, B.C., Schomaker, L.R. Neuromotor noise and poor handwriting in children. Acta Psychol. 1993, 82, 161–178.
- Wann, J.P., Kardirkamanathan, M. Variability in children’s handwriting: Computer diagnosis of writing difficulties. In The Development of Graphic Skills; Wann, J., Wing, A.M., Sovik, N., Eds.; Academic Press: Cambrige, MA, USA, 1991.
- Zesiger, P. Acquisition et troubles de l’écriture [Learning to write and hardwriting disorders]. Enfance 2003, 55, 56-64.
- Lopez, C., Hemimou, C., Golse, B., Vaivre-Douret, L. Developmental dysgraphia is often associated with minor neurological dysfunction in children with developmental coordination disorder (DCD). Neurophysiol. 2018, 48, 207-217.
- Charles, M., Soppelsa, R., Albaret, J.M. Échelle d’évaluation rapide de l’écriture chez l’enfant (BHK) [Concise Evaluation Scale for children’s handwriting (BHK)]. ECPA-Pearson: Paris, 2004.
- Hamstra-Bletz, L., DeBie, J., Den Brinker, B. Concise Evaluation Scale for children’s handwriting. Swets Zeitlinger: Lisse, The Netherlands, 1987.
- Vaivre-Douret, L. Batterie d’évaluation des fonctions neuro-psychomotrices (NP-MOT) [Neuro-psychomotor functions assessment battery]. ECPA-Pearson: Paris, 2006.
- Bruininks, R.H. Bruininks Oseretsky test of motor proficiency. American Guidance Service: Minnesota, 1978.
- Vaivre-Douret, L. Outil numérique de notation et cotation normées de l’évaluation développementale standardisée des fonctions Neuro-Psychomotrices de l’enfant, NP-MOT jusqu’à 12 ans et plus [Digital scoring tool and standardized scoring of the developmental standardized assessment of the neuro-psychomotor functions of the child, NP-MOT up to 12 years and older]. In Battery of Child Neuro-Psychomotor Functions; Vaivre-Douret, L., Digital ed.; Editions Neuralix: Paris, France, 2021.
- Marquet-Doléac, J., Soppelsa, R., Albaret, J.M. MABC-2 Batterie d’évaluation du mouvement chez l’enfant, 2e éd. [MABC-2 Movement assessment battery for children, 2d ed.]. ECPA-Pearson: Paris, France, 2016.
- Henderson, S.E., Sugden, D.A., Barnett, A. Movement Assessment Battery for Children-2: Movement ABC-2.  Pearson: United Kingdom, 2007.
- Vaivre-Douret, L. Evaluation de la motricité gnosopraxique distale EMG [Assessment of distal gnosopraxic motor skills EMG]. ECPA-Pearson: Paris, France, 1997.
- Vaivre-Douret, L. Normes 8-12 ans pour la notation automatisée de l’évaluation de la motricité gnosopraxique (EMG). In Evaluation de la motricité gnosopraxique distale : révision et adaptation du test de Bergès-Lézine; Vaivre-Douret, L., CD-ROM Ed. ECPA-Pearson: Paris, France, 2021.
- Vaivre-Douret, L. Evaluation de la motricité gnoso-praxique distale chez l’enfant. Une adaptation du test d’imitation de gestes de Bergès-Lézine [Evaluation of the distal motor gnosopraxia. An adaptation of Berges and Lezine's Imitation of Gestures test]. ANAE 1999, 51, 13-20.
- Meljac, C., Fauconnier, E., Scalabrini, J. Schéma corporel-R. Épreuve de Schéma Corporel - Révisée [Body schema-R. Body Schema Test - Revised]. ECPA-Pearson: Paris, France, 2010.
- Frostig, M. Test de développement de la perception visuelle: Manuel [Visual Perception Development Test: Manual]. ECPA-Pearson: Paris, France, 1973.
- Kaufman, A.S., Kaufman, N.L. KABC-II - Batterie pour l’examen psychologique de l’enfant, 2e éd. [KABC-II - Battery for the psychological examination of the child, 2nd ed.]. ECPA-Pearson: Paris, France, 2008.
- Beery, K.E., Buktenica, N.A., Beery, N.A. The Beery–Buktenica Developmental Test of Visual–Motor Integratio : Administration, scoring, and teaching manual, 6th ed. Pearson: London, United Kingdom, 2010.
- Korkman, M., Kirk, U., Kemp, S. NEPSY-II - Bilan neuropsychologique de l’enfant, 2e ed. [NEPSY-II - Neuropsychological assessment of the child, 2nd ed.]. ECPA-Pearson: Paris, France, 2012.
- Wallon, P., Mesmin, C. FCR - Test de la figure complexe de Rey [CFR - Test of the complex figure of Rey]. ECPA-Pearson: Paris, France, 2009.
- Wechsler, D. WISC-IV - Echelle d’intelligence de Wechsler pour enfants et adolescents, 4e éd.) [WISC-IV - Wechsler Intelligence Scale for Children and adolescents, 4th ed.]. ECPA-Pearson: Paris, France, 2005.
- Kohs, S. Test des cubes de Kohs: Manuel d’application [Kohs Cube test: Manual]. ECPA-Pearson: Paris, France, 1972.
- Manly, T., Robertson, I.H., Anderson, V., Mimmo-Smith, I. TEA-Ch - Test d’Évaluation de l’Attention chez l’enfant [TEAC-Ch - Childhood Attention Assessment Test]. ECPA-Pearson: Paris, France, 2006.
- Marquet-Doléac, J., Soppelsa, R., Albaret, J.M. Laby 5-12: Test des labyrinthes pour les enfants de 5 à 12 ans [Laby 5-12: Labyrinths test for children aged 5 to 12]. Éditions Hogrefe: Paris, France, 2010.
- Albaret, J.M., Migliore, L. Stroop - Test d’attention sélective de Stroop [Stroop - Stroop selective attention test]. ECPA-Pearson: Paris, France, 1999.
- Jacquier-Roux, M., Valdois, S., Zorman, M., Lequette, C., Pouget, G. ODEDYS - Outil de dépistage des dyslexies, 2e éd. [ODEDYS - Dyslexia Screening Tool, 2nd ed.]. Cogni-sciences: Grenoble, France, 2005.
- Berndt, D.J., Kaiser, C.F. MDI-C - Échelle composite de dépression pour enfants [MDI-C - Children's Composite Depression Scale]. ECPA-Pearson: Paris, France, 1999.
- Jazz Novo Ober Consulting. Ober Consulting Homepage®, 2018. http://www.ober-consulting.com/9/lang/1/
- Wilson, P.H., Ruddock, S., Smits-Engelsman, B.C.M., Polatajko, H., Blank, R. Understanding performance deficits in developmental coordination disorder: A meta-analysis of recent research. Med. Child Neurol. 2013, 55, 217-228.
- Longcamp, M., Lagarrigue, A., Nazarian, B., Roth, M., Anton, J.L., Alario, F.X. et al. Functional specificity in the motor system: Evidence from coupled fMRI and kinematic recordings during letter and digit writing: Functional Specificity During Letter and Digit Writing. Brain Mapp. 2014, 35, 6077-6087.
- Richards, T.L., Berninger, V.W., Stock, P., Altemeier, L., Trivedi, P., Maravilla, K.R. Differences between good and poor child writers on fMRI contrasts for writing newly taught and highly practiced letter forms. Writ. 2011, 24, 493-516.
- Nicolson, R.I., Fawcett, A.J., Dean, P. Developmental dyslexia: The cerebellar deficit hypothesis. Trends Neurosci. 2001, 24, 508-511.
- Zwicker, J.G., Missiuna, C., Boyd, L.A. Neural Correlates of Developmental Coordination Disorder: A Review of Hypotheses. Child Neurol. 2009, 24, 1273-1281.
- Nicolson, R.I., Fawcett, A.J., Berry, E.L., Jenkins, I.H., Dean, P., Brooks, D.J. Association of abnormal cerebellar activation with motor learning difficulties in dyslexic adults. The Lancet 1999, 353, 1662-1667.
- Fawcett, A.J., Nicolson, R.I. Performance of Dyslexic Children on Cerebellar and Cognitive Tests. Mot. Behav. 1999, 31, 68-78.
- Velay, JL. Interhemispheric sensorimotor integration in pointing movements : A study on dyslexic adults. Neuropsychologia 2002, 40, 827-834.
- Lee, D.N., Lishman, J.R. Visual proprioceptive control of stance. Hum. Mov. Stud. 1975, 1, 87–95.
- Berthoz, A., Lacour, M., Soechting, J.F., Vidal, P.P. The role of vision during linear motion. In Reflex control of posture and movement. Progress in brain research, Vol 50.; Granit, R., Pompeiano, O., eds. Elsevier: Amsterdam, The Netherlands, 1979.
- Gaymard, B., Giannitelli, M., Challes, G., Rivaud-Péchoux, S., Bonnot, O., Cohen, D. et al. Oculomotor Impairments in Developmental Dyspraxia. The Cerebellum 2017, 16, 411-420.
- Robert, M.P., Ingster-Moati, I., Albuisson, E., Cabrol, D., Golse, B., Vaivre-Douret, L. Vertical and horizontal smooth pursuit eye movements in children with developmental coordination disorder. Med. Child Neurol. 2014, 56, 595-600.
- Planton, S., Longcamp, M., Péran, P., Démonet, J.F., Jucla, M. How specialized are writing-specific brain regions? An fMRI study of writing, drawing and oral spelling. Cortex 2017, 88, 66-80.
- Gheysen, F., Van Waelvelde, H., Fias, W. Impaired visuo-motor sequence learning in Developmental Coordination Disorder. Dev. Disabil. 2011, 32, 749-756.
- Ben-Pazi, H., Kukke, S., Sanger, T.D. Poor Penmanship in Children Correlates With Abnormal Rhythmic Tapping: A Broad Functional Temporal Impairment. Child Neurol. 2007, 22, 543-549.
- Collewijn, H., Kowler, E. The significance of microsaccades for vision and oculomotor control. Vis. 2008, 8, 20-21.
- Paillard, J. Le Corps et ses langages d’espace [The body and its languages of space]. In Le corps en psychiatrie [The body in psychiatry]; Jeddi, E., ed. Masson: Paris, France, 1982.
- Pfeiffer, B., Moskowitz, B., Paoletti, A., Brusilovskiy, E., Zylstra, S.E., Murray, T. Developmental Test of Visual–Motor Integration (VMI): An Effective Outcome Measure for Handwriting Interventions for Kindergarten, First-Grade, and Second-Grade Students? J. Occ. Ther. 2015, 69, 1-7.
- Proske, U., Gandevia, S. The Proprioceptive Senses : Their Roles in Signaling BodyShape, Body Position and Movement, and Muscle Force. Reviews 2012, 92, 1651-1697.
- Auzias, M., de Ajuriaguerra, J. Les fonctions culturelles de l’écriture et les conditions de son développement chez l’enfant [The cultural functions of writing and the conditions for its development in children]. Enfance 1986, 39, 145-167.
- Assaiante, C. Action and representation of action during childhood and adolescence: A functional approach. Neurophysiol. 2012, 42, 43-51.
- Assaiante, C., Schmitz, C. Construction des représentations de l’action chez l’enfant : Quelles atteintes dans l’autisme ? [Construction of representations of action in children: What impairment in autism?] Enfance 2009, 1, 111-120.
- Bara, F., Gentaz, E. Haptics in teaching handwriting: The role of perceptual and visuo-motor skills. Mov. Sci. 2011, 30, 745-759.
- Cornhill, H., Case-Smith, J. Factors that relate to good and poor handwriting. J. Occ. Ther. 1996, 50, 732-739.
- Tseng, M.H., Chow, S.M. Perceptual-motor function of school-age children with slow handwriting speed. J. Occ. Ther. 2000, 54, 83–88.
- Weintraub, N., Graham, S. The contribution of gender, orthographic, finger function, and visual–motor processes to the prediction of handwriting status. Ther. J. Res. 2000, 20, 121-141.
- Williams, J., Zolten, A.J., Rickert, V.I., Spence, G.T., Ashcraft, E.W. Use of non verbal tests to screen for writing dysfluency in school-age children. Mot. Ski. 1993, 76, 803-809.
- Yochman, A., Parush, S. Differences in Hebrew handwriting skills between Israeli children in second and third grade. Occup. Ther. Ped. 1998, 18, 53-65.
- Levine, M.D. Developmental variation and learning disorders. Educators Publishing Service: MA, 1987.
- Schneck, C.M. Comparison of pencil-grip patterns in first graders with good and poor writing skills. J. Occ. Ther. 1991, 45, 701–706.
- Hong, S.Y., Jung, N.H., Kim, K.M. The correlation between proprioception and handwriting legibility in children. Phys. Ther. Sci. 2016, 28, 2849-2851.
- Richards, T.L., Berninger, V.W., Fayol, M. FMRI activation differences between 11-years-old good and poor spellers’ access in working memory to temporary and long-term orthographic representations. Neurolinguistics 2009, 22(4), 327-353.
- Van Hoorn, J.F., Maathuis, C.G.B., Hadders-Algra, M. Neural correlates of paediatric dysgraphia. Med. Child Neurol. 2013, 55, 65-68.

Reviewer 2 Report
Statistical values of Kruskall-Wallis must be presented. Bonferroni correction must be considered, and, if adequate, post-hoc Mann-Withney test must be done, with effect size estimation (mandatory).
Additionally, if consideration are made relative to a statistical treatment, it must be presented (mandatory). ("We performed a Bartlett's sphericity test and then a factorial correspondence analysis. We have added this information in the "Statistical analysis". The results are not presented in this document so as not to weigh down the results, but they are presented in the referenced article (reference 17)."- If it was presented in another article, does it mean that authors have already published this study or part of it elsewhere?)
Discussion must be reviewed in accordance.
Author Response
Response to the reviewer 2
" Exploratory investigation of handwriting disorders in school-aged children from 1st to 5th grade"
Statistical values of Kruskall-Wallis must be presented. Bonferroni correction must be considered, and, if adequate, post-hoc Mann-Withney test must be done, with effect size estimation (mandatory).
Thank so much for your insistence about this comment. After a further analysis of our statistics, we realized that the statistician provided some confusion in the presentation of our statistical methodology. The comparison test used was a pearson chi-square test. We present the adapted statistical methodology below and modified the article accordingly. What is important is that there is no impact on the results and interpretation.
"The statistical analysis were carried out on R software (version 3.5.3). The degree of significance retained for all assignments was set at 0.05. Qualitative variables are described by numbers and percentages. A total of 71 binary variables (clinical variables) or tasks were considered. Tasks were scored by the psychomotor therapist 0 (success) and 1 (failure) based on percentile or standard deviation (below 1 SD or 10th percentile, depending on the test) in accordance with standardized instructions and developmental norms. For assessments of developmental features of handwriting, we used the developmental standards published in a previous study [14]. In order to compare the frequency of failure of clinical variables between the different levels of handwriting disorder (HD not detected by BHK scale, moderate HD, dysgraphia), a Pearson chi-square test was performed."
As these variables are not ordinal, we have not used the Sperman correlation test. A Fisher test on contingency tables (or a Yates correction) could be envisaged, but the number of children in each group is greater than five.
However, even if we apply a Fisher test on contingency tables, the results remain statistically significant:
- disorder of visuo-perceptual capacities: p=0.02192,
- disorder of balance: p=0.02138,
- bodily spatial integration disorder: p=0.0217,
- tendency to pessimism: p=0.04589.
We added the results of Pearson chi-square test as follows:
"The percentage of failures in clinical functions was significantly different depending on the level of handwriting disorder underwent by BHK for only 4 clinical features. the. Thus, the more pronounced the writing disorder, the more frequent the disorder of visuo-perceptual capacities (χ2(2)=7.51, p=0.016 [95% CI, 26%-64%]) and the disorder of balance (χ2(2)= 8.17, p=0.016 [95% CI, 12%-47%]) in the population, and more particularly in the "dysgraphia" group. Bodily spatial integration disorder is absent in the "HD not detected by BHK" group and predominant in the "moderate HD" group (χ2(2)=7.88, p=0.019 [95% CI, 17%-54%]). The tendency to pessimism is strongly present in the "HD not detected by BHK" group and decreases with increasing level of HD (χ2(2)=6.00, p=0.046 [95% CI, 5%-35%])."
Additionally, if consideration are made relative to a statistical treatment, it must be presented (mandatory). ("We performed a Bartlett's sphericity test and then a factorial correspondence analysis. We have added this information in the "Statistical analysis". The results are not presented in this document so as not to weigh down the results, but they are presented in the referenced article (reference 17)."- If it was presented in another article, does it mean that authors have already published this study or part of it elsewhere?) Discussion must be reviewed in accordance.
Thank you for this comment. This study has not been published in another article. On the other hand, the factorial analysis mentioned is actually published in another article (the one to which we refer).
For greater precision, we have removed this analysis from the "Statistical analyses" paragraph. We have also made it clearer in the discussion that these are the results of another study.
The following changes have been made:
"Our whole sample of 27 children with HD included in the present study underwent a complete developmental battery of neuropsychomotor, neuropsychological and oculomotor assessments. The aim of this transdisciplinary study was to better understand the complexity of the semiology of HD because the literature is poor. The present study is an important state of handwriting disorders, as to our knowledge, no research has explored such a broad set of skills to better understand the etiology of HD. Our results underline a high heterogeneity of the children presenting a HD and allows us to identify etiological hypotheses. A previous study highlighted the co-occurrence of difficulties associated with handwriting disorders in children [16]. However, the factor analysis carried out in this same study could only explain 28% of the variance in the sample, which is consistent with the heterogeneity of handwriting disorders highlighted in this article."
Please find below the corrections inside the article:
Article
Exploratory investigation of handwriting disorders in school-aged children from 1st to 5th grade
Abstract: Handwriting disorders (HDs) are prevalent in school-aged children with significant interference with academic performances. The current study offers a transdisciplinary approach with the use of normed and standardized clinical assessments of neuropsychomotor, neuropsychological and oculomotor functions. The aim is to provide objective data for a better understanding of the nature and the etiology of HDs. Data from these clinical assessments were analysed for 27 school-aged children with HD (1st to 5th grade). The results underline a high heterogeneity of the children presenting HDs with many co-occurrences often unknown. However, it was possible to highlight three levels of HDs based on BHK scores: mild HD not detected by the BHK test (26% of children), moderate HD (33%), dysgraphia (41% of children). The mild nature of HDs not detected by BHK seems to co-occurrences than children with dysgraphia but more difficulties than children with milder HDs. This highlights the importance of differentiating between different degrees of HDs that do not respond to the same semiologies. Our findings support the interest of performing a transdisciplinary and standardized clinical examination with developmental standards (neuropsychomotor, neuropsychological and oculomotor) in children with HD. Indeed, HDs can therefore be associated with a multitude of disorders of different nature ranging from poor coordination of the graphomotor gesture to a more general and more complex impairment affecting perceptual-motor, cognitive and/or psycho-affective functions.
Keywords: handwriting disorders; dysgraphia; children; semiology; neuropsychomotor assessment; neuropsychological assessment; oculomotricity
1. Introduction
As children spend 31-60% of their school day writing and performing other fine motor tasks [1], the development of handwriting skills is necessary for academic success [2,3] and the proper development of self-esteem [4,5]. According to the old version of the Diagnostic and Statistical Manual of Mental Disorders (DSM-IV-TR [6]), handwriting disorders (HDs) can be diagnosed in the case of "writing skills significantly lower than expected given chronological age, of measured intelligence and of an appropriate education”. The DSM-5 [7] described HDs as “Impairment in written expression” and dysgraphia is not described in the DSM-5. However, handwriting disorders (HD) affect between 10 and 30 % of school-aged children [8-10]. This is observed both by health professionals in clinical consultations and by teachers who struggle to adapt to the differences in the individual rhythms of these children and their learning difficulties. In this context, current studies on handwriting and its disorders attempt to provide new fundamental knowledge on the different processes involved in the development of handwriting, while clinical and therapeutic aspects remain little explored. Thus, handwriting disorder appears as an umbrella term defining a heterogeneous class of children exhibiting graphic impairments. The study of these disorders is complex as their understanding, both on the semiological level and on the etiological level, is still in the literature only in its early stages, and the definition of dysgraphia is unclear. Since the 1960s, it has been characterized by poor writing quality without any neurological or intellectual disorder being able to explain it [11]. This definition has been clarified by other authors who define dysgraphia as a disorder in written language partly linked to a lack of fine motor control in the execution of motor programs [12,13]. Recently, a relevant study [14] has shown phenotyping features in the genesis of pre-scriptural gestures in children to assess handwriting developmental levels because no recent research has previously think to study the developmental prerequisites of the handwriting organization. The better the quality of the handwriting gesture, the less variation there is in the inter-segmental organization coordinated during the writing task. This makes it possible to assess handwriting development levels in the context of screening for handwriting disorders [14]. Hence, another study was able to demonstrate for the first time the immaturity of the graphomotor gesture in children with a handwriting disorder, characterized both by a lack of synergistic coordination of the different segments of the writing arm and by an impairment of the temporal and kinematic characteristics of prescriptural traces (decrease in fluidity characterized by an increase in the number of strokes and velocity peaks, and an increase in drawing time and in-air pauses) [15,16]. The results about the impairment of the temporal and kinematic characteristics of handwriting are also corroborated by Asselborn et al. [17]). Moreover, generally, the authors highlight a lack in motor programming or in motor execution. Wann [18] suggests a motor programming defect characterized by altered temporal organization of writing (dysfunction, high pause times) due to the child's over-reliance on visual feedback. Lopez & Vaivre-Douret [19] suggest both proprioceptive/kinesthetic feedback deficits and a disruptive effect of visual control on the quality of pre-script drawings in these children, many of whom have kinaesthetic memory and visuo-spatial deficits. Thus, the ability to direct strokes would remain dependent on sensory feedback, itself insufficiently effective, leading to difficulties in achieving proactive control of handwriting. Other authors [20-22] suppose an impairment of the motor execution processes, which is characterized by a spatial, temporal, and kinematic irregularity of the writing characteristic in dysgraphic children. This would be the consequence of excessive neuromotor noises [21,23]. However, only one study has proposed a transdisciplinary investigation of handwriting disorders (HDs) [16]. The results highlighted a typology of three groups of HDs (mild ; mild to moderate; dysgraphia), each being associated with co-occurrences of specific neurodevelopmental dysfunctions: a co-occurrence of psycho-affective disorders that can be considered as a predictor of mild and moderate HDs; a co-occurrence of tone disorders and gross coordination that can be considered as a predictor of mild HDs; a co-occurrence of visual-spatial/constructive and attentional disorders which can be considered as a predictor of the most severe (dysgraphia) and moderate HDs. More specifically, a recent study proposed a transdisciplinary investigation of HDs in a cohort of children with a Developmental Coordination Disorder (DCD) [24]. This highlighted a significant association between neurological soft signs and the presence of dysgraphia in a sample of 65 children with DCD [24]. The dysgraphia appeared to be closely related to several specific dysfunctions of the laterality, to a minor neurological dysfunction of the pyramidal tract manifested by a distal phasic stretch reflex in the lower limb, and to a slowness in digital praxia. In addition to these few studies, it is important to enrich the literature concerning the analysis of the underlying clinical functions involved in handwriting disorders.
In the present study, we offer a transdisciplinary approach in depth with the use of normed and standardized clinical assessments of neuropsychomotor, neuropsychological and oculomotor functions. We aimed to provide objective data to better understand the nature and etiology of HDs.
2. Material and Methods
2.1. Participants
Data from a sample of 27 children with handwriting disorders (HDs) aged 6 years 2 months to 10 years 11 months (mean 8.15 SD 1.51) were collected from primary schools (grades 1 to 5) and in the usual out-patient consultation of Pediatrics, Cochin Port-Royal Hospital, and of Child Psychiatry department, Necker University Hospital, in Paris, France. Children were excluded from the study in case of prematurity (birth <37 WA), sensory, visual, neurological or genetic disorders, dyslexia and severe language disorder, ADHD (according to the DSM-5 criteria [7]), autism spectrum disorder, psychopathology, or motor disorder caused by injury or accident. None of them had repeated or skipped a grade or undergone any handwriting retraining at the time of the study. The institutional research ethics committee of Paris Descartes University approved the study procedures (CER·2018-72) conducted in accordance with the Declaration of Helsinki. All parents/legal guardians of participants provided written informed consent.
2.2. Design and Measures
Handwriting disorders were detected by the teachers and considered to be objective by an analysis of their class exercise books by an experienced psychomotor therapist. In order to assess their handwriting level, each child began to underwent the French adaptation of the standardized assessment of handwriting, the BHK scale [25] adapted from the Concise Evaluation Scale for children’s handwriting [26]. Fig 1 shows the study design previously published with the permission of the editor to reproduce it [15]. Then, a neuropsychomotor assesment (NP-MOT) was administrated and folowed by other tests (psychomotor, neurovisual, neuropsychological) proposed in different orders according to each child's motivation and time constraints. All the assesments were administrated on a single day, with breaks with a total around six hours of testing. The examination of oculomotor functions was recorded about twenty minutes at a second appointment on a different day. The psychomotor therapist investigator in the study administered all the tests.
Figure 1. Study design.
2.3. Handwriting assessment.
In addition to the BHK test, the children performed a previously validated cycloid loop line copying test[14,15]. Data on postural organization and inter-segmental coordination of the writing arm were systematically collected by video recording as described in previous studies [14,15]. Features about the proximal (head, trunk axis, shoulder, elbow, and forearm) and distal (wrist and fingers) gestural organization of the drawing process are collected. Variables relating the material (sheet, drawing line, pen) positioning and observational clinical variables related to the semiology of the motor characteristics of the gesture (control, pressure, synkinesis). In addition, spatio-temporal and kinematic measures were recorded using an Anoto digital pen with Elian Research software (Version 4.2, http://www.seldage.com, accessed on 24 September 2022) for which we have developed specific algorithms to record the measures above.
2.4. Clinical assessments
2.4.1. Neuropsychomotor assessment
All children performed neuropsychomotor physical tasks with the NP-MOT battery [27], including assessment of minor neurological dysfunctions (MND) exploring neurological soft signs like synkinesis (NSS). The age-standardized child assessment using the French NPMOT test battery is applicable to children as young as 4 years old. It has been found to have adequate test-retest reliability and internal consistency. Correlation coefficients of the NP-MOT with the BOTMP [28] range from 0.72 to 0.84, for motor coordination and balance. The NP-MOT battery enables physical assessment of passive/active muscular tone of limbs and axial tone (dangling and extensibility of wrist, shoulder, foot, heel-ear angle, popliteal angles, adductor angles, trunk), highlighting NSS denoting the existence of MND, such as limb pyramidal dysfunction. This is complemented by the assessment of basic motor function, control and regulation in gross motor tasks, gait, balance, coordination, manual dexterity, praxis, gnosopraxis (non-meaningful hand and finger imitation of gestures), digital perception, laterality, bodily spatial integration, rhythmic, and auditory attention tasks. The standardized NP-MOT battery is a developmental assessment because each subtest and milestone is scored from qualitative and quantitative viewpoints according to age, with each score converted to a standard deviation vs. mean, based on normative data for age and applicable to children as young as 4 years old [29]. There is a saturation of the maturation scores between 8 and 10 years, allowing to assess with the NP-MOT older children or adults.
2.4.2. Psychomotor assessment
The MABC-2 children's movement assessment battery (2nd edition) [30], adapted from the American battery [31] was used to assess psychomotor skills. It aims to assess motor impairments and is divided into three categories: manual dexterity (uni-manual, bi-manual test and visual-motor graphic tasks), target and catch (to throw a weighted bag/ball on a target and to catch a weighted bag or a ball), balance (static balance, walking and jumping tasks).
The gnosopraxis imitation of gestures assessment EMG [32,33] was used to assess the distal and digital gnosopraxic efficiency and to measure the child's ideomotor adaptation skills. It consists of performing imitations of arbitrary simple (with the hands) and complex (with the fingers) gestures in the absence of verbal command. This is an adaptation of the Bergès-Lézine assessment [34], paying particular attention to the gesture programming in the notation.
The Body Schema Test – Revised [35] was used to assess the child's representation of his own body and the relationships between different parts of his body. The task consists of a puzzle (non-contiguous pieces) of the body and face from the front (for children aged 3 to 8) and/or from a side view (for children aged 8 and over).
Spatial and temporal identification questions were asked in order to assess the knowledge and mastery of the spatio-temporal vocabulary.
A kinesthetic memorization test consisting of a reproduction test of asymbolic postures which had previously been printed and felt with the eyes closed, has been proposed in order to assess the body's perceptual skills [24].
2.4.3. Neurovisual assessment
Neurovisual aspects including visual gnosias, visual-perceptual, perceptual visual-motor, visuospatial, visuo-constructive skills and oculomotricity were assessed.
Visual perception was assessed using form recognition tasks [36], tangled lines and visual gnosia with outlines of animals, outlines of muddled fruits.
The KABC-II Shape Recognition subtest [37] consists in recognizing and naming drawings of various objects whose image has been altered (some lines of the drawing appear while others have been erased). This item assesses the child's ability to mentally represent the missing parts of the drawing to form a complete mental image, making it possible to name the represented object.
The Developmental Test of Visual-Motor Integration (VMI) (6th ed) [38] assessed pure perceptual abilities (the perceptual subtest of the test consisting of visual recognition of identical insignificant geometric shapes) and visuomotor integration abilities (subtest copy of the test figure consisting of the reproduction of simple and more complex insignificant geometric figures).
The NEPSY-II Arrows subtest [39] consisted of judging the direction, orientation and angles of different lines.
Rey’s complex geometric figure [40] allows the evaluation of aptitudes for perception, structuring and spatial organization (visual-spatial and visuo-constructive praxies). By copying and then reproducing from memory a complex geometric figure, it studies the ability to structure different elements in a graphic space.
The Code and Symbols subtests of the Wechsler intelligence scale for children and adolescents WISC-IV [41] for measuring a mental processing speed index (IVT) in connection with graphomotor capacities consist in analyzing and distinguishing non-significant signs.
The NEPSY-II Cubes subtest [39] and the Kohs block design [42] respectively assessed 3D visual-constructive skills and visual-spatial/constructive skills.
The examination of oculomotor functions was performed using an eye-tracking device made up of two infrared cameras positioned at the level of the inner corner of each eye (Ober consulting eye-Tracker Eyefant® [48]) recording the movements of fixation, smooth visual pursuit, and horizontal and vertical eye saccades. The device records in the horizontal and vertical planes at a sampling rate of 1000 Hz, a spatial resolution of 0.1 ° and a linearity range of +/- 35 ° horizontally and +/- 20 ° vertically.
2.4.4. Neuropsychological assessment
Visual-spatial attention, sustained auditory attention, and divided attention skills were assessed by a crossing test and the Childhood Attention Assessment Test (TEA-Ch [43]).
Executive functions (planning, inhibition skills, mental flexibility, working memory, and verbal fluency) were assessed by the Laby 5-12 labyrinths test [44], the NEPSY-II Categorization and Verbal Fluency subtests [42] and the Stroop's Selective Attention Test [45].
The visual, auditory and working memory skills were assessed by the Face Memory subtest of the NEPSY-II and by a face-up and back-up number span test (Odedys [46]).
The MDI-C Composite Childhood Depression Scale [47] assessed the emotional state of the child in order to identify a possible depressive state, this through 8 dimensions: self-esteem, anxiety, sad mood, social introversion, pessimism, mistrust, low energy and feelings of helplessness.
2.4.5. Language skills
The regular, irregular and pseudo-word reading test from the Odedys DYSlexia Screening Tool [46] assessed reading level and allowed to ruling out a diagnosis of dyslexia.
2.5. Statistical analyses
The statistical analysis were carried out on R software (version 3.5.3). The degree of significance retained for all assignments was set at 0.05. Qualitative variables are described by numbers and percentages. A total of 71 binary variables (clinical variables) or tasks were considered. Tasks were scored by the psychomotor therapist 0 (success) and 1 (failure) based on percentile or standard deviation (below 1 SD or 10th percentile, depending on the test) in accordance with standardized instructions and developmental norms. For assessments of developmental features of handwriting, we used the developmental standards published in a previous study [14]. In order to compare the frequency of failure of clinical variables between the different levels of handwriting disorder (HD not detected by BHK scale, moderate HD, dysgraphia), a Pearson chi-square test was performed.
3. Results
3.1. Characteristics of the sample
Twenty-seven children with handwriting disorders were included in this study, 4 girls (15%) and 23 boys (85%), aged 6 years 2 months to 10 years 11 months (mean 8.15 SD 1.51). Eleven of them (41%) presented dysgraphia on the BHK scale and nine (33%) presented more moderate handwriting disorders. In contrast, seven (26%) were not identified by the BHK test as presenting any handwriting disorder (see Table 1). Among the 27 children, six (22%) presented developmental coordination disorder (DCD) according to the DSM-5 criteria, and two (7%) had high intellectual potential (>130 IQ).
Table 1. Characteristics of the children.
|
|
Handwriting disorder not identified by the BHK test (n=7) |
Moderate handwriting disorder (n=9) |
Dysgraphia (n=11) |
Total (n=27) |
|
Age (years) [m (SD)] |
8.30 (0.81) |
7.79 (1.53) |
8.36 (1.87) |
8.15 (1.51) |
|
Gender [n(F/M)] |
0/7 |
1/8 |
3/8 |
4/23 |
n: number; m: mean; SD: standard deviation; F: female; M: male.
3.2. Results of the handwriting assessment
The detailed results of the sample about the postural and gestural organization and their spatial, temporal, and kinematic features of the drawings were described in a previous study [16]. Children with handwriting disorders have poor synergistic coordination of the handwriting arm characterised by the persistence, whatever the age, of a progression along the line, consisting in moving the forearm and the elbow rather than a more mature rotation movement of the forearm at the elbow. They have also an instability of the wrist and a slow and hyper-controlled hand gesture. Alongside, the drawing is characterised by poor quality, lack of fluidity, and slowness.
3.3. Percentage of clinical test failures (neuropsychomotor, psychomotor, neurovisual, neuropsychological, oculomotor, and language assessements)
Fig 2 presents the percentage of failures in neuropsychomotor and neuropsychological functions assessed by standardized clinical tests and of oculomotricity disorders identified during the examination of oculomotor functions. The variables are ordered by decreasing frequency of failure.
Figure 2. Percentage of failures in clinical functions assessed in the whole group.
Thus, the descriptive analysis of the results of the clinical tests highlights a set of quite varied disorders such as tone disorder, visual-motor graphic disorder, oculomotor disorder, lack of kinesthetic memory, disturbance of visual perceptual functions, and disorder of executive functions.
3.4. Frequency of failures in clinical assessments between the different levels of handwriting disorder (HD not detected by BHK scale, moderate HD, dysgraphia)
A more precise typology of HD was demonstrated by analyzing the distribution of failures in clinical functions according to the degree of writing disorder revealed by the BHK (see Table 2).
Table 2. Percentage of failures in clinical functions assessed for each of the groups classified by the BHK test.
|
Fonctions |
Whole HD group (n=27) |
HD not detected by BHK scale (n=7) |
Moderate HD on BHK (n=9) |
Dysgraphia on BHK (n=11) |
p-value |
|
Tone disorder (NP-MOT) |
74 |
71 |
78 |
73 |
0.95 |
|
Heel-ear angle reduction |
59 |
71 |
56 |
55 |
0.76 |
|
Affirmation of tonic laterality (NP-MOT) |
67 |
43 |
67 |
82 |
0.24 |
|
Visual memory (Rey, NEPSY-II) |
63 |
43 |
78 |
64 |
0.86 |
|
Hand-eye coordination (NP-MOT, MABC-2) |
56 |
43 |
67 |
55 |
0.64 |
|
Manual dexterity (NP-MOT , MABC-2) |
56 |
43 |
67 |
55 |
0.64 |
|
Graphic visual-spatial coordination |
78 |
71 |
78 |
82 |
0.88 |
|
One-hand coordination |
37 |
29 |
33 |
45 |
0.75 |
|
Bimanual coordination |
30 |
14 |
44 |
27 |
0.43 |
|
Perceptual visual-motor skills |
52 |
14 |
67 |
64 |
0.076 |
|
Codes test (WISC-IV) |
41 |
14 |
44 |
55 |
0.24 |
|
Copying figures (VMI) |
22 |
0 |
22 |
36 |
0.21 |
|
Symbols test (WISC-IV) |
15 |
0 |
11 |
27 |
0.28 |
|
Visual-spatial organization (Rey) |
7 |
0 |
11 |
9 |
0.69 |
|
Kinesthetic memory |
52 |
29 |
67 |
55 |
0.32 |
|
Oculomotricity |
44 |
29 |
44 |
55 |
0.26 |
|
Smooth pursuits |
37 |
14 |
33 |
55 |
0.16 |
|
Saccadic aye movement |
26 |
29 |
11 |
36 |
0.25 |
|
Fixation |
19 |
0 |
33 |
18 |
0.22 |
|
Visual-perceptual skills |
44 |
14 |
33 |
73 |
0.016 * |
|
Visual gnosia |
26 |
14 |
33 |
27 |
0.69 |
|
Perceptual (VMI) |
19 |
0 |
22 |
27 |
0.34 |
|
Mishmash of lines |
15 |
0 |
11 |
27 |
0.28 |
|
Pattern recognition |
15 |
0 |
22 |
18 |
0.44 |
|
Positions in space (Frostig) |
7 |
0 |
0 |
18 |
0.22 |
|
Exécutive functions |
41 |
29 |
33 |
55 |
0.49 |
|
Planning |
26 |
29 |
22 |
27 |
0.95 |
|
Categorization (Nepsy-II) |
15 |
14 |
0 |
27 |
0.25 |
|
Inhibition (Laby 5-12, Stroop) |
7 |
14 |
11 |
0 |
0.48 |
|
Verbal fluency (Nepsy-II) |
4 |
0 |
0 |
9 |
0.48 |
|
Auditory-attentional skills (TEA-Ch) |
41 |
29 |
33 |
55 |
0.49 |
|
Visual-attentional skills (TEA-Ch) |
41 |
43 |
56 |
27 |
0.45 |
|
Low énergy (MDI-C) |
41 |
43 |
44 |
36 |
0.93 |
|
Social introversion (MDI-C) |
37 |
14 |
22 |
64 |
0.06 |
|
Coordination between upper and lower limbs (NP-MOT) |
37 |
43 |
33 |
36 |
0.93 |
|
Dynamic balance coordination (NP-MOT) |
37 |
43 |
33 |
36 |
0.93 |
|
Static balance (NP-MOT) |
37 |
29 |
11 |
64 |
0.052 |
|
Aim and catch (MABC-2) |
37 |
14 |
22 |
64 |
0.06 |
|
Reading (Odedys) |
33 |
43 |
33 |
27 |
0.80 |
|
Time tracking (NP-MOT) |
33 |
0 |
33 |
55 |
0.06 |
|
Bodily spatial integration (NP-MOT) |
33 |
0 |
67 |
27 |
0.019 * |
|
Bimanual praxia (NP-MOT) |
33 |
0 |
44 |
45 |
0.10 |
|
Divided attention skills (TEA-Ch) |
30 |
29 |
22 |
36 |
0.79 |
|
Affirmation of manual laterality (NP-MOT) |
30 |
0 |
56 |
27 |
0.053 |
|
Neurological soft signs (NP-MOT) |
30 |
0 |
33 |
45 |
0.12 |
|
Imitation of fingers gestures (EMG) |
26 |
14 |
22 |
36 |
0.57 |
|
Balance (MABC-2) |
26 |
0 |
11 |
55 |
0.016 * |
|
Dressing praxia |
22 |
14 |
11 |
36 |
0.35 |
|
Anxiety (MDI-C) |
22 |
14 |
33 |
18 |
0.62 |
|
Feeling of helplessness (MDI-C) |
22 |
14 |
11 |
36 |
0.35 |
|
Dépressive disorder (MDI-C) |
19 |
14 |
22 |
18 |
0.92 |
|
Working memory (Odedys) |
19 |
14 |
22 |
18 |
0.92 |
|
usual discordant laterality (NP-MOT) |
19 |
0 |
33 |
18 |
0.25 |
|
Auditory memory (Odedys) |
15 |
0 |
11 |
27 |
0.28 |
|
Visual-constructive skills |
15 |
0 |
11 |
27 |
0.28 |
|
3D constructions (Nepsy-II) |
7 |
0 |
0 |
18 |
0.22 |
|
2D constructions (Kohs) |
7 |
0 |
11 |
9 |
0.69 |
|
Digital gnosia (NP-MOT) |
15 |
0 |
22 |
18 |
0.44 |
|
Oral-facial praxia |
15 |
0 |
11 |
27 |
0.29 |
|
Sad mood (MDI-C) |
15 |
29 |
22 |
0 |
0.20 |
|
Pessimism (MDI-C) |
15 |
43 |
11 |
0 |
0.046 * |
|
Body image (CORP-R) |
11 |
0 |
0 |
27 |
0.09 |
|
Rhythmic adaptation (NP-MOT) |
11 |
0 |
0 |
27 |
0.09 |
|
Digital praxia (NP-MOT) |
11 |
0 |
11 |
18 |
0.50 |
|
Spasticity (NP-MOT) |
11 |
0 |
22 |
9 |
0.37 |
|
Discordant tonic latérality (NP-MOT) |
7 |
0 |
11 |
9 |
0.69 |
|
Visual-spatial perception (Nepsy-II) |
7 |
14 |
11 |
0 |
0.48 |
|
Distrust (MDI-C) |
7 |
0 |
11 |
9 |
0.69 |
|
Imitation of hands gestures (EMG) |
4 |
0 |
0 |
9 |
0.48 |
|
Self estim (MDI-C) |
4 |
14 |
0 |
0 |
0.24 |
|
Non symbolic organisation of the gesture (NP-MOT) |
0 |
0 |
0 |
0 |
na |
The percentage of failures in clinical functions was significantly different depending on the level of handwriting disorder underwent by BHK for only 4 clinical features. the. Thus, the more pronounced the writing disorder, the more frequent the disorder of visuo-perceptual capacities (χ2(2)=7.51, p=0.016 [95% CI, 26%-64%])and the disorder of balance (χ2(2)= 8.17, p=0.016 [95% CI, 12%-47%]) in the population, and more particularly in the "dysgraphia" group. Bodily spatial integration disorder is absent in the "HD not detected by BHK" group and predominant in the "moderate HD" group (χ2(2)=7.88, p=0.019 [95% CI, 17%-54%]). The tendency to pessimism is strongly present in the "HD not detected by BHK" group and decreases with increasing level of HD (χ2(2)=6.00, p=0.046 [95% CI, 5%-35%]).
A trend close to statistical significance is observed for 8 clinical features (p <0.10) with a higher frequency of 6 of them in the "dysgraphia" group and of 2 of them in the "moderate HD" group. Their frequency is never the highest in the "HD not detected by BHK" group. Thus, when the level of HD increases on the one hand, so do static balance disorders, aiming and catching difficulties, temporal identification, body diagram, rhythmic adaptation disorders, and social introversion. These disabilities are particularly common in the "dysgraphia" group. Disorders of visuomotor perceptual capacities and affirmation of manual laterality are respectively the least frequent (14%) and absent in the "HD not detected by BHK" group. They are in the majority in the "moderate HD" group.
The descriptive analysis of the whole clinical features reveals a higher proportion of co-occurrences in the "dysgraphia" group than in the "moderate HD" group, whereas the children identified as having a " HD not detected by BHK "appear less affected”.
Thus, when the level of HD increases, there is a greater proportion of neuro-psychomotor, neuropsychological and oculomotricity disorders. On the other hand, psycho-affective disorders such as a sad mood, a tendency to pessimism, and low self-esteem appear in the majority of children with a HD not detected by BHK. Psycho-affective disorders are therefore a possible origin of the less pronounced writing disorders.
3.5. Failures greater than 40% in each of the three groups identified by the BHK test (HD not detected by BHK, moderate HD, dysgraphia)
The analysis of Table 2 shows the following results.
In the "HD not detected by BHK" group, the tone disorder (reduction in joint angles measured during passive tone examination), the heel-ear angle reduction, and the graphic visual-spatial coordination disorder appear at a frequency greater than 50%. However, there is no significant difference between groups for these clinical variables. This is explained by the presence of these difficulties in the clinical group as a whole, irrespective of the level of HD. The disorders of the coordination between the upper and lower limbs, of the coordination of static balance and a slowness of reading appear at a frequency greater than 40% but not significantly. Because they also appear to be highly unsuccessful throughout the whole clinical sample. The tendency to pessimism appears at a frequency greater than 40% in the "HD not detected by BHK" group with a difference between groups close to significance (p=0.07). In addition, a factor analysis revealed a co-occurrence of psycho-affective disorders (depression, lack of self-esteem, sad mood, feeling of helplessness, pessimism, low energy, anxiety) associated with the "HD not detected by BHK" group.
In the "moderate HD" group, tone disorder, heel-ear angle reduction, poorly asserted tonic laterality, poorly asserted manual laterality, visual memory disorder, manual and oculo-manual disorders, graphic visual-spatial coordination disorder, perceptual visual-motor disorder, kinesthetic memory disorder, visual-attentional disorder, and bodily spatial integration disorder appear at a frequency greater than 50%. Among these disabilities, only a trouble of spatial integration of the body appears to be significantly more frequent in the "moderate HD" group (p=0.03). Once again this seems to be because the variables appear to be mostly unsuccessful in the whole clinical group. Bimanual coordination and praxia disorders, low energy, oculomotricity disorder and Codes test failure occur at a frequency greater than 40% but not specifically in the group "moderate HD".
In the "Dysgraphia" group, appear at a frequency greater than 50% : a tone disorder with a heel-ear angle reduction, a poorly asserted tonic laterality, disorders of visual memory, manual and oculo-manual skills, visual-motor visual coordination, perceptual visuo-motor capacities (in particular with the test of the WISC-IV Codes), kinesthetic memory, oculomotricity and especially in smooth pursuits, disturbances of visuo-perceptual capacities, of executive functions, of auditory-attentional capacities, of static balance and of the capacities to aim and catch, and a disorder of temporal identification. Among these disorders, the static balance disorders identified with MABC-2 and NP-MOT is significantly more frequent in the "dysgraphia" group (respectively p=0.01 and p=0.049), as well as the disorder of aiming and catching abilities (p=0.049) and visual-perceptual skills disorder (p=0.04). Neurological soft signs (synkinesis), a disorder of uni-manual and bimanual coordinations appear at a frequency greater than 40% and are in the majority in the "Dysgraphia" group, but they are not specific to this group. In addition, a factor analysis revealed a co-occurrence of visual-spatial/constructive and attentional disorders related to an oculomotor disorder (visual fixation) and associated with the "dysgraphia" group.
4. Discussion
Our whole sample of 27 children with HD included in the present study underwent a complete developmental battery of neuropsychomotor, neuropsychological and oculomotor assessments. The aim of this transdisciplinary study was to better understand the complexity of the semiology of HD because the literature is poor. The present study is an important state of handwriting disorders, as to our knowledge, no research has explored such a broad set of skills to better understand the etiology of HD. Our results underline a high heterogeneity of the children presenting a HD and allows us to identify etiological hypotheses. A previous study highlighted the co-occurrence of difficulties associated with handwriting disorders in children [16]. However, the factor analysis carried out in this same study could only explain 28% of the variance in the sample, which is consistent with the heterogeneity of handwriting disorders highlighted in this article.
In a previous study about the influence of visual control on the quality of the graphic gesture in children with handwriting disorders [19], we hypothesized the involvement of the cortico-striatal and cortico-cerebellar pathways in HD. The hypothesis of cerebellar dysfunction in children with HDs is accepted in the literature [49]. Our clinical sample being very heterogeneous, other etiological hypotheses can be proposed.
Our findings showed an important percentage of children (26%) exhibiting a HD penalizing them at school and being notable in class notebooks but not detected by the BHK test. This lack of detection of HDs is explained by a slight degradation of the handwriting when copying a paragraph (like proposed in the BHK test). Probably, the dual and evaluative situation induced by a copy with the BHK and not by a spontaneous handwriting leads to a particular concentration and application of these children, who then obtain a sufficient qualitative handwriting. Indeed, they fail to achieve proficient handwriting at school, where handwriting times are longer and where the child is constantly in double-tasking. Moreover, the mild nature of HD in these children seems to occur to a relatively low frequency of the associated disorders identified during clinical assessments. This lower frequency would also lead to lighter consequences at the perceptual, perceptual-motor and motor control levels. Thus, HDs in these children appears mainly associated with a tone disorder characterized by a reduction in joint angles, particularly in the heel-ear angle which may underline an abnormal strengthening of the muscle chain leading to a certain tonicity emphasizing a non-release. This can be the cause of coordination difficulties and poor or limited coordination of the graphomotor gesture. Indeed, in our sample, children with handwriting disorders have poor synergistic coordination of the handwriting arm characterized by the persistence, whatever the age, of a progression along the line, consisting in moving the forearm and the elbow rather than a more mature rotation movement of the forearm at the elbow. They have also an instability of the wrist and a slow and hyper-controlled hand gesture. We can therefore assume that these children should benefit from better flexibility thanks to stretching activities and rehabilitation of coordination, and in particular about the prerequisites of the gestural tracing and segmental coordination of the graphomotor gesture by a psychomotor therapist. The slowness of handwriting in 43% of them can highlight possible deficits in phonological or metaphonological processing and in phoneme-grapheme conversion. Interestingly, the right anterior insula is strongly activated when writing letters, possibly related to phoneme-grapheme conversion [50]. It is logical to think that a child with poor recognition and phonological knowledge of the letter will have difficulty integrating the sensorimotor spatial form of the same letter. This hypothesis is confirmed by neuroimaging studies which show stronger activation of the premotor cortex, parietal cortex, cerebellum, and fusiform gyrus when typical children write an unknown letter (pseudoletter) than when they write a known letter. This is visible even though there is no difference in activation among poor writers [51]. These results support the hypothesis that the spatial shape of known letters is difficult to remember for children with HD. Even though we excluded detected speech impairments from our study, it is possible that some of the children had a phonological disturbance that would not have been detected. In the present study, we did not find any children for whom the HD could be explained exclusively by a depressive disorder. However, it is possible that a HD profile without other comorbidities may also reflect different assumptions about a psycho-affective problem. Our findings allow to assume that depression can be the cause of co-occurring difficulties leading to HD (executive, attentional or memory disorders). This can also be the consequence of HD which can lead to academic difficulties and remarks from adults even when the child is making significant efforts. It therefore seems that psychological care for these children could be useful in helping them improve their self-esteem and well-being, but it is not sufficient to treat the cause of the HD. It is interesting to note that 43% of the children showing a HD not detected by BHK are characterized by low energy (MDI-C). This may reflect the fatigability related to the cost of handwriting for these children. The high proportion of graphic visual-spatial coordination disabilities (71%) may be related to poor coordination of the graphomotor gesture [15], which would disturb the precision of the strokes, or to visuospatial perceptual difficulties. The mental planning disorder identified at Laby 5-12 (29% of children with HD not detected by BHK) is probably related to difficulties in visual-spatial perception and visual-motor coordination. Indeed, the Laby 5-12 requires significant visual-motor coordination skills, the fact of crossing the walls of the labyrinth with the pen being counted in the scoring. The association, in children with HDs not detected by BHK, between motor coordination and visual-motor graphic coordination disorders and slow reading (specifically when reading pseudo-words, which involve phonological skills) may suggest the involvement of the cerebellum. Indeed, several studies highlight a dysfunction at the level of the cerebellum in the comorbidity between DCD and dyslexia [52,53]. This hypothesis is also corroborated by Nicolson et al. [54] who demonstrate different cerebellar activity in dyslexic adults compared to typical adults. In addition, several studies highlight an association between mild disorders of gestural coordination and dyslexia [55,56].
At the same time, a significant proportion of children (41%) are classified by BHK test as having dysgraphia. The more pronounced character of the HD in these children seems to occur to a high frequency of the associated disorders identified during clinical assessments. This higher frequency would also lead to higher consequences at the perceptual, perceptual-motor, and motor control levels. Thus, children with dysgraphia have disabilities in motor coordination, manual skills and praxis, organization impairments of muscle tone with the presence of neurological soft signs, establishment of laterality, temporal identification, memory functions (kinesthetic and visual), visual perceptual functions, visual-motor integration, oculomotricity, auditory-attentional capacities and executive functions. They differ significantly (p <0.05) or almost significantly (p <0.10) from children presenting a less pronounced HD. We assume in these children that the oculomotor disorder (55%) may be the cause of visual-perceptual disorders and of the static balance disorder noted in 64% of children. Indeed, vision has a proprioceptive function and participates in tonico-postural regulation [57,58]. The impairment of oculomotor functions in our sample of children suggests a possible delay in the maturation of the oculomotor system, which notably involves the cerebello-cortical and cerebellar networks [59,60]. Furthermore, the visual-perceptual difficulties noted corroborate the studies in neuroimaging, which for the most part highlight an involvement of the ventral occipito-temporal cortex in writing [61], a structure involved in visual perception. The difficulties of temporal regulation and rhythmic adaptation that are only noted in children with dysgraphia support the hypothesis of a specific dysfunction of the cortical-subcortical pathways, which involve the cortical structures, the basal ganglia, and the thalamus. Indeed, these pathways would be involved in the motor adaptation skills and the learning of gestural sequences [62] in the temporal regularity of writing [63]. This again signals the importance of differentiating the diagnosis of dysgraphia from that of a less severe HD because dysgraphia is a neurodevelopmental disorder for which handwriting is really difficult or impossible. Thus, it is not acceptable today to put all HDs under the same umbrella, the remediation should be different according to the HD. The preponderance of oculomotricity disorder supporting the hypothesis of dysfunction of the cerebellum basal ganglia and superior colliculus structures being involved in oculomotor control [64] and sensorimotor functions (involving the cortico-subcortical pathways) in dysgraphic children highlights the importance of a transdisciplinary assessment of HDs. It is important that children identified as dysgraphic could undergo a complete visual and neurovisual assessment including oculomotricity. The body image disorder identified more specifically in dysgraphic children is combined with difficulties in integrating an internal representation of body segments in motion. This co-occurrence could be the result of sensory integration difficulties, especially on the proprioceptive level [65-67]. Our results attest to a multiplicity of functional impairments in children who are effectively dysgraphic, highlighting the need for a transdisciplinary panel of assessments, both of the graphic gesture, and at the neuropsychomotor, neuropsychological and oculomotor level. Depending on the situation, these children will need rehabilitation in psychomotricity, orthoptics, neuropsychology, or an occupational therapy to learn to use a computer in the classroom to compensate for his HD. It is also important to identify potential language disorders in these children, who will then need speech therapy.
Finally, 33% of children are classified by BHK test as having a moderate HD. They have fewer co-occurrences than children with dysgraphia but more difficulties than children with milder HDs. This again highlights the importance of differentiating between different degrees of HDs that do not respond to the same semiologies. The significantly higher frequency (p<0.05) of body spatial integration, visuomotor perceptual disorders, and of poorly asserted manual laterality in these children, reinforces the preponderance of the difficulties of sensorimotor integration in HD. Since sensorimotor skills are necessary for the internal representation of action, it is not surprising that impairment of these skills is involved in HD. These results are in line with the empirical data according to which the graphic space appears as the projection of a representation of the body with an up/down, left/right organization separated by an imaginary vertical median axis reference corresponding to the axis of the body [68]. The multi-modal and redundant integration of sensory information participates in the development of the internal sensorimotor representation of movement, which itself participates in the function of anticipation and planning of the action [69,70]. These results imply difficulties in anticipation and motor planning in children with HD who fail to correctly parameterize the spatial and temporal characteristics of the graphic trajectory. In addition, the high proportion of visuomotor perceptual disorders is congruent with the literature which concludes to an implication of visuomotor integration skills in HD [71-76]. The lack of kinesthetic memory in 67% of children with a moderate HD can lead to poor efficiency of the sensory feedbacksnecessary for the proper anticipation and planning of strokes when writing. The high proportion of these difficulties is congruent with the studies showing an influence of the kinesthetic capacities on the grip of the writing tool [77-78] and on the graphic quality [72,79].
5. Conclusions
Thus, our depth clinical examination made it possible to make underlying hypotheses for the involvement of different areas of the brain in HDs. These hypotheses would require further study in brain imaging. Our findings in the present study support the interest of performing a transdisciplinary and standardized clinical examination with developmental standards (neuropsychomotor, neuropsychological and oculomotor) in children with HD. Our results also highlighted the multiplicity of HDs and co-occurrences. This heterogeneity of the disorders is congruent with the neuroimaging studies, which underline the involvement of very large cortical areas as well as the parietal, temporal, frontal, occipital areas, and cerebellum [80,81]. HDs can therefore be associated with a multitude of different disorders ranging from a poor coordination of the graphomotor gesture to a more general and more complex impairment affecting perceptual-motor, cognitive and/or psycho-affective functions. However, it would be interesting to replicate our results with a larger sample of children in order to be able to carry out analyses school class by school class or by smaller age groups.
Author Contributions: C.L. and L.V.-D. conceived the study and designed the methodology; C.L. recruited the participants and collected the data; C.L. and L.V.-D. analyzed the data; C.L. and L.V.-D. performed the interpretation of results; C.L. prepared the tables and figures; C.L. and L.V.-D. wrote the manuscript. All authors have read and agreed to the published version of the manuscript.
Funding: This research was funded Institut Universitaire de France (IUF), Grant n° L17P99-IUF004, Paris, France, and partly by the Fondation de France, with the Fondation pour la Recherche en Psychomotricité et Maladies de Civilisation (FRPMC).
Institutional Review Board Statement: The institutional research ethics committee of Paris Descartes University approved the study procedures (CER·2018-72, date 18 June 2018) conducted in accordance with the Declaration of Helsinki.
Informed Consent Statement: All the parents and the children provided written informed consent. The data collected was anonymised.
Data Availability Statement: The data presented in this study are available on request from the corresponding author.
Acknowledgments: We are grateful to the children and their families who took part in the study. The authors thank Lucie Coussement for her help to check the English writing of the manuscript.
Conflicts of Interest: The authors have stated that they had no interest which might be perceived as posing a conflict or bias.
References
- McHale, K.; Cermak, S.A. Fine motor activities in elementary school: Preliminary findings and provisional implications for children with fine motor problems. J. Occ. Ther. 1992, 46, 898–903.
- Sassoon, R. Handwriting : A New Perspective; Stanley Thornes: Chelthenham, UK, 1990.
- Stewart, S.R. Developement of written language proficiency: Methods for teaching text structure. In Communication Skills and Classroom Success; Simon, C.C., Ed.; Taylor & Francis: United Kingdom, 1992.
- Chang, S.H., Yu, N.Y. Handwriting movement analyses comparing first and second graders with normal or dysgraphic characteristics. Dev. Disabil. 2013, 34, 2433-2441.
- Feder, K.P., Majnemer, A. Handwriting development, competency, and intervention. Med. Child Neurol. 2007, 49, 312-317.
- Diagnostic and Statistical Manual of Mental Disorders, DSM IV-TR, 4th ed.; American Psychiatric Publishing: Arlington, MA, USA, 2000.
- Diagnostic and Statistical Manual of Mental Disorders, DSM-5, 5th ed.; American Psychiatric Publishing: Arlington, MA, USA, 2013.
- Karlsdottir, R., Stefansson, T. Problems in developing functional handwriting. Mot. Ski. 2002, 94, 623-662.
- Maeland, A.E. Handwriting and perceptual motor skills in clumsy, dysgraphic, and normal children. Mot. Ski. 1992, 75, 1207-1217.
- Naider-Steinhart, S., Katz-Leurer, M. Analysis of proximal and distal muscle activity during handwriting tasks. J. Occ. Ther. 2007, 61, 392–398.
- Ajuriaguerra (de), J., Auzias, M., Denner, A. L’écriture de l’enfant. L’évolution de l’écriture et ses difficultés, 4e , Vol. 1 [The child's writing. The evolution of writing and its difficulties, 4e ed., Vol. 1]; Delachaux et Niestlé: Paris, France, 1964.
- Hamstra-Bletz, L., Blöte, A.W. A Longitudinal Study on Dysgraphic Handwriting in Primary School. J Learn Disabil. 1993, 26, 689–699.
- Van Dorn, R.R.A., Keuss, P.J.G. Dysfluency in children’s handwriting. In The Development of Graphic Skills; Wann, J., Wing, A.M., Sovik, N., Eds.; Academic Press; Cambridge, MA, USA, 1991.
- Vaivre-Douret, L., Lopez, C., Dutruel, A., Vaivre, S. Phenotyping features in the genesis of pre-scriptural gestures in children to assess handwriting developmental levels. Sci Rep. 2021, 11, 731.
- Lopez, C., Vaivre-Douret, L. Concurrent and predictive validity of a cycloid loops copy task to assess handwriting disorders in children. Children 2023, 10 (305), 1-14
- Lopez, C., Vaivre-Douret, L. Investigations exploratoires des troubles de l’écriture chez des enfants scolarisés du CP au CM2 [Exploratory investigations of handwriting disorders in children from 1st to 5th grade]. ANAE 2021, 170, 77-89.
- Asselborn, T., Chapatte, M., Dillenbourg, P. Extending the Spectrum of Dysgraphia: A Data Driven Strategy to Estimate Handwriting Quality. Rep.2020, 10, 31-40.
- Wann, J.P. Handwriting disturbance: Developmental trends. In Themes in motor development; Whiting, H.T.A., Wade, M.G., Eds.; Martinus Nijhoff Publishers: Leyde, The Netherlands, 1986.
- Lopez, C., Vaivre-Douret, L. Influence of visual control on the quality of graphic gesture in children with handwriting disorders. Rep. 2021, 11, 23537.
- Smits-Engelsman, B.C., van Galen, G.P. Dysgraphia in Children: Lasting Psychomotor Deficiency or Transient Developmental Delay? Exp. Child Psychol. 1997, 67, 164-184.
- van Galen, G.P., Portier, S.J., Smits-Engelsman, B.C., Schomaker, L.R. Neuromotor noise and poor handwriting in children. Acta Psychol. 1993, 82, 161–178.
- Wann, J.P., Kardirkamanathan, M. Variability in children’s handwriting: Computer diagnosis of writing difficulties. In The Development of Graphic Skills; Wann, J., Wing, A.M., Sovik, N., Eds.; Academic Press: Cambrige, MA, USA, 1991.
- Zesiger, P. Acquisition et troubles de l’écriture [Learning to write and hardwriting disorders]. Enfance 2003, 55, 56-64.
- Lopez, C., Hemimou, C., Golse, B., Vaivre-Douret, L. Developmental dysgraphia is often associated with minor neurological dysfunction in children with developmental coordination disorder (DCD). Neurophysiol. 2018, 48, 207-217.
- Charles, M., Soppelsa, R., Albaret, J.M. Échelle d’évaluation rapide de l’écriture chez l’enfant (BHK) [Concise Evaluation Scale for children’s handwriting (BHK)]. ECPA-Pearson: Paris, 2004.
- Hamstra-Bletz, L., DeBie, J., Den Brinker, B. Concise Evaluation Scale for children’s handwriting. Swets Zeitlinger: Lisse, The Netherlands, 1987.
- Vaivre-Douret, L. Batterie d’évaluation des fonctions neuro-psychomotrices (NP-MOT) [Neuro-psychomotor functions assessment battery]. ECPA-Pearson: Paris, 2006.
- Bruininks, R.H. Bruininks Oseretsky test of motor proficiency. American Guidance Service: Minnesota, 1978.
- Vaivre-Douret, L. Outil numérique de notation et cotation normées de l’évaluation développementale standardisée des fonctions Neuro-Psychomotrices de l’enfant, NP-MOT jusqu’à 12 ans et plus [Digital scoring tool and standardized scoring of the developmental standardized assessment of the neuro-psychomotor functions of the child, NP-MOT up to 12 years and older]. In Battery of Child Neuro-Psychomotor Functions; Vaivre-Douret, L., Digital ed.; Editions Neuralix: Paris, France, 2021.
- Marquet-Doléac, J., Soppelsa, R., Albaret, J.M. MABC-2 Batterie d’évaluation du mouvement chez l’enfant, 2e éd. [MABC-2 Movement assessment battery for children, 2d ed.]. ECPA-Pearson: Paris, France, 2016.
- Henderson, S.E., Sugden, D.A., Barnett, A. Movement Assessment Battery for Children-2: Movement ABC-2.  Pearson: United Kingdom, 2007.
- Vaivre-Douret, L. Evaluation de la motricité gnosopraxique distale EMG [Assessment of distal gnosopraxic motor skills EMG]. ECPA-Pearson: Paris, France, 1997.
- Vaivre-Douret, L. Normes 8-12 ans pour la notation automatisée de l’évaluation de la motricité gnosopraxique (EMG). In Evaluation de la motricité gnosopraxique distale : révision et adaptation du test de Bergès-Lézine; Vaivre-Douret, L., CD-ROM Ed. ECPA-Pearson: Paris, France, 2021.
- Vaivre-Douret, L. Evaluation de la motricité gnoso-praxique distale chez l’enfant. Une adaptation du test d’imitation de gestes de Bergès-Lézine [Evaluation of the distal motor gnosopraxia. An adaptation of Berges and Lezine's Imitation of Gestures test]. ANAE 1999, 51, 13-20.
- Meljac, C., Fauconnier, E., Scalabrini, J. Schéma corporel-R. Épreuve de Schéma Corporel - Révisée [Body schema-R. Body Schema Test - Revised]. ECPA-Pearson: Paris, France, 2010.
- Frostig, M. Test de développement de la perception visuelle: Manuel [Visual Perception Development Test: Manual]. ECPA-Pearson: Paris, France, 1973.
- Kaufman, A.S., Kaufman, N.L. KABC-II - Batterie pour l’examen psychologique de l’enfant, 2e éd. [KABC-II - Battery for the psychological examination of the child, 2nd ed.]. ECPA-Pearson: Paris, France, 2008.
- Beery, K.E., Buktenica, N.A., Beery, N.A. The Beery–Buktenica Developmental Test of Visual–Motor Integratio : Administration, scoring, and teaching manual, 6th ed. Pearson: London, United Kingdom, 2010.
- Korkman, M., Kirk, U., Kemp, S. NEPSY-II - Bilan neuropsychologique de l’enfant, 2e ed. [NEPSY-II - Neuropsychological assessment of the child, 2nd ed.]. ECPA-Pearson: Paris, France, 2012.
- Wallon, P., Mesmin, C. FCR - Test de la figure complexe de Rey [CFR - Test of the complex figure of Rey]. ECPA-Pearson: Paris, France, 2009.
- Wechsler, D. WISC-IV - Echelle d’intelligence de Wechsler pour enfants et adolescents, 4e éd.) [WISC-IV - Wechsler Intelligence Scale for Children and adolescents, 4th ed.]. ECPA-Pearson: Paris, France, 2005.
- Kohs, S. Test des cubes de Kohs: Manuel d’application [Kohs Cube test: Manual]. ECPA-Pearson: Paris, France, 1972.
- Manly, T., Robertson, I.H., Anderson, V., Mimmo-Smith, I. TEA-Ch - Test d’Évaluation de l’Attention chez l’enfant [TEAC-Ch - Childhood Attention Assessment Test]. ECPA-Pearson: Paris, France, 2006.
- Marquet-Doléac, J., Soppelsa, R., Albaret, J.M. Laby 5-12: Test des labyrinthes pour les enfants de 5 à 12 ans [Laby 5-12: Labyrinths test for children aged 5 to 12]. Éditions Hogrefe: Paris, France, 2010.
- Albaret, J.M., Migliore, L. Stroop - Test d’attention sélective de Stroop [Stroop - Stroop selective attention test]. ECPA-Pearson: Paris, France, 1999.
- Jacquier-Roux, M., Valdois, S., Zorman, M., Lequette, C., Pouget, G. ODEDYS - Outil de dépistage des dyslexies, 2e éd. [ODEDYS - Dyslexia Screening Tool, 2nd ed.]. Cogni-sciences: Grenoble, France, 2005.
- Berndt, D.J., Kaiser, C.F. MDI-C - Échelle composite de dépression pour enfants [MDI-C - Children's Composite Depression Scale]. ECPA-Pearson: Paris, France, 1999.
- Jazz Novo Ober Consulting. Ober Consulting Homepage®, 2018. http://www.ober-consulting.com/9/lang/1/
- Wilson, P.H., Ruddock, S., Smits-Engelsman, B.C.M., Polatajko, H., Blank, R. Understanding performance deficits in developmental coordination disorder: A meta-analysis of recent research. Med. Child Neurol. 2013, 55, 217-228.
- Longcamp, M., Lagarrigue, A., Nazarian, B., Roth, M., Anton, J.L., Alario, F.X. et al. Functional specificity in the motor system: Evidence from coupled fMRI and kinematic recordings during letter and digit writing: Functional Specificity During Letter and Digit Writing. Brain Mapp. 2014, 35, 6077-6087.
- Richards, T.L., Berninger, V.W., Stock, P., Altemeier, L., Trivedi, P., Maravilla, K.R. Differences between good and poor child writers on fMRI contrasts for writing newly taught and highly practiced letter forms. Writ. 2011, 24, 493-516.
- Nicolson, R.I., Fawcett, A.J., Dean, P. Developmental dyslexia: The cerebellar deficit hypothesis. Trends Neurosci. 2001, 24, 508-511.
- Zwicker, J.G., Missiuna, C., Boyd, L.A. Neural Correlates of Developmental Coordination Disorder: A Review of Hypotheses. Child Neurol. 2009, 24, 1273-1281.
- Nicolson, R.I., Fawcett, A.J., Berry, E.L., Jenkins, I.H., Dean, P., Brooks, D.J. Association of abnormal cerebellar activation with motor learning difficulties in dyslexic adults. The Lancet 1999, 353, 1662-1667.
- Fawcett, A.J., Nicolson, R.I. Performance of Dyslexic Children on Cerebellar and Cognitive Tests. Mot. Behav. 1999, 31, 68-78.
- Velay, JL. Interhemispheric sensorimotor integration in pointing movements : A study on dyslexic adults. Neuropsychologia 2002, 40, 827-834.
- Lee, D.N., Lishman, J.R. Visual proprioceptive control of stance. Hum. Mov. Stud. 1975, 1, 87–95.
- Berthoz, A., Lacour, M., Soechting, J.F., Vidal, P.P. The role of vision during linear motion. In Reflex control of posture and movement. Progress in brain research, Vol 50.; Granit, R., Pompeiano, O., eds. Elsevier: Amsterdam, The Netherlands, 1979.
- Gaymard, B., Giannitelli, M., Challes, G., Rivaud-Péchoux, S., Bonnot, O., Cohen, D. et al. Oculomotor Impairments in Developmental Dyspraxia. The Cerebellum 2017, 16, 411-420.
- Robert, M.P., Ingster-Moati, I., Albuisson, E., Cabrol, D., Golse, B., Vaivre-Douret, L. Vertical and horizontal smooth pursuit eye movements in children with developmental coordination disorder. Med. Child Neurol. 2014, 56, 595-600.
- Planton, S., Longcamp, M., Péran, P., Démonet, J.F., Jucla, M. How specialized are writing-specific brain regions? An fMRI study of writing, drawing and oral spelling. Cortex 2017, 88, 66-80.
- Gheysen, F., Van Waelvelde, H., Fias, W. Impaired visuo-motor sequence learning in Developmental Coordination Disorder. Dev. Disabil. 2011, 32, 749-756.
- Ben-Pazi, H., Kukke, S., Sanger, T.D. Poor Penmanship in Children Correlates With Abnormal Rhythmic Tapping: A Broad Functional Temporal Impairment. Child Neurol. 2007, 22, 543-549.
- Collewijn, H., Kowler, E. The significance of microsaccades for vision and oculomotor control. Vis. 2008, 8, 20-21.
- Paillard, J. Le Corps et ses langages d’espace [The body and its languages of space]. In Le corps en psychiatrie [The body in psychiatry]; Jeddi, E., ed. Masson: Paris, France, 1982.
- Pfeiffer, B., Moskowitz, B., Paoletti, A., Brusilovskiy, E., Zylstra, S.E., Murray, T. Developmental Test of Visual–Motor Integration (VMI): An Effective Outcome Measure for Handwriting Interventions for Kindergarten, First-Grade, and Second-Grade Students? J. Occ. Ther. 2015, 69, 1-7.
- Proske, U., Gandevia, S. The Proprioceptive Senses : Their Roles in Signaling BodyShape, Body Position and Movement, and Muscle Force. Reviews 2012, 92, 1651-1697.
- Auzias, M., de Ajuriaguerra, J. Les fonctions culturelles de l’écriture et les conditions de son développement chez l’enfant [The cultural functions of writing and the conditions for its development in children]. Enfance 1986, 39, 145-167.
- Assaiante, C. Action and representation of action during childhood and adolescence: A functional approach. Neurophysiol. 2012, 42, 43-51.
- Assaiante, C., Schmitz, C. Construction des représentations de l’action chez l’enfant : Quelles atteintes dans l’autisme ? [Construction of representations of action in children: What impairment in autism?] Enfance 2009, 1, 111-120.
- Bara, F., Gentaz, E. Haptics in teaching handwriting: The role of perceptual and visuo-motor skills. Mov. Sci. 2011, 30, 745-759.
- Cornhill, H., Case-Smith, J. Factors that relate to good and poor handwriting. J. Occ. Ther. 1996, 50, 732-739.
- Tseng, M.H., Chow, S.M. Perceptual-motor function of school-age children with slow handwriting speed. J. Occ. Ther. 2000, 54, 83–88.
- Weintraub, N., Graham, S. The contribution of gender, orthographic, finger function, and visual–motor processes to the prediction of handwriting status. Ther. J. Res. 2000, 20, 121-141.
- Williams, J., Zolten, A.J., Rickert, V.I., Spence, G.T., Ashcraft, E.W. Use of non verbal tests to screen for writing dysfluency in school-age children. Mot. Ski. 1993, 76, 803-809.
- Yochman, A., Parush, S. Differences in Hebrew handwriting skills between Israeli children in second and third grade. Occup. Ther. Ped. 1998, 18, 53-65.
- Levine, M.D. Developmental variation and learning disorders. Educators Publishing Service: MA, 1987.
- Schneck, C.M. Comparison of pencil-grip patterns in first graders with good and poor writing skills. J. Occ. Ther. 1991, 45, 701–706.
- Hong, S.Y., Jung, N.H., Kim, K.M. The correlation between proprioception and handwriting legibility in children. Phys. Ther. Sci. 2016, 28, 2849-2851.
- Richards, T.L., Berninger, V.W., Fayol, M. FMRI activation differences between 11-years-old good and poor spellers’ access in working memory to temporary and long-term orthographic representations. Neurolinguistics 2009, 22(4), 327-353.
- Van Hoorn, J.F., Maathuis, C.G.B., Hadders-Algra, M. Neural correlates of paediatric dysgraphia. Med. Child Neurol. 2013, 55, 65-68.
